# Chimeric GPCRs mimic distinct signaling pathways and modulate microglia responses

Rouven Schulz [1], Medina Korkut-Demirbaş [1], Alessandro Venturino [1], Gloria Colombo [1] & Sandra Siegert [1] ✉

G protein-coupled receptors (GPCRs) regulate processes ranging from immune responses to neuronal signaling. However, ligands for many GPCRs remain unknown, suffer from off-target effects or have poor bioavailability. Additionally, dissecting cell type-specific responses is challenging when the same GPCR is expressed on different cells within a tissue. Here, we overcome these limitations by engineering DREADD-based GPCR chimeras that bind clozapine-N-oxide and mimic a GPCR-of-interest. We show that chimeric DREADD-β2AR triggers responses comparable to β2AR on second messenger and kinase activity, post-translational modifications, and protein-protein interactions. Moreover, we successfully recapitulate β2AR-mediated filopodia formation in microglia, an immune cell capable of driving central nervous system inflammation. When dissecting microglial inflammation, we included two additional DREADD-based chimeras mimicking microglia-enriched GPR65 and GPR109A. DREADD-β2AR and DREADD-GPR65 modulate the inflammatory response with high similarity to endogenous β2AR, while DREADD-GPR109A shows no impact. Our DREADD-based approach allows investigation of cell type-dependent pathways without known endogenous ligands.

The translation of extracellular signals into an intracellular response is critical for proper tissue function. G protein-coupled receptors (GPCRs) are key mediators in this process with their strategic placement at the cell membrane to bind diverse molecule classes[1,2]. Successful ligand-GPCR interaction triggers intracellular signaling cascades with far-reaching impacts on cell functions like growth, migration, metabolism, and cell–cell communication[3,4]. Approximately 35% of all food and drug administration (FDA)-approved drugs target GPCRs[5,6], stressing their importance for biomedical research and drug development. However, major challenges exist in investigating GPCR signaling. First, several GPCRs have unidentified ligand and are therefore classified as orphan receptors, including more than 100 potential drug targets as well as the majority of olfactory receptors[5,7,8]. Second, GPCR expression and signaling are cell type-specific. For example, β2-adrenergic receptor (β2AR/ADRB2) modulates inflammation in immune cells[9], relaxes smooth muscle in bronchial tubes[10], and impacts pancreatic insulin secretion and hepatic glucose metabolism[11]. Such response diversities hinder dissecting cell type-dependent effects in vivo. Third, GPCR

ligands often suffer from poor bioavailability or cause off-target effects. For instance, norepinephrine acts as ligand for β2AR but can also activate other adrenoceptors in the central nervous system[12]. Therefore, novel strategies are required to overcome the limitations of unknown or unsuitable ligands and simultaneously allow selective investigation of GPCR signaling in a cell type-of-interest.

So far, over 800 GPCRs are known, which are structurally conserved with seven transmembrane helices (TM) connected by three extracellular (ECL) and intracellular (ICL) loops[13]. Ligand binding involves N-terminus, ECLs, and TM domains and consequently triggers ICL interaction with heterotrimeric G proteins. These G proteins are composed of α- and βγ-subunits that act as effectors on downstream signaling partners[13]. Specific subunit recruitment of either $G\alpha_s$, $G\alpha_q$, or $G\alpha_i$ activates defined canonical pathways[14]. Besides ICLs as critical components for proper GPCR signal transduction[15–17], the C-terminus interacts with β-arrestins, which contribute to receptor desensitization[18–20] and kinase recruitment[21–25]. Several studies have exploited the concept of ligand binding and signaling domains to

[1]Institute of Science and Technology Austria (ISTA), Am Campus 1, 3400 Klosterneuburg, Austria. ✉e-mail: ssiegert@ist.ac.at

control GPCR function[26–31]. Airan et al. generated light-inducible GPCR chimeras that mimic the signaling cascades of distinct GPCRs[27]. However, caveats exist with this optogenetic approach, as it relies on strong light stimulation, which induces phototoxicity[32–35]. Additionally, light exposure in vivo requires invasive procedures that will disrupt tissue integrity and alter the response of resident immune cells[36]. Yet, immune cells are interesting targets for studying GPCR signaling as their function and ability to induce inflammation is tightly controlled by these receptors[3,37,38]. Several immune cells such as circulating leukocytes and lymphocytes are not confined to any light-accessible tissue and therefore cannot be manipulated through light-inducible GPCRs.

Here, we designed chemical-inducible GPCR chimeras based on the DREADD system (Designer Receptor Exclusively Activated by Designer Drugs)[39–41]. DREADDs are modified muscarinic acetylcholine receptors, which are inert to their endogenous ligand acetylcholine and respond to clozapine-N-oxide (CNO), a small injectable compound with minimal off-target effects and suitable bioavailability for in vivo usage[42]. The DREADDs hM3Dq and hM4Di are frequently employed to manipulate neuronal activity[41], whereas rM3Ds has been designed to induce a $G\alpha_s$ response[43]. In our approach, we identified the ligand binding and signaling regions of hM3Dq and 292 other GPCRs. This enabled us to engineer CNO-responsive chimeras with β2AR being our proof-of-concept candidate due to its well-known ligands and broad physiological importance[10,11], which includes modulating inflammation in various immune cells[9] such as microglia[44]. By exchanging the corresponding signaling domains we obtained chimeric hM3Dq-β2AR, now referred to as DREADD-β2AR, which fully recapitulated the signaling pathways of levalbuterol-stimulated non-chimeric β2AR including the impact on microglia motility[45]. Finally, we identified immunomodulatory effects of DREADD-β2AR and two additionally generated DREADD chimeras for the microglia-enriched GPR65 and GPR109A/HCAR2. This underlines that our approach can be applied to different GPCRs-of-interest allowing cell type-targeted manipulation of GPCR signaling. In our study, we offer a straightforward design for CNO-responsive chimeras to mimic a GPCR-of-interest. This will be especially useful to study GPCRs with yet unidentified pathways, orphan receptors with unknown ligands[5,7,8], or GPCRs with non-canonical signaling properties that might not be captured by available DREADDs.

## Results

### Establishing a library for in-silico design of DREADD-based GPCR chimeras

Microglia are tissue-resident macrophages of the central nervous system. They maintain homeostasis during physiological conditions and induce an inflammatory response upon tissue damage and pathogen encounter[46,47]. GPCRs are critical for these functions as they allow fast adaption to local perturbations. To identify which GPCRs are selectively enriched in microglia, we compared GPCR expression across different cell types in a previously established retina transcriptome database[48]. We found approximately one-third of the most abundant GPCRs enriched in microglia, which also included the well-defined β2-adrenergic receptor (β2AR)[9–11,44,45,49], making it a prime candidate for establishing our strategy (Fig. 1a).

To design CNO-responsive DREADD-based chimeras mimicking a GPCR-of-interest (Fig. 1b), we first identified GPCR ligand binding and signaling domains including either N-terminus, extracellular loops (ECLs) and transmembrane helices (TMs), or intracellular loops (ICLs) and C-terminus, respectively (Fig. 1c). We performed multiple protein sequence alignment using the established domains of bovine rhodopsin (RHO)[27] as reference. We aligned rhodopsin with the CHRM3-based hM3Dq, human β2-adrenergic receptor (β2AR), and 292 other potential GPCRs-of-interest (Fig. 1d, Supplementary Data 1). As internal controls, we included human α1-adrenergic receptor (hα1AR) and

hamster β2-adrenergic receptor (hamβ2AR) and confirmed that our alignment successfully reproduced the rhodopsin-based chimeras from Airan et al.[27]. To further verify alignment accuracy, we utilized the TMHMM algorithm, which predicts TM domains in a protein sequence[50]. For all GPCRs shown in Fig. 1d and Supplementary Fig. S1, predicted TMs displayed the expected tight flanking of ICLs and C-termini identified by our alignment. The only exception was GPR109A, where TMHMM failed to identify the seventh TM with sufficient certainty. Occasionally, minor deviations from seamless flanking occurred but were within the observed range for the reference and internal controls (Supplementary Fig. S1). Finally, we exploited published crystal structures for three key GPCRs: the alignment reference RHO; hM3Dq, for which we used rat CHRM3 as surrogate with over 90% sequence similarity; and our primary GPCR-of-interest β2AR. We mapped our identified ligand binding and signaling domains together with predicted TMs on these crystal structures and found that they closely matched the expected extracellular, transmembrane, or intracellular locations (Fig. 1e). These results suggest that our alignment correctly predicts GPCR domains and serves as a library for generating DREADD-based GPCR chimeras.

### Engineering chimeric DREADD-β2AR

Next, we designed our first CNO-inducible GPCR chimera DREADD-β2AR in-silico by combining hM3Dq ligand binding and β2AR signaling domains (Fig. 2a). Additionally, we introduced two modifications to the N-terminus: a hemagglutinin-derived signal peptide[51,52] followed by a VSV-G epitope[53] to probe for cell surface expression. The signal peptide supports co-translational import into the endoplasmic reticulum and subsequent plasma membrane incorporation[51,52]. Neither DREADD nor β2AR contain such a peptide sequence according to the SignalP algorithm[54] (Supplementary Fig. S2a). When we re-analyzed both GPCRs after introducing our N-terminal modifications in-silico, SignalP identified the signal peptide and its cleavage site upstream of the VSV-G tag (Supplementary Fig. S2b). We synthesized the DREADD-β2AR coding sequence and cloned it into a mammalian expression vector utilizing the ubiquitous CMV promoter. Then, we transfected HEK cells and after 24 h performed immunostaining for the VSV-G tag under non-permeabilizing conditions. We confirmed that DREADD-β2AR successfully incorporated into the cell membrane based on the strong VSV-G signal (Fig. 2b), whereas non-transfected HEK cells lacked this staining (Supplementary Fig. S2c). For comparison, we also synthesized hM3Dq, non-chimeric β2AR, rM3Ds, and hM4Di containing the same N-terminal modifications. In all cases, we detected successful surface expression (Supplementary Fig. S2d–g).

### Functional validation of chimeric DREADD-β2AR

To validate DREADD-β2AR functionality, we investigated whether CNO stimulation mimics the signaling pathways of non-chimeric β2AR as outlined in Fig. 2c. First, we focused on the induction of second messenger cascades. β2AR is classically known to recruit $G\alpha_s$ upon ligand binding, resulting in an increase of cytoplasmic cAMP due to adenylyl cyclase (AC) activation[55] (Fig. 2d). We co-transfected HEK cells with DREADD-β2AR and a modified firefly luciferase that increases luminescence in the presence of cAMP[56]. Indeed, we found a CNO-dose-dependent increase in cAMP with DREADD-β2AR (Fig. 2e), which was not observed in cells transfected with empty vector backbone. For comparison, we transfected cells with non-chimeric β2AR and applied the selective β2AR-agonist levalbuterol[57,58]. DREADD-β2AR and non-chimeric β2AR elicited a similar fold change around 25 when stimulated with their respective ligand at a 10 μM concentration (Fig. 2f), which we subsequently used for all further assays unless specified otherwise. As a note, HEK cells endogenously express β2AR[59], which explains the partial response of empty vector-transfected cells to levalbuterol. Supplementary Fig. S3a shows that endogenous β2AR

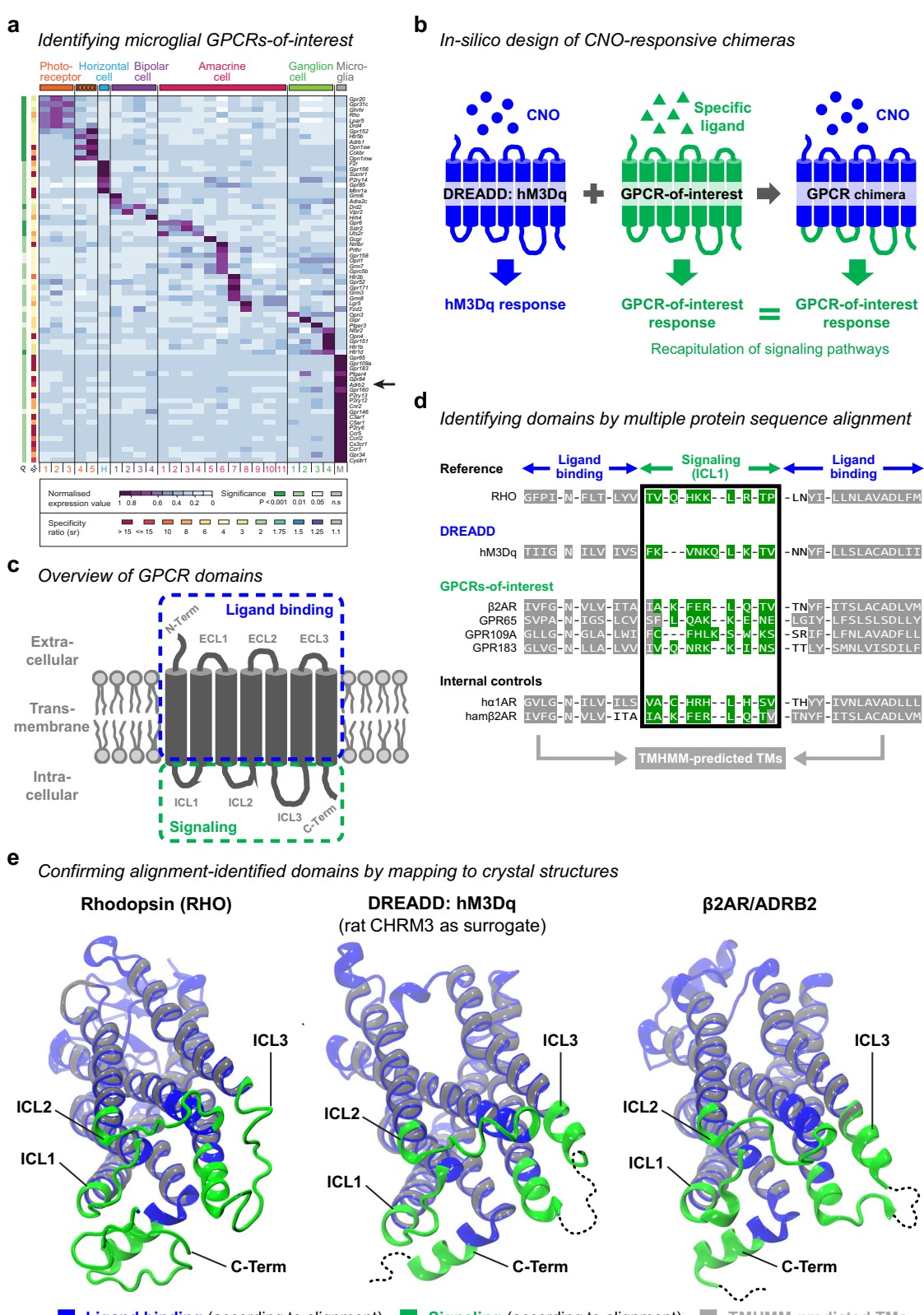

**a** *Identifying microglial GPCRs-of-interest*

**b** *In-silico design of CNO-responsive chimeras*

**c** *Overview of GPCR domains*

**d** *Identifying domains by multiple protein sequence alignment*

**e** *Confirming alignment-identified domains by mapping to crystal structures*

**Rhodopsin (RHO)**  **DREADD: hM3Dq** (rat CHRM3 as surrogate)  **β2AR/ADRB2**

■ **Ligand binding** (according to alignment)  ■ **Signaling** (according to alignment)  ■ **TMHMM-predicted TMs**

contributes to cAMP induction in a ligand dose-dependent manner. CNO stimulation of the $G\alpha_q$-coupled hM3Dq only triggered a comparatively minor 2.5-fold increase (Fig. 2f). As a control, we also included chimeric rM3Ds, which was generated by replacing ICL2-3 of rat-derived M3Dq with corresponding turkey β1-adrenergic receptor (β1AR) domains to facilitate strong $G\alpha_s$-coupling and induction of cAMP synthesis[43]. As expected, rM3Ds raised cAMP levels upon CNO

application, resulting in a 12-fold increase that did not reach the extent of DREADD-β2AR or non-chimeric β2AR (Fig. 2g). The DREADD-β2AR and rM3Ds responses were both saturated at 10 µM CNO (Supplementary Fig. S3b), suggesting that the data in Fig. 2e–g reflect their maximal cAMP induction capabilities. hM4Di also significantly elevated cAMP by approximately 2.5-fold in our HEK cell assay (Fig. 2g), despite being commonly described as $G\alpha_i$-coupled receptor expected

**Fig. 1 | GPCR domains can be identified in-silico to engineer chimeric receptors.**
**a** GPCR gene expression analysis across different cell types in the mouse retina.
Columns represent distinct cell types and show clusters of selectively enriched
GPCRs. Purple indicates high gene expression. *P*-value and specificity ratio (s.r.) are
color-coded as indicated in the figure. Arrow points to β2AR/ADRB2. **b, c** Schematic
of GPCR domains and chimera design. **b** The intracellular domains of the DREADD
hM3Dq (blue) are replaced with the intracellular domain of a GPCR-of-interest
(green), generating a chimeric receptor that induces the signaling cascade of the
GPCR-of-interest (green) upon CNO stimulation. **c** GPCRs consist of seven trans-
membrane domains, three extracellular loops (ECL1–3), three intracellular loops
(ICL1–3), the N-terminus (N-Term) and C-terminus (C-Term). Ligand binding (blue)
involves extracellular and transmembrane domains and consequently triggers
conformational changes, which are transmitted to intracellular domains for

induction of signaling cascades (green). **d** Zoomed-in view on the multiple protein
sequence alignment. Bovine rhodopsin (RHO) served as Ref. 27. to identify ligand
binding and signaling domains of CHRM3-based hM3Dq, the human β2-adrenergic
receptor (β2AR), three out of 292 GPCRs-of-interest (see Supplementary Data 1),
human α1-adrenergic receptor (hα1AR) and hamster β2-adrenergic receptor
(hamβ2AR). In gray: TMHMM-predicted transmembrane helices. In green: signaling
domain for the first intracellular loop (ICL1). **e** Crystal structures representing
bovine RHO, rat CHRM3 as surrogate for hM3Dq, and human β2AR. Ligand binding
domains (blue), signaling domains (green), and TMHMM-predicted sequences
(gray) are highlighted. Structural representations are rotated with the intracellular
domains facing the screen. Dotted lines: sequences not available within the crystal
structures.

to decrease cAMP[39]. Our data shows that DREADD-β2AR successfully
recapitulated the Gα$_s$-induced cAMP upregulation of non-chimeric
β2AR. Importantly, hM3Dq alone was clearly distinguishable through
its marginal impact on cAMP levels, indicating that the DREADD-β2AR
response was mediated through properly identified β2AR signaling
domains.

Next, we investigated kinase activity (Fig. 2c). Gα$_q$-coupled GPCRs
trigger the mitogen-activated protein kinase (MAPK) pathway and
induce transcription through a serum responsive element (SRE)[60].
Therefore, we measured luciferase activity driven by an SRE reporter
(Fig. 2h). As anticipated, HEK cells transfected with hM3Dq increased
luciferase activity 2.5-fold upon stimulation with CNO compared to
vehicle (Fig. 2i). We hypothesized that this effect would be absent in
DREADD-β2AR-transfected HEK cells. Indeed, DREADD-β2AR did not
increase luciferase activity; instead, the activity decreased more than
4-fold, which was similar to the non-chimeric β2AR response upon
levalbuterol treatment. In empty vector-transfected cells, CNO had no
impact on SRE-dependent reporter transcription, while levalbuterol
reduced luciferase activity due to endogenous β2AR expression in HEK
cells[59]. Supplementary Fig. S3c demonstrates the contribution of
endogenous β2AR which depends on the ligand concentration. rM3Ds
and hM4Di also inhibited the SRE reporter signal by 2-fold (Fig. 2i). The
opposing responses with DREADD-β2AR and hM3Dq further sub-
stantiate the correct identification of β2AR signaling domains.

Several GPCRs possess constitutive activity[61] and can initiate sig-
naling pathways even in the absence of ligand stimulation (Fig. 3a). To
evaluate constitutive signaling, we used a cAMP-dependent luciferase
assay suitable for measuring baseline activity[62] and recorded lumi-
nescence for 30 min. Consistent with previous reports[43], we found
elevated cAMP levels in rM3Ds-transfected HEK cells compared to
empty vector controls (Fig. 3b, c). DREADD-β2AR only increased cAMP
3-fold, suggesting less constitutive activity. Notably, HEK cells trans-
fected with non-chimeric β2AR also displayed baseline activity
exceeding that of DREADD-β2AR. This is in accordance with previous
studies that found constitutive signaling in several non-chimeric
GPCRs[61]. hM3Dq and hM4Di did not impact the cAMP baseline. We also
evaluated constitutive activity on the MAPK pathway and compared
baseline SRE reporter signals in GPCR-transfected HEK cells with
empty vector controls (Fig. 3d). Again, rM3Ds showed the most pro-
nounced effect with a 15-fold decrease of SRE reporter activity (Fig. 3e).
In comparison, DREADD-β2AR and non-chimeric β2AR only moder-
ately inhibited the SRE reporter by approximately 3- and 6-fold,
respectively. Interestingly, hM3Dq also caused a small but significant
1.5-fold inhibition of baseline SRE activity while hM4Di had no impact.
These results confirm the constitutive activity of rM3Ds[43] and indicate
that DREADD-β2AR has a comparatively lower tendency to initiate
signaling pathways in the absence of CNO stimulation.

Next, we focused on protein-protein interactions and post-
translational modifications regulated by β2AR signaling (Fig. 2c).
Ligand-activated β2AR recruits β-arrestin 2, which creates a scaffold
for attracting signaling kinases[25] and further plays a role in receptor

internalization[19]. To investigate the interaction between β-arrestin 2
and non-chimeric β2AR or DREADD-β2AR, we attached the com-
plementary luciferase subunits LgBiT and SmBiT to their C-termini,
respectively[63–65]. Upon β-arrestin 2 recruitment, both subunits are
brought into close proximity, resulting in a bioluminescent signal
(Fig. 4a). Levalbuterol stimulation of non-chimeric β2AR, as well as
CNO stimulation of DREADD-β2AR, immediately increased biolumi-
nescence compared to vehicle treatment (Fig. 4b). This indicates that
DREADD-β2AR recapitulates the fast β-arrestin 2 recruitment observed
with non-chimeric β2AR.

β2AR signaling also involves the rapid phosphorylation of
extracellular signal-regulated kinases 1 and 2 (ERK1/2), which is partly
mediated through recruitment of β-arrestins[25,55]. So, we investigated
whether ERK1/2 phosphorylation occurred in HEK cells transfected
with non-chimeric β2AR or DREADD-β2AR following treatment with
levalbuterol or CNO, respectively. For both constructs, phosphor-
ylation peaked two minutes after ligand stimulation and gradually
declined after five minutes (Fig. 4c, Supplementary Fig. S4), sug-
gesting the recapitulation of post-translational modification
dynamics. CNO exposure of empty vector-transfected HEK cells did
not impact ERK1/2 phosphorylation. As a note, we also found con-
stitutive ERK1/2 phosphorylation in the absence of ligand in both
non-chimeric β2AR and DREADD-β2AR when compared to their
empty vector controls (Supplementary Fig. S5).

β-arrestin recruitment also mediates GPCR internalization[18–20],
which provides a regulatory feedback loop for receptor activity after
ligand stimulation (Fig. 2c)[66,67]. To visualize receptor trafficking, we
engineered DREADD-β2AR with EGFP attached at the C-terminus
(Fig. 5a, b). We transfected this construct into HEK cells and 24 h later
incubated them for 30 min with anti-VSV-G antibody to distinguish cell
surface-incorporated DREADD-β2AR from receptors retained within
the cell. Colocalization of VSV-G antibody and EGFP occurred on the
cell surface. We barely found VSV-G signal within transfected cells
suggesting that DREADD-β2AR internalization is largely absent without
ligand stimulation (Fig. 5c). In contrast, when we applied CNO for
either 15, 30, or 60 min following VSV-G antibody labeling, VSV-G/EGFP
signals colocalized within the cytoplasm. Internalization increased
after 30 min and became significantly higher after 60 min of CNO
exposure compared to vehicle treatment (Fig. 5d–f). We conclude that
DREADD-β2AR can undergo ligand-induced receptor internalization.

Together, our results confirm that DREADD-β2AR successfully
recapitulates the signaling cascades (Fig. 2c) of non-chimeric β2AR
with similar dynamics.

### Chimeric DREADD-β2AR recapitulates β2AR-mediated effects on microglia motility

Microglia are highly motile cells that constantly scan their environ-
ment for signs of disrupted tissue homeostasis. Activation of β2AR
signaling was recently shown to rapidly induce filopodia formation as a
consequence of elevated cAMP levels[45]. Indeed, when we performed
live-imaging of primary microglia cultures, we confirmed filopodia

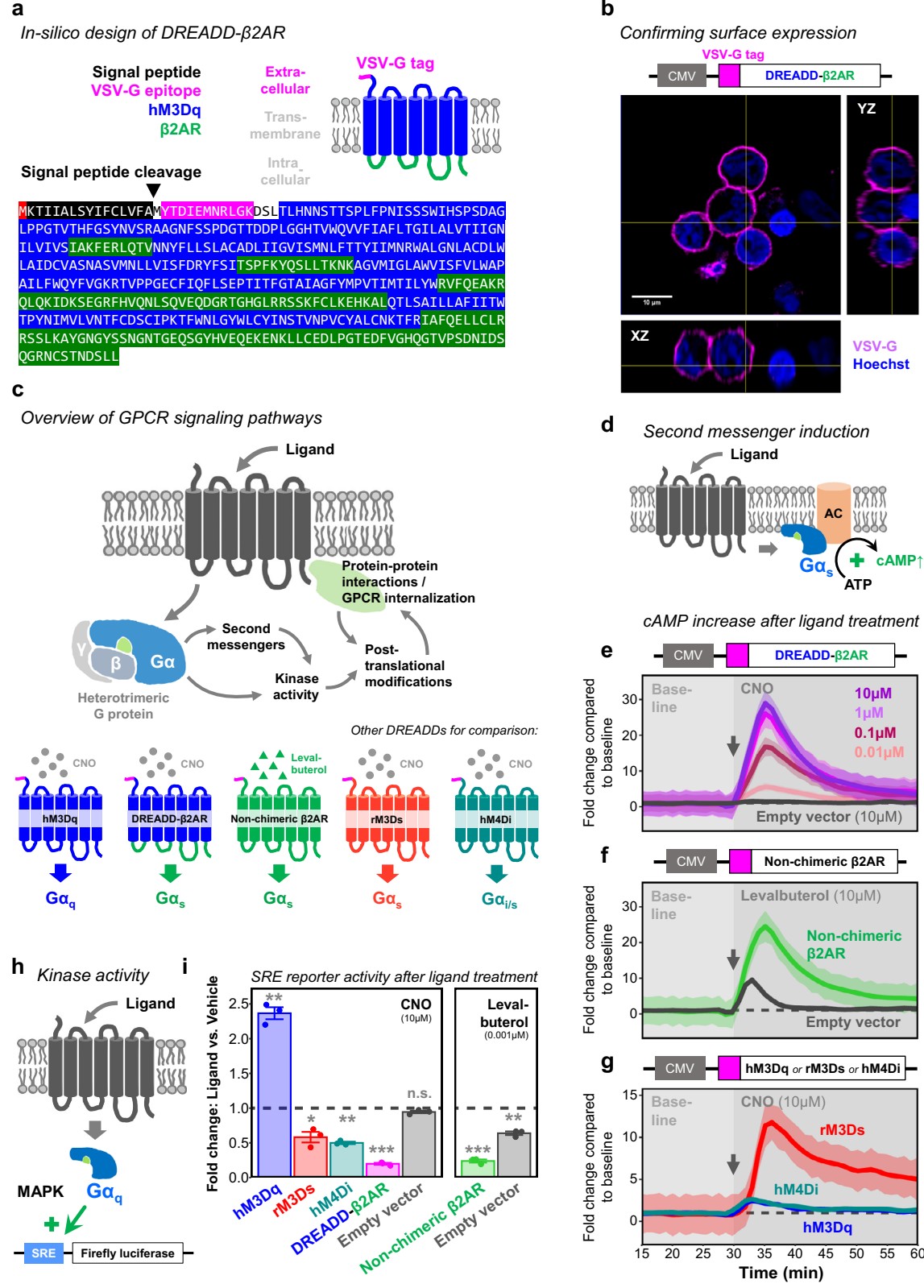

extension and an increase in total microglia area after levalbuterol application (Fig. 6). During the first 10 min of baseline recordings, microglia were motile and changed their area only marginally. After levalbuterol stimulation, the cell area significantly increased throughout the following 45 min of imaging compared to the baseline (Fig. 6a, e). To recapitulate this phenotype with our DREADD-β2AR, we first generated a bi-cistronic GPCR-P2A-EGFP vector containing a self-

cleaving P2A peptide site[68] that allows simultaneous GPCR and cytoplasmic EGFP expression (Supplementary Fig. S6a). We transfected HEK cells with this DREADD-β2AR-P2A-EGFP vector and confirmed the expected cytoplasmic EGFP localization co-existing with anti-VSV-G immunostaining on the cell membrane (Supplementary Fig. S6b). Subsequently, we packaged our DREADD-β2AR-P2A-EGFP construct into lentiviral vectors and transduced primary microglia. Successful

**Fig. 2 | DREADD-β2AR recapitulates second messenger induction and MAPK activity of non-chimeric β2AR. a** Schematic of DREADD-β2AR and corresponding protein sequence encoding for signal peptide (black), VSV-G epitope (magenta), hM3Dq ligand binding domains (blue), and β2AR signaling domains (green). Black arrow: start of the mature GPCR after post-translational cleavage of the signal peptide. **b** Orthogonal view of DREADD-β2AR-transfected HEK cells immunostained for the N-terminal VSV-G tag under non-permeabilizing conditions. Magenta: VSV-G tag. Blue: nuclear staining with Hoechst. CMV, human cytomegalovirus promoter. **c** Schematic of signaling pathways for functional validation of the DREADD-based chimeras. The heterotrimeric G protein consists of an α- and βγ-subunit. Below: hM3Dq is a $G\alpha_q$-coupled receptor, whereas non-chimeric β2AR recruits G proteins with a $G\alpha_s$ subunit. DREADD-β2AR contains the β2AR signaling domains to recruit $G\alpha_s$. The DREADDs rM3Ds couples to $G\alpha_s$ and hM4Di is associated with $G\alpha_i$. **d** Schematic of $G\alpha_s$-coupled GPCR inducing cAMP synthesis after ligand stimulation through adenylyl cyclase (AC) activation. **e–g** Real-time measurement of cAMP-dependent luciferase activity in HEK cells transfected with DREADD-β2AR (**e**); non-chimeric β2AR (**f**), hM3Dq, rM3Ds or hM4Di (**g**); or empty vector (**e–f**). Baseline measurements of 30 min (first 15 min not shown) followed by ligand application (gray arrow for onset) of either CNO or levalbuterol. Measure of center: Mean fold change compared to baseline mean (dashed line) in the same experimental repetition. Ribbons: 95% confidence intervals. N = four (DREADD-β2AR: CNO 0.1–10 μM), seven (Empty vector: CNO), four (Non-chimeric β2AR: Levalbuterol; Empty vector: Levalbuterol; hM3Dq: CNO; hM4Di: CNO), or three (rM3Ds: CNO) experimental repetitions. Source data are provided as a Source Data file. **h** Schematic of $G\alpha_q$-coupled GPCR engaging in the mitogen-activated protein kinase (MAPK) pathway which induces transcription of a firefly luciferase reporter from a serum responsive element (SRE). **i** Endpoint measurement of SRE-dependent luciferase activity in HEK cells transfected with hM3Dq (blue), DREADD-β2AR (magenta), non-chimeric β2AR (green), rM3Ds (red), hM4Di (cyan), or empty vector (gray). Ligand stimulation either with 10 μM CNO (left) or 0.001 μM levalbuterol (right). Dashed line: level of vehicle control. Error bars: standard error of the mean. Two-sided one-sample T-test for comparing to a mean of 1 representing the vehicle control: \*\*\*$p < 0.001$; \*\*$p < 0.01$; $^{n.s}p > 0.05$. Exact p-values of individual T-tests without multiple testing correction: $p = 0.004$ (hM3Dq: CNO); $p = 0.03$ (rM3Ds: CNO); $p = 0.001$ (hM4Di: CNO); $p < 0.002$ (DREADD-β2AR: CNO); $p = 0.09$ (Empty vector: CNO); $p < 0.001$ (Non-chimeric β2AR: Levalbuterol); $p = 0.01$ (Empty vector: Levalbuterol). N = three experimental repetitions. Source data are provided as a Source Data file.

transduction was sparse but individual cells could be clearly identified by their EGFP expression (Supplementary Fig. S6c). When we imaged these EGFP-positive cells, we found that CNO application induced filopodia formation (Fig. 6b, e) similar to levalbuterol. Non-transduced microglia stimulated with either vehicle or CNO did not significantly increase their area (Fig. 6c–e). Supplementary Fig. S6d provides a statistical comparison across all experiment groups at each indicated time point and confirms that levalbuterol treatment and DREADD-β2AR are significantly different from the control conditions. It is worth mentioning that filopodia extension in cultured microglia does not present the complexity of microglial process dynamics observed in vivo[45,49]. Yet, given the difficulty of microglial transduction in vivo[69], our simpler but more accessible in vitro system suggests that DREADD-β2AR successfully mimics β2AR signaling in microglia and modulates their function.

## Generating DREADD-based chimeras for additional microglial GPCRs-of-interest

After confirming the functionality of our strategy with DREADD-β2AR, we decided to extend our approach to GPR65 and GPR109A/HCAR2, which like β2AR, showed microglia-enriched gene expression (Fig. 1a). GPR65 and GPR109A respond to protons[70] and ketone bodies[71], respectively, and were shown to modulate inflammatory responses such as cytokine expression in microglia in vitro systems[72,73]. Both of their ligands are prone to cause off-target effects as acidic environments trigger various unpredictable responses in immune cells[74,75], and the ketone β-hydroxybutyrate can impact histone modification in a GPCR-independent manner[76,77]. This makes GPR65 and GPR109A interesting candidates for DREADD-based chimeras to dissect their inflammatory role with a well-defined ligand.

Thus, we designed DREADD-GPR65 and DREADD-GPR109A with the same N-terminal modifications as DREADD-β2AR (Fig. 2a). First, we transfected HEK cells with these chimeras and confirmed successful cell membrane incorporation through immunostaining for the VSV-G tag (Fig. 7a, b). Then, we investigated whether both chimeras triggered their expected second messenger cascades and kinase activity. Like β2AR, GPR65 belongs to the $G\alpha_s$-coupling family[70]. Therefore, we applied our previously established validation strategy for second messenger induction (Fig. 2d–e). We measured cAMP levels in HEK cells transfected with DREADD-GPR65 and found a significant increase after CNO stimulation, which was not detected in empty vector-transfected cells (Fig. 7c). Stimulation of DREADD-GPR65 also impacted the MAPK pathway and reduced SRE-mediated reporter expression, similar to β2AR (Figs. 2h, i, 7d).

In contrast to GPR65 and β2AR, GPR109A couples to $G\alpha_i$ and suppresses cAMP synthesis by inhibiting adenylyl cyclase (AC)[71] and therefore competes with the AC activator forskolin[78,79] (Fig. 7e). To measure $G\alpha_i$-mediated decreases in cAMP, we adapted a cAMP-dependent luciferase assay with kinetics suitable for $G\alpha_i$-signaling[62]. Within 10 min following CNO stimulation, DREADD-GPR109A-transfected HEK cells decreased cAMP levels by approximately 15% compared to vehicle (Fig. 7e). After 30 min of CNO exposure, we added forskolin as a competing component to induce cAMP synthesis. DREADD-GPR109A-transfected cells exposed to CNO kept their cAMP signal approximately 15% below the vehicle control suggesting robust AC inhibition. Empty vector-transfected HEK cells did not respond to CNO, and their cAMP levels always remained at vehicle control levels (Fig. 7e, see Supplementary Fig. S7a for non-normalized values).

To further substantiate the $G\alpha_i$ effect, we tested for the ability of DREADD-GPR109A to compete with $G\alpha_s$ signaling (Fig. 7f). For this, we used a reporter that drives luciferase expression through a cAMP-responsive element (CRE)[60], which is induced by $G\alpha_s$ activity. First, we confirmed successful $G\alpha_s$ induction in empty vector-transfected HEK cells through stimulation with 5′-N-ethylcarboxamidoadenosine (NECA), a potent agonist of the endogenously expressed $G\alpha_s$-coupled A2B adenosine receptor (A2BAR)[80]. The expression of the CRE reporter was NECA dose-dependent and reached saturation at 5 μM, while concomitant CNO application did not interfere (Supplementary Fig. S7b). Subsequently, we transfected HEK cells with DREADD-GPR109A and applied CNO together with 5 μM NECA. As anticipated, CNO significantly inhibited $G\alpha_s$-mediated transcription from the CRE reporter by approximately 20% when compared to vehicle (Fig. 7g). As a note, CNO stimulation of DREADD-GPR109A did not dampen CRE reporter activity without simultaneous induction through NECA (Supplementary Fig. S7c), possibly because the assay is not suitable for detecting minor reductions from baseline levels[62]. In addition, stimulation of DREADD-GPR109A had no impact on the MAPK pathway measured through SRE reporter activity (Supplementary Fig. S7d). Even though DREADD-GPR65 and DREADD-GPR109A differed in their second messenger and kinase activity, both GPCRs were able to recruit β-arrestin 2 upon CNO application, emphasizing that individual GPCRs can display diverse signaling patterns (Supplementary Fig. S7e). Lastly, we evaluated constitutive signaling of both chimeras as previously done (Fig. 3) by measuring baseline cAMP levels and MAPK activity in the absence of CNO. DREADD-GPR65 increased baseline cAMP comparable to rM3Ds, while DREADD-GPR109A was indistinguishable from empty vector controls (Supplementary Fig. S8a–c). When assessing constitutive MAPK signaling, DREADD-GPR65 performed similarly to DREADD-β2AR and induced less SRE reporter inhibition than rM3Ds, while DREADD-GPR109A displayed no activity when compared to empty vector controls (Supplementary Fig. S8d, e). In conclusion, the results

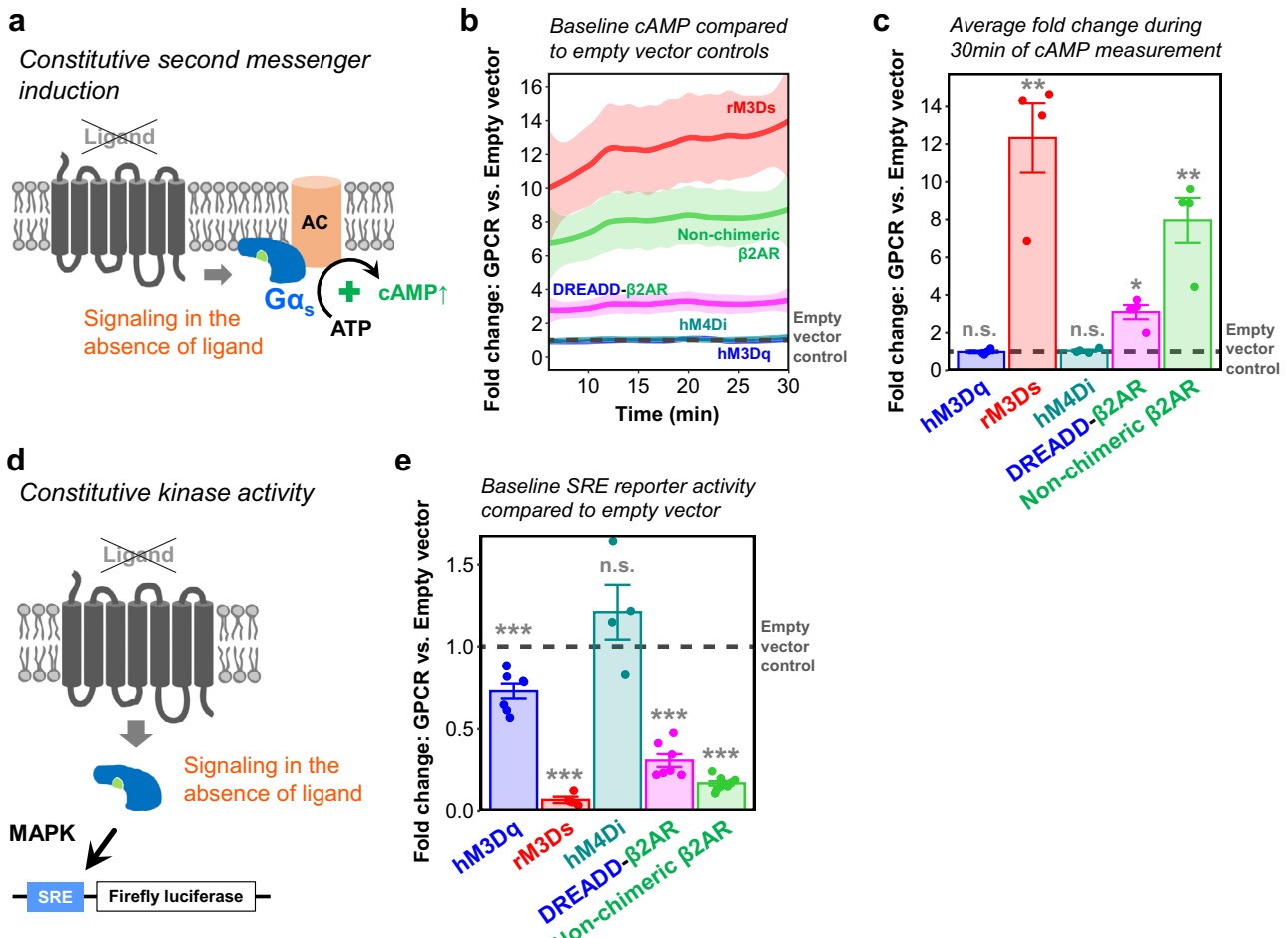

**Fig. 3 | DREADD-β2AR displays lower constitutive activity compared to rM3Ds and non-chimeric β2AR. a** Schematic of GPCR increasing baseline cAMP levels through constitutive activity. **b** Real-time measurement of cAMP-dependent luciferase activity during a 30 min baseline in HEK cells transfected with hM3Dq (blue), rM3Ds (red), hM4Di (cyan), DREADD-β2AR (magenta), or non-chimeric β2AR (green). Measure of center: Mean fold change compared to empty vector (dashed line) in the same experimental repetition. Ribbons: 95% confidence intervals. $N =$ four experimental repetitions. Source data are provided as a Source Data file. **c** Graph shows average fold changes compared to empty vector control during the 30 min measurement in (**b**). Dashed line: level of empty vector control. Error bars: standard error of the mean. $N =$ four experimental repetitions. Two-sided one-sample $T$-test for comparing to a mean of 1 representing the empty vector control: ***$p < 0.001$; **$p < 0.01$; n.s.$p > 0.05$. Exact $p$-values of individual $T$-tests without

multiple testing correction: $p = 0.81$ (hM3Dq); $p = 0.009$ (rM3Ds); $p = 0.48$ (hM4Di); $p = 0.01$ (DREADD-β2AR); $p = 0.009$ (Non-chimeric β2AR). Source data are provided as a Source Data file. **d** Schematic of GPCR with constitutive activity impacting baseline MAPK signaling measured through an SRE reporter. **e** Endpoint measurement of SRE-dependent luciferase activity in transfected HEK cells. Dashed line: level of empty vector control. Error bars: standard error of the mean. $N =$ seven (hM3Dq, DREADD-β2AR), four (rM3Ds, hM4Di), or nine (Non-chimeric β2AR) experimental repetitions. of four to nine repetitions. Two-sided one-sample $T$-test for comparing to a mean of 1 representing the empty vector control: ***$p < 0.001$; **$p < 0.01$; n.s.$p > 0.05$. Exact p-values of individual $T$-tests without multiple testing correction: $p < 0.001$ (hM3Dq); $p < 0.001$ (rM3Ds); $p = 0.30$ (hM4Di); $p < 0.001$ (DREADD-β2AR); $p < 0.001$ (Non-chimeric β2AR). Source data are provided as a Source Data file.

suggest that our DREADD-based strategy is reproducible and can be extended to other GPCRs-of-interest.

**Using DREADD-GPCRs to investigate microglia function**
Finally, we utilized our DREADD-based chimeras to investigate functional consequences of GPCR signaling in a microglia context. We took advantage of the microglia-like cell line HMC3[81], which allows generation of cell lines with stable DREADD-GPCR expression. This provides a homogeneous cell population and is advantageous for reliable quantification of GPCR responses, which cannot be achieved in primary microglia due to suboptimal transduction efficiencies with available vectors[82] (Supplementary Fig. S6c). Thus, we cloned and packaged each DREADD-GPCR-P2A-EGFP construct into genome-integrating lentiviral vectors, transduced HMC3 cells, and fluorescence-activated cell sorted for EGFP-positive cells. We confirmed successful incorporation of GPCR chimeras in EGFP-expressing

HMC3 cells through VSV-G immunostaining under non-permeabilizing conditions (Supplementary Fig. S9a–c). In parallel, we also confirmed with quantitative reverse transcription PCR (RT-qPCR) that β2AR endogenously occurs in HMC3 cells at moderate mRNA levels (Supplementary Fig. S10a), allowing stimulation with the selective β2AR agonist levalbuterol.

Subsequently, we investigated whether our GPCRs can induce $Ca^{2+}$ signaling, which occurs in microglia upon sensing perturbations in their neuronal tissue environment[83]. We imaged HMC3 cell lines after incubation with a $Ca^{2+}$-sensitive fluorescent dye and applied either levalbuterol, CNO, vehicle, or ATP as positive control, which is known to trigger $Ca^{2+}$ transients in microglia[83]. During six minutes of recording, HMC3 cells commonly displayed spontaneous $Ca^{2+}$ currents (Supplementary Fig. S11a), evidenced by fluctuations in fluorescence intensity and software-based[84] $Ca^{2+}$ peak detection. ATP treatment resulted in rapid and synchronized accumulation of $Ca^{2+}$ events

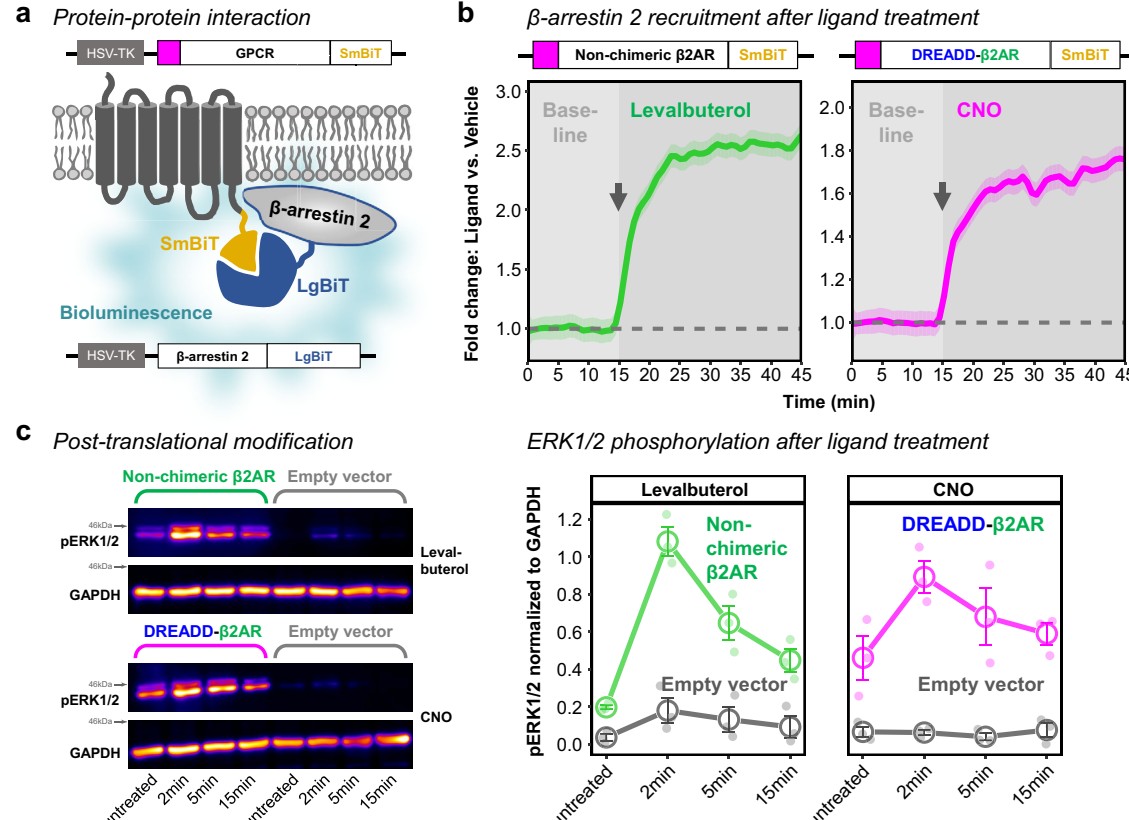

**Fig. 4 | DREADD-β2AR recruits β-arrestin 2 and phosphorylates ERK1/2.**
**a** Schematic of induced bioluminescence upon GPCR-SmBiT and β-arrestin 2-LgBiT interaction. GPCR, G protein-coupled receptor. HSV-TK, herpes simplex virus thymidine kinase promoter. **b** Real-time measurement of bioluminescence in HEK cells transfected with non-chimeric β2AR (left) or DREADD-β2AR (right). Baseline measurements followed by ligand application (gray arrow shows onset) of either levalbuterol (left) or CNO (right). Measure of center: Mean of baseline-normalized fold change compared to vehicle (dashed line) in the same experimental repetition. Ribbons: 95% confidence intervals. N = four (Non-chimeric β2AR) or five (DREADD-β2AR) experimental repetitions. Source data are provided

as a Source Data file. **c** Phosphorylation analysis of extracellular signal-regulated kinases 1 and 2 (ERK1/2) in untreated, levalbuterol- or CNO-treated HEK cells transfected with non-chimeric β2AR, DREADD-β2AR, or empty vector. Left: Western blot for pERK1/2 and GAPDH (loading control). The anti-pERK1/2 antibody results in an upper band for pERK1 (44 kDa) and a lower band for pERK2 (42 kDA). Right: Densitometry analysis of combined pERK1/2 normalized to GAPDH. Error bars: standard error of the mean. N = three experimental repetitions. Supplementary Fig. S4 shows full scan Western blot membranes from all repetitions. Source data are provided as a Source Data file.

---

(Supplementary Fig. S11b, c). This was not observed with levalbuterol stimulation of endogenous β2AR and neither with DREADD-β2AR, DREADD-GPR65, or DREADD-GPR109A upon CNO application (Supplementary Fig. S11b, c). Thus, we conclude that these GPCRs are not mediators of Ca²⁺ signaling in the microglia-like cell line HMC3.

### DREADD-based chimeras modulate microglial gene expression under inflammatory conditions

Next, we investigated immunomodulatory consequences of GPCR signaling and induced inflammation by exposing HMC3 cells to recombinant interferon γ (IFNγ) and interleukin 1β (IL1β). Both cytokines can trigger the transcription of inflammatory genes like interleukin 6 (*IL6*)[85]. Tumor necrosis factor (*TNF*) and *IL1β* expression are also part of the HMC3 cell inflammatory signature[86]. RT-qPCR confirmed that IFNγ/IL1β stimulation increased the expression of these three inflammatory genes (Supplementary Fig. S10b). When we treated non-transduced HMC3 cells with levalbuterol and compared *IL6*, *TNF*, and *IL1β* transcript abundance to the mean of untreated samples, we did not observe a response. However, when we combined levalbuterol with IFNγ/IL1β, we found significant changes compared to IFNγ/IL1β stimulation alone. Transcript levels of *IL6* increased and *TNF* decreased, while at the same time *IL1β* stayed unaltered (Fig. 8a). Strikingly, we recapitulated the same response-pattern with DREADD-β2AR-expressing cells upon CNO application

(Fig. 8b). Importantly, CNO did not impact the inflammatory response in the absence of GPCR chimeras. (Fig. 8c). We repeated this experiment with the DREADD-GPR65 cell line and found a similar effect with increased *IL6* and dampened *TNF* expression (Fig. 8d). In contrast, DREADD-GPR109A did not significantly modify inflammatory gene expression induced by IFNγ/IL1β stimulation (Fig. 8e).

Since all three Gαs-coupled receptors modulated gene expression in the same manner, and we have previously shown their ability to induce the second messenger cAMP (Figs. 2e, f, 7c), we hypothesized that elevated cAMP levels during IFNγ/IL1β exposure are responsible for the shared gene expression signature. To test this, we performed IFNγ/IL1β stimulation in the presence of forskolin, which induces cAMP synthesis in a GPCR-independent manner[78,79]. Indeed, we observed a significant increase of *IL6* and decrease of *TNF* compared to IFNγ/IL1β treatment alone (Fig. 8f). Interestingly, forskolin prevented the IFNγ/IL1β-mediated upregulation of *IL1β* mRNA levels, which we did not observe with endogenous β2AR, DREADD-β2AR, or DREADD-GPR65. These results suggest that our DREADD-based strategy provides the means to mimic GPCR signaling with high fidelity, which is not achieved solely by triggering the underlying second messenger cascade with forskolin.

To substantiate that DREADD-β2AR recapitulates the endogenous β2AR response, we performed next generation mRNA

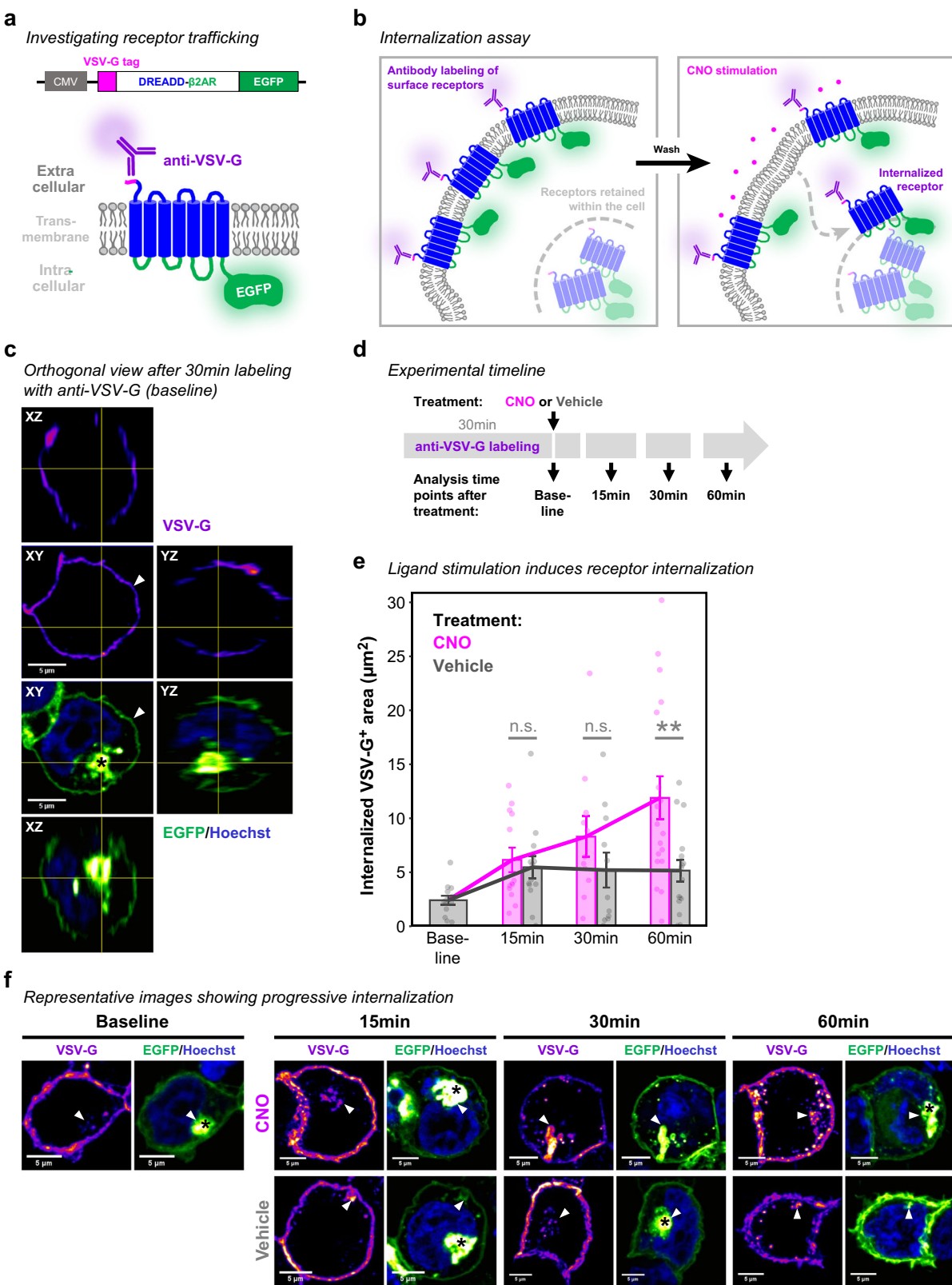

sequencing of our HMC3 cell lines with GPCR signaling under inflammatory conditions (Supplementary Fig. S12a). Principal component analysis and hierarchical clustering of sample-to-sample distances resulted in three clusters (Supplementary Fig. S12b, c, respectively): Cluster 1 summarized the biological triplicate of non-transduced HMC3 cells without exposure to inflammatory cytokines. Cluster 2 contained all cell lines that have been treated with IFNγ/IL1β

but without simultaneous GPCR stimulation. The only exception is DREADD-GPR109A. Ligand stimulation of DREADD-β2AR, DREADD-GPR65, and endogenous β2AR in non-transduced cells altered the inflammatory signature and formed Cluster 3. CNO-treated DREADD-GPR109A stayed in Cluster 2, supporting the previous RT-qPCR data (Fig. 8e) and indicating that this receptor is not involved in modulating inflammation in HMC3 cells.

**Fig. 5 | Internalization of DREADD-β2AR following CNO stimulation.**
**a** Schematic of the DREADD-β2AR-EGFP construct for internalization analysis. C-terminal EGFP visualizes receptor trafficking within the cell. Cell surface-expressed receptors are labeled with an antibody against the VSV-G epitope. CMV, human cytomegalovirus promoter. **b** Schematic exemplifying the strategy for internalized receptor detection. **c** Orthogonal view of DREADD-β2AR-EGFP-transfected HEK cell fixed immediately after 30 min of VSV-G antibody labeling. VSV-G signal visualized with an intensity-based color code (purple-red-yellow) to display signals of varying intensities. Green: EGFP visualized with intensity-based color code (green-white). Blue: nuclear staining with Hoechst. White arrow head: VSV-G/EGFP signal at the cell surface. Black asterisk: accumulation of cytoplasmic EGFP indicating receptors retained within the cell. **d** Schematic of experimental design. Following 30 min of antibody incubation, cells were fixed either

immediately (baseline) or 15, 30, and 60 min after exposure to CNO or vehicle. **e** Stimulation of DREADD-β2AR-EGFP-transfected HEK cells with CNO (magenta) or vehicle (gray). Each dot shows a single cell and its internalized VSV-G-positive area ($\mu m^2$). Error bars: standard error of the mean within each condition. Magenta and gray lines connect the mean values of CNO or vehicle exposure times, respectively. Linear regression analysis: $**p < 0.01$; $^{n.s.}p > 0.05$. Two-sided post-hoc comparisons corrected for multiple testing: $p = 0.98$ (15 min); $p = 0.45$ (30 min); $p = 0.001$(60 min). $N = 13$ (Baseline), 13 (15 min: CNO), 14 (15 min: Vehicle), 11 (30 min: CNO), 11 (30 min: Vehicle), 18 (60 min: CNO), 16 (60 min: Vehicle) cells examined over one experiment. **f** Representative maximum intensity projections of individual cells analyzed for internalized receptors confirmed by colocalizing VSV-G/EGFP signal (white arrow heads). Source data are provides as a Source Data file.

Subsequently, we performed differential expression analysis and first compared non-transduced untreated with IFNγ/IL1β-treated cells. We found 420 differentially expressed genes (Supplementary Fig. S13a, b), which we confirmed with gene ontology enrichment analysis to be associated with inflammation (Supplementary Fig. S13c). Next, we were interested to identify genes that are modulated by GPCR signaling during the inflammatory response. Levalbuterol stimulation of endogenous β2AR (Fig. 9a) resulted in 79 differentially expressed genes while DREADD-β2AR (Fig. 9b) and DREADD-GPR65 (Fig. 9c) altered 164 and 99 genes, respectively. We did not find any differentially expressed genes with DREADD-GPR109A (Fig. 9d), and CNO proved to be largely inert in the absence of DREADD-GPCRs (Fig. 9e).

To compare the signatures of ligand-stimulated endogenous β2AR, DREADD-β2AR and DREADD-GPR65, we calculated fold changes to the respective IFNγ/IL1β alone treatments for differentially expressed genes identified in Fig. 9a–c. Pearson's coefficient showed a high correlation of approximately 0.8 between endogenous β2AR, DREADD-β2AR and DREADD-GPR65 (Fig. 9f). Hierarchical clustering organized the genes in three groups (Fig. 9g, Supplementary Data 2) which we analyzed through gene ontology enrichment. Whereas gene cluster 2 and 3 indicated biologically diverse processes, gene cluster 1 pointed towards MAPK and cAMP activity (Supplementary Fig. S14), which we earlier identified as downstream targets in our HEK cell assays (Figs. 2d–i, 7c, d). Supplementary Fig. S15 highlights the response similarity of the Gα$_s$-coupled receptors across the topmost differentially expressed genes with one exception: regulator of G protein signaling 2 (*RGS2*), which was selectively upregulated upon DREADD-β2AR and DREADD-GPR65 stimulation. Since Gα$_s$ activity induces RGS2 and provides negative feedback regulation on cAMP synthesis[87], the overexpression of our DREADD-GPCRs might have triggered a stronger response compared to endogenous β2AR.

Even though DREADD-β2AR and DREADD-GPR65 had highly correlated signatures due to canonical Gα$_s$-coupling, we found unique features upon hierarchical clustering of their respective response signatures (Supplementary Fig. S16a, Supplementary Data 2). A small set of genes located in cluster A and C exhibited a distinct expression pattern that was not particularly enriched for a distinct biological pathway (Supplementary Fig. S16b). To compare this unique response with another Gα$_s$-coupled GPCR, we generated stable rM3Ds-expressing HMC3 cells (Supplementary Fig. S16c) and performed RT-qPCR on cluster A and C genes. In addition, we included the three inflammatory genes *IL6*, *TNF*, and *IL1β*, which we have quantified in our previously established HMC3 cell lines (Fig. 8). We found that the rM3Ds response was distinguished from endogenous β2AR, DREADD-β2AR, and DREADD-GPR65 based on principal component analysis (Supplementary Fig. S16d) and hierarchical clustering (Supplementary Fig. S16e). At the same time, DREADD-β2AR intermingled with endogenous β2AR and was separated from DREADD-GPR65 (Supplementary Fig. S16d, e). Notably, rM3Ds also differed in the expression pattern of *TNF* and *IL1β*. rM3Ds did not induce the robust

downregulation of *TNF* but instead increased *IL1β*, which was not affected by endogenous β2AR and our DREADD-GPCRs (Supplementary Fig. S16e). Upregulation of *IL6* was similar between rM3Ds and the other receptors, indicating that this might be a more conserved feature of Gα$_s$ signaling. Our data suggest that Gα$_s$-driven modulation of gene expression can display subtle differences depending on individual GPCRs and show that DREADD-β2AR successfully mimics endogenous β2AR more closely compared to rM3Ds.

## Discussion

Here, we illustrate the utility of a DREADD-based GPCR chimera strategy to selectively dissect the impact of GPCR activation in microglia. DREADD-based chimeras exploit the advantages of the DREADD system, which responds to CNO with high affinity and in a concentration range that minimizes potential off-target effects[39–41]. This strategy complements existing light-inducible GPCR chimera approaches[26–30] and overcomes two main caveats associated with light stimulation such as phototoxicity[32–35] and the necessity for invasive surgical procedures for deep tissue light delivery[36]. Not only can CNO be administered intraperitoneally and pass the blood brain barrier[42], but it also provides future opportunities to manipulate cells outside of light-accessible tissues such as circulating T-cells, B-cells, monocytes and granulocytes.

Even though our study focuses on immune cell function, GPCRs modulate a wide range of biological processes in other cell types as well. We generated a table with the protein sequences of putative signaling domains for all 292 GPCRs-of-interest included in our alignment, separated into ICL1-3 and C-terminus (Supplementary Data 1). These sequences can be inserted in-between the hM3Dq ligand binding domains as outlined in Supplementary Fig. S17 to generate a CNO-responsive chimera mimicking a GPCR-of-interest, which provides a framework for straightforward in-silico design. Thereby, our approach also complements the previously published DREADD chimera rM3Ds[43], which has been generated by combining the rat equivalent of hM3Dq with ICL2-3 from turkey β1AR to achieve canonical Gα$_s$-coupling. Our design strategy utilizes all signaling domains including ICL1-3 and C-terminus for high fidelity recapitulation of a GPCR-of-interest. Our library of signaling domains also offers the means to create a large variety of possible chimeras and to evaluate the contribution of different ICLs and C-termini to certain pathways.

Our engineering approach utilized multiple protein sequence alignment to identify CNO-binding DREADD domains and signaling domains of potential GPCRs-of-interest rather than protein domain identification on crystal structures; the latter are not available for most GPCRs. We used published crystal structures for RHO, CHRM3, and β2AR, and confirmed alignment accuracy by comparing our identified domains with these structural representations (Fig. 1e). To validate our strategy, we focused on β2AR, given the extensive literature sources on its function and importance for the immune system[9–11,44,45,49]. Indeed, we found that CNO stimulation of DREADD-β2AR in HEK cells successfully mimicked β2AR signaling and induced cAMP synthesis

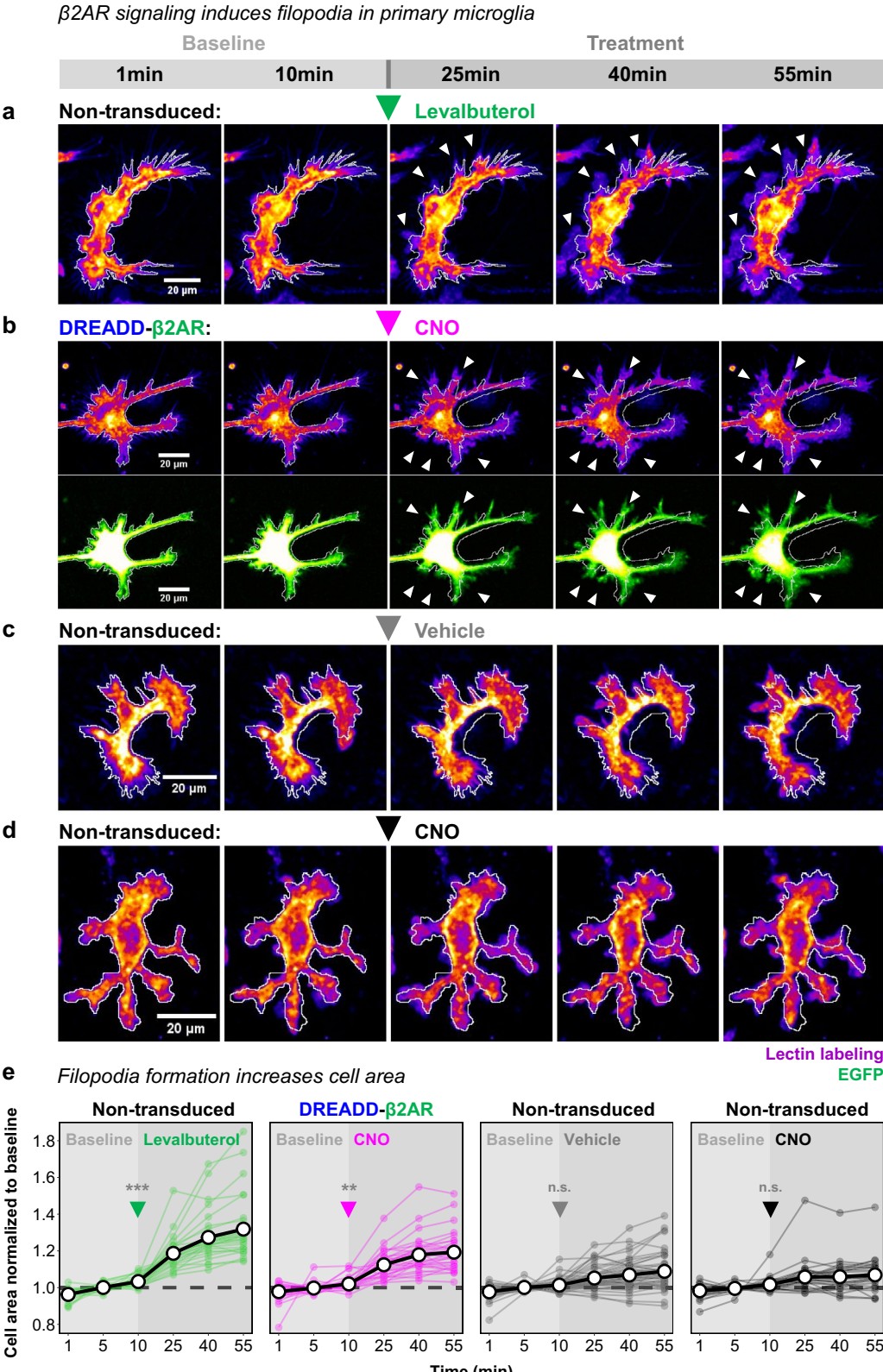

*β2AR signaling induces filopodia in primary microglia*

*Filopodia formation increases cell area*

Lectin labeling
EGFP

(Fig. 2e, f), acted on the MAPK pathway by suppressing transcription from an SRE reporter (Fig. 2i), recruited β-arrestin 2 (Fig. 4b), and phosphorylated ERK1/2 (Fig. 4c). This suggests that we properly identified ligand binding and signaling domains in DREADD and β2AR. Additionally, we compared DREADD-β2AR with rM3Ds and found that both GPCRs imitated the cAMP and MAPK activity of β2AR with high fidelity (Fig. 2e–g, i) while our construct displayed lower constitutive

activity (Fig. 3). We also found that DREADD-β2AR-expressing primary microglia responded to CNO with filopodia formation, replicating previously reported effects of endogenous β2AR activation (Fig. 6)[45]. This underlines that our chimeric approach is able to modulate microglia function in primary culture systems.

To dissect immunomodulatory properties of GPCR activation, we utilized the HMC3 microglia-like cell line and established cultures with

**Fig. 6 | Stimulation of DREADD-β2AR induces filopodia formation in primary microglia. a, d** Representative images of primary microglia during 55 min of live imaging. After a 10 min baseline, non-transduced cells were stimulated with either levalbuterol (**a**), vehicle (**c**), or CNO (**d**). Cells transduced with DREADD-β2AR-P2A-EGFP (**b**) were treated with CNO after 10 min of baseline recording. Lectin labeling visualized with an intensity-based color code (purple-red-yellow) to display signals of varying intensities. Green: EGFP visualized with intensity-based color code (green-white). White outline: cell area perimeter at 1 min projected on all other shown time points. White arrow heads: filopodia formation. **e** Quantification of cell area changes throughout 55 min of live imaging. A 10 min baseline was followed by ligand application (arrow heads for onset) of either levalbuterol, CNO, or vehicle. Graphs show the fold change of individual cells

normalized to their baseline mean at selected time points of 1, 5, 10, 25, 40, and 55 min. Thick black lines: mean of all cells (30–50 per experimental group). Thin colored lines: individual cells. Dashed lines: baseline mean. Linear regression analysis modeling cell area (μm²) of individual cells across time to compare baseline with treatment period: ***$p < 0.001$; **$p < 0.01$; n.s.$p > 0.05$. Two-sided post-hoc comparisons corrected for multiple testing: $p < 0.001$ (Non-transduced: Levalbuterol); $p = 0.004$ (DREADD-β2AR: CNO); $p = 0.58$ (Non-transduced: Vehicle); $p = 0.84$ (Non-transduced: CNO). $N = 30$ (Non-transduced: Levalbuterol), 32 (DREADD-β2AR: CNO), 50 (Non-transduced: Vehicle), 30 (Non-transduced: CNO) cells examined over three, ten, nine, and eight experiments, respectively. Source data are provided as a Source Data file.

stable DREADD-GPCR expression to reliably quantify the impact on inflammation. We challenged these cells with the inflammation mediators IFNγ and IL1β, which can induce prominent inflammatory gene expression in the HMC3 line[85] in contrast to commonly used lipopolysaccharide (LPS)[81]. Using RT-qPCR, we found that, in the presence of IFNγ/IL1β, β2AR activation with levalbuterol induced pro- and anti-inflammatory properties reflected by enhanced *IL6* and reduced *TNF* expression, respectively (Fig. 8a). We successfully mimicked this response with DREADD-β2AR upon CNO stimulation (Fig. 8b). These findings are in line with studies reporting similar pro- and anti-inflammatory effects of β2AR in different in vitro systems[44,88,89]. In our study, we also generated DREADD-based chimeras imitating the proton-sensing GPR65 and ketone-binding GPR109A. DREADD-GPR65 and DREADD-GPR109A triggered their expected cascades by either inducing (Fig. 7c, d) or inhibiting (Fig. 7e–g) Gαs signaling, respectively. As a note, the comparatively weaker DREADD-GPR109A response is in line with the technical challenges of quantifying inhibitory effects on cAMP[62].

Following CNO stimulation, DREADD-GPR65 modified inflammatory gene expression similar to β2AR (Fig. 8d), whereas DREADD-GPR109A did not alter *IL6*, *TNF*, or *IL1β* mRNA levels during IFNγ/IL1β-induced inflammation (Fig. 8e). A previous study[72] observed an anti-inflammatory effect of GPR65 in primary mouse microglia by inhibiting LPS-induced *Il1β* expression after acidification of the culture medium. Our results suggest that this effect is not present in HMC3 cells when using the cytokines IFNγ/IL1β to trigger inflammation. Another study[73] reported an anti-inflammatory role of GPR109A in the murine N9 microglia-like cell line by downregulating LPS-induced *Tnf* and *Il1β* after dimethyl fumarate treatment, an immunomodulatory drug and GPR109A agonist. Such discrepancies highlight the response diversity with different in vitro systems and inflammatory mediators[72,73].

To further support that DREADD-β2AR can replicate β2AR, we used next-generation mRNA sequencing and confirmed a strong correlation between the two responses across approximately 200 differentially expressed genes (Fig. 9f, g). This analysis also confirmed that DREADD-GPR65 modulated inflammation in a highly similar manner and gene ontology enrichment hinted that cAMP and MAPK activity are partly responsible for the shared gene expression pattern (Supplementary Fig. S14). The highly correlated signatures of β2AR, DREADD-β2AR and DREADD-GPR65 suggest that canonical Gαs-coupling modulates IFNγ/IL1β-mediated inflammation in a similar manner. Interestingly, we identified a unique transcriptional signature for a small set of genes for each DREADD-chimera. We confirmed these genes with RT-qPCR and found that this expression pattern is distinct from Gαs-coupled rM3Ds (Supplementary Fig. S16). Our data shows that DREADD-based chimeras coupled to the same canonical pathway are capable of recapitulating unique transcriptional profiles. Recently, the DREADD system has been explored for selective microglia manipulation in-vivo to study their role during neuropathic pain in mice[90]. This study exploited a transgenic mouse line to achieve microglia-specific expression of

the Gα$_i$-coupled hM4Di and to shed light on this broad signaling pathway. However, microglia might express Gα$_i$-coupled receptors with non-canonical signaling cascades that are not captured by this DREADD approach. Moreover, our HEK cell data surprisingly showed that hM4Di was capable of inducing cAMP synthesis (Fig. 2g) despite being expected to do the opposite[39], which stresses the importance of potential cell type-specific consequences. In this context, DREADD-based chimeras could offer a strategy for a more fine-tuned dissection of specific GPCRs and their role in regulating microglia function. While microglia in vitro models are critical for neuroimmunological research, it is important to note that different culture systems display distinct genetic signatures and only partially reflect the phenotype of microglia in-vivo[91]. Therefore, it would be ultimately desirable to apply DREADD-chimeras in an in-vivo context. However, in-vivo targeting of microglia is a major challenge within the field due to a current lack of efficient and specific vectors[69]. Yet, GPCR signaling is critical for many other cell types that might be more accessible for chimeric GPCR expression. Our strategy complements existing methods for GPCR investigation and offers an alternative approach to dissect GPCR signaling in various contexts and model systems.

## Methods
### Analysis of retina transcriptome data
A list of GPCRs was manually collected from Class A (rhodopsin-like, excluding olfactory receptors), Class B (secretin receptor family), Class C (metabotropic glutamate), Class D (fungal mating pheromone receptors), Class E (cAMP receptors), and Class F (Frizzled/Smoothened) and contained in total 361 GPCRs. 58 GPCRs were orphans.

Retinal transcriptome data (GSE33089) and analysis was obtained from Siegert et al.[48]. After array data normalization and removing rod contamination[48], we calculated the mean gene expression for each biological triplicate and organized this data as *Mean Data* matrix with $n$ columns and $m$ rows, where $n$ is the number of biological triplicates and $m$ the number of genes represented on the chip. We formed a selected combination (*sc*) of *Mean Data* by selecting and gluing together different columns from the *Mean Data*. *sc* has $n_{sc}$ columns and $m$ rows, where $n_{sc}$ is less than $n$. Some columns of *Mean Data* that were not selected for *sc* were glued together, column by column, and called "non-selected combination" (*n-sc*). *n-sc* has $n_{n-sc}$ columns and $m$ rows, where $n_{n-sc} + n_{sc} \leq n$. Each *sc* matrix has several corresponding *n-sc* matrices, depending on how many non-selected columns were picked. Our analysis compared columns or rows in different *sc* matrices to the columns or rows of the corresponding *n-sc* matrices. In the analysis, two matrices were treated as equivalent if they had the same set of columns but the columns were ordered differently. We refer to the columns of *Mean Data*, *sc*, and *n-sc* by the names of the cell groups. The specificity ratio (*sr*) was defined for each gene in the context of the chosen *sc* and *n-sc* matrices. For each of the corresponding row of *sc* and *n-sc*, the minimum expression value of the *sc* row was divided by the maximum expression value of the *n-sc* row. Significance (*P*) of difference in gene expression levels in the context of chosen *sc* and

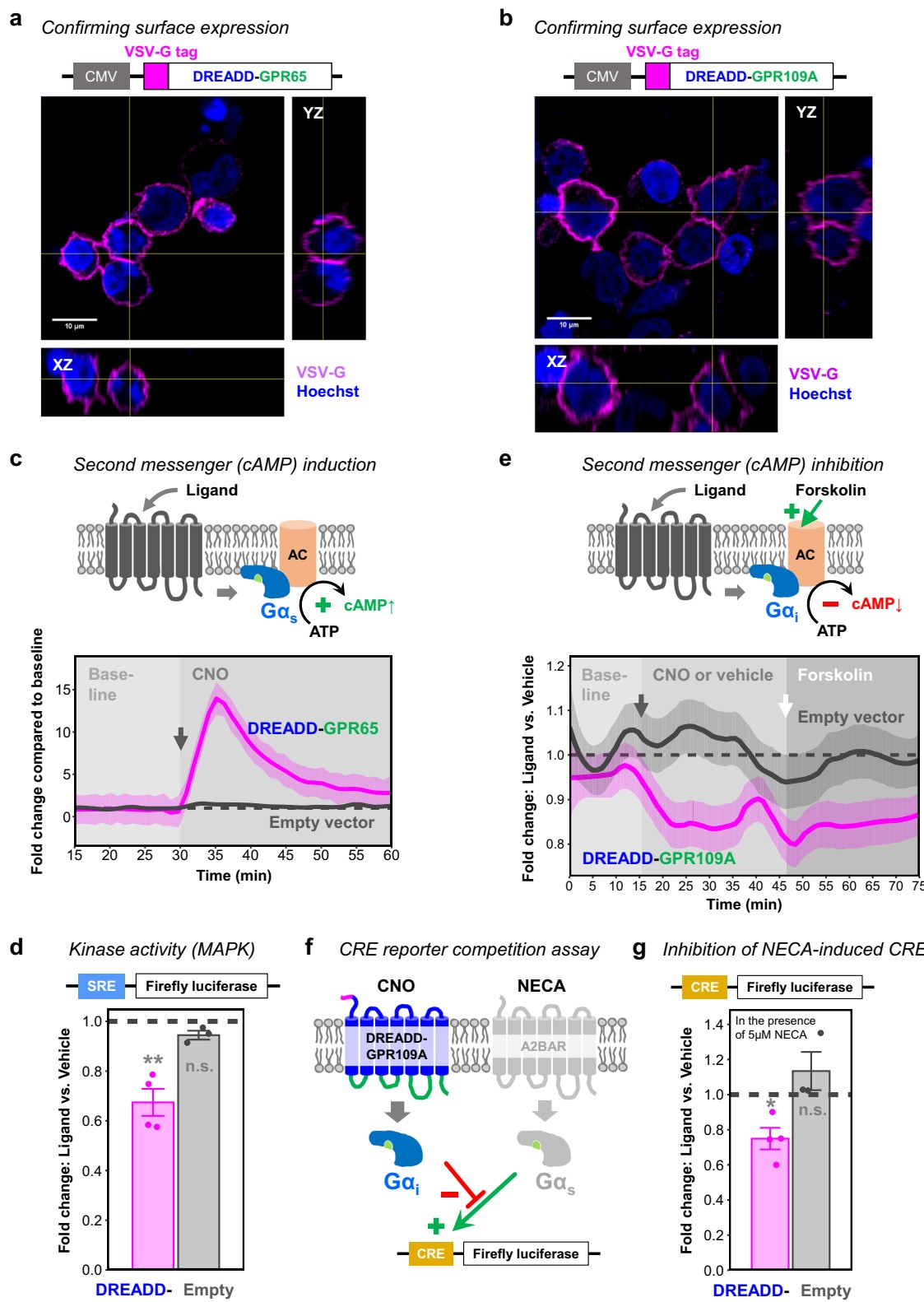

n-sc matrices was determined using the Wilcoxon rank sum test. Each of the corresponding sc rows and n-sc rows were compared using the test. A heatmaps was created to visualize differences in gene expression across cell classes and types for the 361 GPCRs. For the heat map of cell classes and types, the expression values were normalized to the maximum expression. The sr and P values were color-coded and plotted next to the heat map.

**Multiple protein sequence alignment**

The previously established domains of bovine rhodopsin (RHO)[27] served as reference for the identification of ligand binding and signaling domains. In total, 294 protein sequences were aligned, including RHO, hM3Dq, two sequences (hamβ2AR, hα1AR) from Airan et al.[27] as internal control, and GPCRs-of-interest (human and mouse class A GPCRs available at IUPHAR/BPS; www.guidetopharmacology.org).

**Fig. 7 | DREADD-GPR65 and DREADD-GPR109A respond with their expected signaling cascades. a, b** Orthogonal view of HEK cells transfected with DREADD-GPR65 (**a**) or DREADD-GPR109A (**b**) immunostained for the VSV-G tag under non-permeabilizing conditions. Magenta: VSV-G tag. Blue: nuclear staining with Hoechst. CMV, human cytomegalovirus promoter. **c** Top: Schematic of Gα$_s$-coupled GPCR inducing cAMP synthesis after ligand stimulation through adenylyl cyclase (AC) activation. Below: Real-time measurement of cAMP-dependent luciferase activity in HEK cells transfected with DREADD-GPR65 (magenta) or empty vector (gray). Baseline measurement of 30 min (first 15 min not shown) followed by CNO application (gray arrow for onset). Measure of center: Mean fold change compared to baseline mean (dashed line) in the same experimental repetition. Ribbons: 95% confidence intervals. $N$ = four (DREADD-GPR65) or seven (Empty vector) experimental repetitions. Source data are provided as a Source Data file. **d** Endpoint measurement of serum responsive element (SRE)-dependent luciferase activity in HEK cells transfected with DREADD-GPR65 (magenta) or empty vector (gray). Ligand stimulation with CNO. Dashed line: level of the respective vehicle control. Error bars: standard error of the mean. Two-sided one-sample $T$-test for comparing to a mean of 1 representing the vehicle control: *$p < 0.05$; $^{n.s.}p > 0.05$. Exact $p$-values of individual $T$-tests without multiple testing correction: $p = 0.009$ (DREADD-GPR65); $p = 0.09$ (Empty vector). $N$ = four (DREADD-GPR65) or three (Empty vector) experimental repetitions. Source data are provided as a Source Data

file. **e** Top: Schematic of Gα$_i$-coupled GPCR reducing cAMP levels after ligand stimulation through adenylyl cyclase (AC) inhibition. Forskolin induces cAMP synthesis through AC activation. Below: Real-time measurement of cAMP-dependent luciferase activity in HEK cells transfected with DREADD-GPR109A (magenta) or empty vector (gray). Baseline measurements followed by application of CNO or vehicle (gray arrow for onset) and forskolin (white arrow for onset). Measure of center: Mean fold change compared to vehicle (dashed line) in the same experimental repetition. Ribbons: 95% confidence intervals. $N$ = five experimental repetitions. Source data are provided as a Source Data file. **f** Schematic of competition assay between Gα$_i$-coupled DREADD-GPR109A and Gα$_s$-coupled A2B adenosine receptor (A2BAR). Simultaneous stimulation of Gα$_i$ through CNO and Gα$_s$ through NECA prevents cAMP-responsive element (CRE)-mediated luciferase reporter activity. **g** Endpoint measurement of CRE-dependent luciferase activity in HEK cells transfected with DREADD-GPR109A (magenta) or empty vector (gray). Simultaneous stimulation with CNO and 5 μM NECA. Dashed line: level of the respective vehicle control. Error bars: standard error of the mean. Two-sided one-sample $T$-test for comparing to a mean of 1 representing the vehicle control: *$p < 0.05$; $^{n.s.}p > 0.05$. Exact $p$-values of individual $T$-tests without multiple testing correction: $p = 0.03$ (DREADD-GPR109A); $p = 0.34$ (Empty vector). $N$ = three (DREADD-GPR109A) or four (Empty vector) experimental repetitions. Source data are provided as a Source Data file.

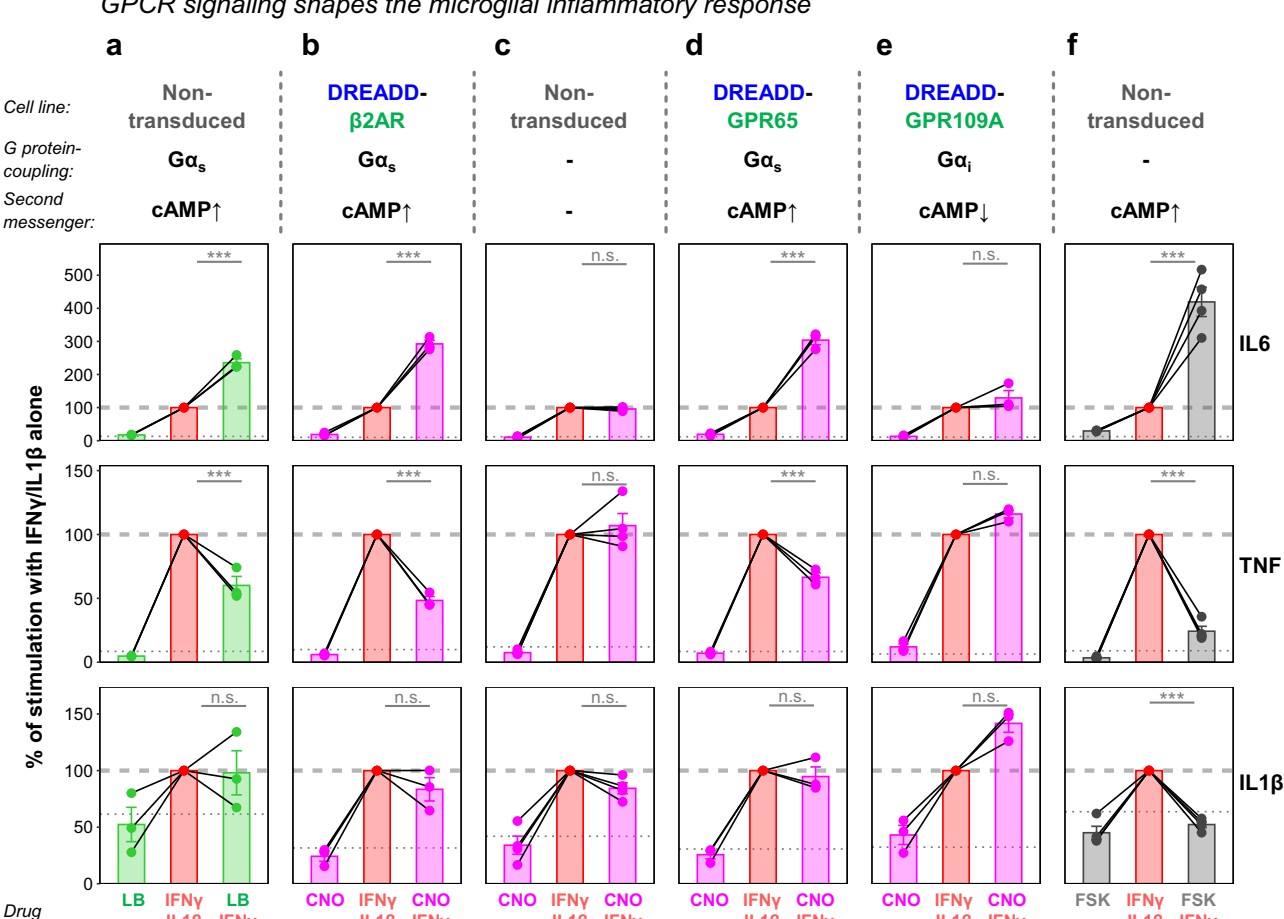

*GPCR signaling shapes the microglial inflammatory response*

**Fig. 8 | DREADD-based chimeras modulate expression of key inflammatory genes. a–f** Quantitative reverse transcription PCR (RT-qPCR) for interleukin 6 (*IL6*, top row), tumor necrosis factor (*TNF*, middle row), and interleukin 1β (*IL1β*, bottom row). Different HMC3 cell lines simultaneously treated with recombinant interferon γ (IFNγ) and interleukin 1β, and combinations of levalbuterol (LB, **a**), CNO (**b**–**e**), or forskolin (FSK, **f**). Graphs show fold changes compared to untreated cells with the IFNγ/IL1β treatment set to 100% within each repetition (dashed line). Dotted line: level of untreated controls. Lines connecting dots: dependent samples within experimental repetitions. Error bars: standard error of

the mean. Linear regression analysis with two-sided post-hoc comparisons corrected for multiple testing: ***$p < 0.001$; $^{n.s.}p > 0.05$. Exact $p$-values: $p < 0.001$ (**a** *IL6*); $p < 0.001$ (**a** *TNF*); $p = 0.96$ (**a** *IL1β*); $p < 0.001$ (**b** *IL6*); $p < 0.001$ (**b** *TNF*); $p = 0.62$ (**b** *IL1β*); $p = 0.94$ (**c** *IL6*); $p = 0.79$ (**c** *TNF*); $p = 0.60$ (**c** *IL1β*); $p < 0.001$ (**d** *IL6*); $p < 0.001$ (**d** *TNF*); $p = 0.93$ (**d** *IL1β*); $p = 0.22$ (**e** *IL6*); $p = 0.64$ (**e** *TNF*); $p = 0.18$ (**e** *IL1β*); $p < 0.001$ (**f** *IL6*); $p < 0.001$ (**f** *TNF*); $p < 0.001$ (**f** *IL1β*). See Supplementary Data 3 for exact $p$-values of all comparisons. $N$ = three (**a, b, d, e**) or four (**c, f**) experimental repetitions. Source data are provided as a Source Data file.

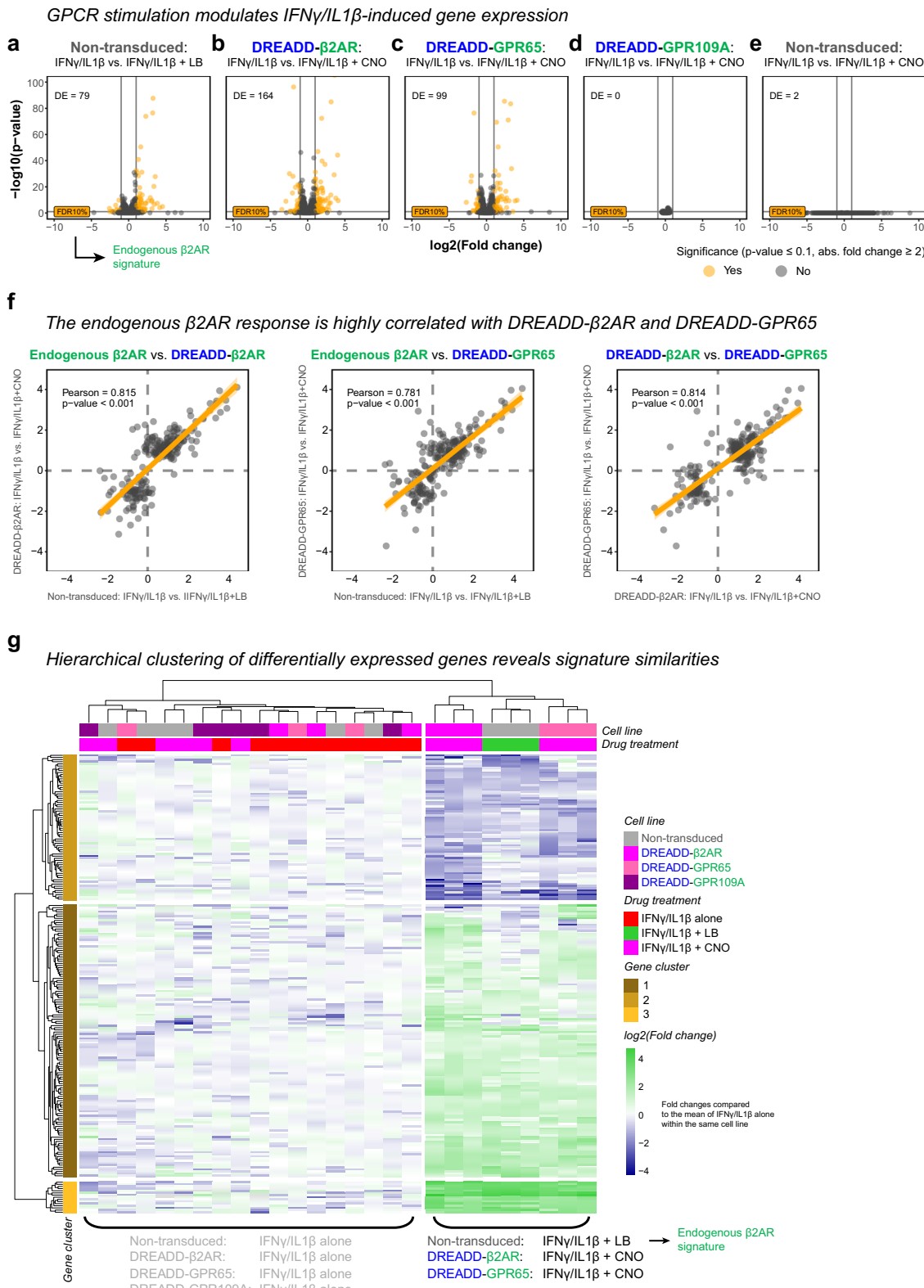

*GPCR stimulation modulates IFNγ/IL1β-induced gene expression*

**a** Non-transduced: IFNγ/IL1β vs. IFNγ/IL1β + LB
**b** DREADD-β2AR: IFNγ/IL1β vs. IFNγ/IL1β + CNO
**c** DREADD-GPR65: IFNγ/IL1β vs. IFNγ/IL1β + CNO
**d** DREADD-GPR109A: IFNγ/IL1β vs. IFNγ/IL1β + CNO
**e** Non-transduced: IFNγ/IL1β vs. IFNγ/IL1β + CNO

Significance (p-value ≤ 0.1, abs. fold change ≥ 2)   Yes   No

**f** *The endogenous β2AR response is highly correlated with DREADD-β2AR and DREADD-GPR65*

Endogenous β2AR vs. DREADD-β2AR
Endogenous β2AR vs. DREADD-GPR65
DREADD-β2AR vs. DREADD-GPR65

**g** *Hierarchical clustering of differentially expressed genes reveals signature similarities*

| Non-transduced: | IFNγ/IL1β alone |
| DREADD-β2AR: | IFNγ/IL1β alone |
| DREADD-GPR65: | IFNγ/IL1β alone |
| DREADD-GPR109A: | IFNγ/IL1β alone |
| DREADD-GPR109A: | IFNγ/IL1β + CNO |

| Non-transduced: | IFNγ/IL1β + LB |
| DREADD-β2AR: | IFNγ/IL1β + CNO |
| DREADD-GPR65: | IFNγ/IL1β + CNO |

Sequences were combined in a FASTA file, which served as input for the alignment algorithm MUSCLE[92]. To visualize results, the alignment output was imported into the software Jalview 2.9.0b2. Sequences were identified as ligand binding or signaling domains based on their alignment with the RHO reference. Signaling domains were labeled according to their location as intracellular loops (ICL) 1–3 and C-terminus (C-Term).

**Predicting transmembrane GPCR domains**
The bioinformatics tool TMHMM (www.cbs.dtu.dk/services/TMHMM)[50] was used to predict transmembrane helices (TMs) for selected GPCRs (RHO, hM3Dq, hamβ2AR, hα1AR, β2AR, GPR65, GPR109A, and GPR183). We highlighted predicted TMs in our alignment output with Jalview 2.9.0b2 (Fig. 1d, Supplementary Fig. S1).

**Fig. 9 | DREADD-based chimeras shape the inflammatory response and recapitulate the β2AR signature with high similarity. a–e** Vulcano plots of next generation mRNA sequencing data from different HMC3 cell lines simultaneously treated with IFNγ/IL1β, and either levalbuterol (LB) or CNO. Graphs show individual genes (points), their log2 fold change and *p*-value in comparison to IFNγ/IL1β alone in the same cell line. DE: number of differentially expressed genes (orange data points) defined by *p* < 0.1 and absolute linear fold change > 2. Horizontal lines: false discovery rate (FDR) cutoff of 10% (*p* < 0.1). Vertical lines: linear fold change cutoff for downregulation (<−2) and upregulation (>2), respectively. Source data are provided as a Source Data file. **f** Pearson correlation of GPCR signatures across all differentially expressed genes (dots) shown in **a–c** based on their log2 fold change compared to IFNγ/IL1β alone in the same cell line. Orange line and ribbon: fitted linear model with 95% confidence intervals. Pearson correlation coefficients: 0.815 (Endogenous β2AR vs. DREADD-β2AR; *p* < 0.001); 0.781 (Endogenous β2AR vs. DREADD-GPR65; *p* < 0.001); 0.814 (DREADD-β2AR vs. DREADD-GPR65; *p* < 0.001). Source data are provided as a Source Data file. **g** Hierarchical clustering of samples (columns) and all differentially expressed genes (rows) shown in **a–c** based on log2 fold changes. Upregulation (green) and downregulation (blue) compared to IFNγ/IL1β alone in the same cell line. Source data are provided as a Source Data file.

## Identifying GPCR domains on available crystal structures

We accessed the PDB database (www.rcsb.org) to download structural representations of bovine RHO, rat CHRM3 as surrogate for hM3Dq, and human β2AR (PDB IDs: 1U19 (www.rcsb.org/structure/1u19), 4U15 (www.rcsb.org/structure/4U15) and 2RH1 (www.rcsb.org/structure/2RH1), respectively). Structural data were imported into the software VMD 1.9.2 and oriented with the intracellular domains facing towards the screen. We then highlighted alignment-identified ICL1–3, C-Term, and TMHMM-predicted TMs to see whether they map to their expected locations. In the case of partly missing structural data, we used dotted lines as representation.

## Adding N-terminal modifications to GPCR chimeras

The bioinformatics tool SignalP (www.cbs.dtu.dk/services/SignalP)[54] was used to predict whether hM3Dq and β2AR contain a signal peptide. Since such a sequence was not found, we added a hemagglutinin-derived signal peptide (KTIIALSYIFCLVFA) at the N-terminus[51,52]. Additionally, we also included a VSV-G epitope (YTDIEMNRLGK) followed by a DSL linker immediately after the signal peptide[53] (Supplementary Fig. S2a, b). In the corresponding DNA sequences, the start codon (ATG) was removed to prevent leaky scanning and to ensure that all proteins contain the VSV-G tag.

## Obtaining DNA sequences for GPCR chimeras via gene synthesis

To generate a chimera for a GPCR-of-interest, we combined ligand-binding hM3Dq domains and GPCR-of-interest signaling domains in silico (Supplementary Fig. S16, Supplementary Data 1). We identified the corresponding DNA sequences of these domains through the NCBI Consensus Coding Sequence (CCDS) database[93] and added our N-terminal modifications. The entire coding sequence was then synthesized (www.eurofinsgenomics.eu) in the pEX-K4 or pEX-A2 vector. During synthesis, recognition sites for restriction enzymes (EcoRI or NotI, BamHI) were added up- and downstream of the chimera. The same strategy was used to obtain N-terminally modified non-chimeric β2AR, hM3Dq, rM3Ds and hM4Di.

## Cloning

For HEK cell assays, if not otherwise stated, GPCRs were excised from pEX-K4 or pEX-A2 and inserted into the mammalian expression vector pcDNA3.1(−) using EcoRI or NotI, and BamHI.

To study protein-protein interactions, we utilized the NanoBiT system (Promega; N2014). GPCRs were amplified from pEX-K4 with primers carrying restriction sites for NheI and EcoRI. These restriction sites were then used to clone GPCRs into pBiT2.1-C[TK-SmBiT] in order to obtain GPCR-SmBiT fusion constructs. β-arrestin 2 was amplifying from pCDNA3.1(+)-CMV-bArrestin2-TEV (Addgene #107245) with Gibson Assembly primers compatible with the NEBuilder HiFi DNA Assembly Kit (New England BioLabs; E2621). β-arrestin 2 was subsequently assembled into pBiT1.1-C[TK-LgBiT], linearized by NheI and XhoI, in order to obtain β-arrestin 2-LgBiT.

To generate DREADD-β2AR-EGFP, we amplified DREADD-β2AR from pEX-K4, and EGFP from PL-SIN-PGK-EGFP (Addgene #21316) with Gibson Assembly primers. Both fragments were then assembled into pcDNA3.1(−), linearized by NotI and BamHI.

Bicistronic constructs encoding for GPCR-P2A-EGFP were obtained through a cloning step involving an intermediate vector, encoding for mCherry-P2A-EGFP, which was previously generated in the laboratory. First, GPCRs were amplified from pEX-K4 with Gibson Assembly primers and assembled into the intermediate vector, linearized by NheI and BamHI in order to excise mCherry and replace it with GPCRs. Finally, GPCR-P2A-EGFP was amplified from these vector intermediates with Gibson Assembly primers and assembled into pcDNA3.1(−), linearized by NotI and BamHI.

For lentivirus production, we used a modified transfer vector based on PL-SIN-PGK-EGFP. This plasmid was modified through Gibson Assembly by introducing a WPRE sequence downstream of EGFP followed by a microRNA9 sponge (miR9T), which was previously described for optimized microglia transduction[94,95]. WPRE was amplified from pAAV-hSyn-tdTomato (a gift from the Jonas group at ISTA). The miR9T sequence was synthesized (www.eurofinsgenomics.eu) in the pEX-A258 vector and subsequently amplified. Both fragments were then assembled into PCR-linearized PL-SIN-PGK-EGFP, which generated PL-SIN-PGK-EGFP-WPRE-miR9T. Finally, GPCR-P2A-EGFP was amplified with Gibson Assembly primers from the previously established pcDNA3.1(−) vectors and assembled into PL-SIN-PGK-EGFP-WPRE-miR9T, linearized by PstI and BsrGI.

## Cell lines

HEK293T cells were obtained from ATCC (CRL-3216) and cultured in HEK-complete medium, containing DMEM (ThermoFisher; 31966; with high glucose content, GlutaMAX and pyruvate), 10% (v/v) fetal bovine serum (FBS; Sigma; 12103C; heat-inactivated for 30 min at 56 °C), 1% (v/v) non-essential amino acids (Sigma; M7145) and 1% (v/v) penicillin-streptomycin (ThermoFisher; 15140-122). Medium was sterile filtered (0.22 μm; TPP; 99505) and stored at 4 °C.

HMC3 cells were obtained from ATCC (CRL-3304) and cultured in EMEM-complete medium, containing EMEM (ATCC, 30-2003), 10% (v/v) FBS and 1% (v/v) penicillin-streptomycin. Medium was sterile filtered (0.22 μm) and stored at 4 °C.

HEK cells are registered as commonly misidentified cells lines (https://iclac.org/databases/cross-contaminations) but are frequently used for studying GPCRs[59] due to their favorable transfection and culture conditions. Certified cell lines were purchased from ATCC and we continuously ensured that they matched the morphological description provided by the manufacturer. Human origin of both cell lines has been confirmed by successful PCR amplification of human gene sequences.

## Cell maintenance

HEK cells were maintained in T75 flasks with 15 ml medium. Culture conditions were 37 °C and 5% CO₂. In order to passage cells, old medium was aspirated and cell layer was washed with 10 ml DPBS (37 °C). PBS was aspirated and 3 ml Trypsin-EDTA (ThermoFisher; 25300-054; 37 °C) were added for approximately 1 min until the cell layer detached. Trypsinization was stopped with 10 ml medium (37 °C). Cells were pelleted at Trypsin-EDTA (ThermoFisher; 25300-054; 37 °C). Supernatant was aspirated and pelleted cells were resuspended thoroughly in 10 ml medium (37 °C). Cells were counted and 0.5–0.75

million cells were transferred to a new culture flask within a final volume of 15 ml medium (37 °C). Cells were passaged every 3–4 days when they reached approximately 80% confluency.

HMC3 cells were maintained in 10 cm dishes (Sigma, CLS430167) with 10 ml medium. Culture conditions were 37 °C and 5% $CO_2$. For passaging, old medium was aspirated and cell layer was washed with 10 ml DPBS (37 °C). PBS was aspirated and 3 ml Trypsin-EDTA were added for approximately 5–15 min until the cell layer detached. Trypsinization was stopped with 10 ml medium (37 °C). Cells were counted from this suspension and 0.25–0.5 million cells were transferred to a new culture dish within a final volume of 10 ml medium (37 °C). Cells were passaged every 3–4 days when they reached approximately 80% confluency.

### Coating plates for HEK cell assays and immunostaining

White clear-bottom 96-well plates (Greiner Bio-One; 655098) and 8-well chamber slides (ibidi; 80826; growth area: 1cm²) were coated with 50 μl and 100 μl poly-L-ornithine (ready-to-use 0.1% (w/v) solution; Sigma; P4957) respectively. After 1 h incubation at room temperature, wells were washed three times with 100 μl sterile Milli-Q water and left to dry with an open lid for 1 h under UV irradiation in a sterile laminar flow hood. Culture dishes were then wrapped with Parafilm and stored at 4 °C.

### HEK cell transfection

Cells were transfected by seeding them into wells of indicated culture vessels containing transfection mix. To avoid toxicity of antibiotics during transfection, cell suspensions were prepared in HEK-complete medium without penicillin-streptomycin. Polyethylenimine (PEI; Polysciences; 24765) was used as transfection reagent. A stock solution (1 mg/ml) was prepared by dissolving PEI in Milli-Q water and adjusting the pH to 7. Aliquots were stored at −20 °C. To make the transfection mix, plasmids were first diluted in Optimem (ThermoFisher; 51985034) to a total concentration of 40 ng/μl. In parallel, PEI stock was diluted 1:10 in Optimem and incubated for 5 min at room temperature. Plasmid and diluted PEI stock were then mixed 1:1 to generate the transfection mix (containing 2.5 μl PEI stock per μg DNA). After 20 min incubation at room temperature, this transfection mix was pipetted into wells followed by adding the desired number of cells (Supplementary Table S1). Assays were performed 24 h after transfection.

### Confocal microscopy

Images of immunostainings were acquired on inverted Zeiss LSM800 or Zeiss LSM880 microscopes with either a 63x oil immersion or 20x air objective. Live imaging of primary microglia was performed on an inverted Zeiss LSM800 using a 20x air objective.

### Preparation of Antifade mounting medium

Mowiol 4–88 (2.4 g; Sigma; 81381) and glycerol (4.8 ml; Sigma; G7757) were combined with 6 ml Milli-Q water and 12 ml Tris buffer (0.2 M; pH 8) and stirred overnight at room temperature. After letting the solution rest for 2 h, it was incubated for 10 min at 50 °C in a water bath and then centrifuged at 4700 × g for 15 min. The supernatant was combined with DABCO (Sigma; D27802) at 2.5% (w/v). Aliquots were stored at −20 °C.

### VSV-G immunostaining for confirming cell surface expression of GPCRs

HEK cells were transfected with GPCR (600 ng) in coated 8-well chamber slides (ibidi; 80826) as described under *HEK cell transfection*. After 24 h, live cells were immunostained under non-permeabilizing conditions. For this, mouse monoclonal anti-VSV-G antibody conjugated to Cy3 (Sigma; C7706; clone P5D4; LOT: 049M4837V; RRID:AB_259043) was first subjected to ultrafiltration to reduce the concentration of cytotoxic $NaN_3$. The required amount of antibody for

a final 1:250 dilution was added to PBS (5 ml) and applied to a Vivaspin 6 concentrator (Sartorius; VS0601; MWCO: 10 kDa; 5 ml volume). After centrifugation (4000 × g; 10 min; 4 °C), the concentrate was added to the required amount of cold cell culture medium for obtaining the final 1:250 dilution. To start the immunostaining, medium was aspirated from the chamber slides and replaced with cold antibody-containing medium. Cells were incubated on ice for 1 h. Wells were then washed three times with cold medium without antibody for 3 min each, followed by fixation with cold 4% (w/v) PFA in PBS for 15 min. After fixing, cells were washed once with PBS for 3 min and subjected to nuclear staining with Hoechst (New England BioLabs; 4082S; diluted 1:5000 in PBS) for 5 min. Wells were briefly washed with PBS before adding 200 μl Antifade mounting medium. Chamber slides were stored at 4 °C until imaging. All incubation steps were carried out with 200 μl of the respective solution and under protection from light to avoid bleaching of the Cy3 fluorophore.

Confocal images were acquired with a ×63 oil immersion objective. Images were processed in Fiji 1.51 f by applying a gamma correction of 0.5 for better visualization of faint VSV-G signals, followed by a rolling ball background subtraction and a 2 × 2 × 2 median filter.

For VSV-G staining of HMC3 cells, uncoated 8-well chamber slides were used and 3500 cells were seeded per well together with GPCR-encoding lentivirus at a multiplicity of infection (MOI) of 5. The above described staining procedure was performed after three to four days when cells reached approximately 80% confluency. Confocal images were acquired with a ×20 air objective. Images were processed with 0.5 gamma correction and 2 × 2 × 2 median filtering.

### Real-time measurement of cAMP levels

All real-time luciferase assays were performed in $CO_2$-independent medium (Leibovitz's L15 (ThermoFisher; 21083027; no phenol red), 10% (v/v) FBS and 1% (v/v) penicillin-streptomycin. The medium was sterile filtered (0.22 μm) and stored at 4 °C.

To measure $G\alpha_s$-induced increases in cAMP upon ligand stimulation, a cAMP-dependent firefly luciferase (GloSensor)[56] was used (the plasmid was a gift from the Janovjak group formerly at ISTA). As luciferase substrate, a 100 mM stock solution of beetle luciferin (Promega; E1602) was prepared in 10 mM HEPES and stored at −20 °C protected from light. HEK cells were transfected in a 96-well plate with GPCR or empty vector backbone (100 ng), and GloSensor (100 ng). After 24 h, medium was replaced with 90 μl $CO_2$-independent medium (37 °C) containing 2.22 mM beetle luciferin (1:45 dilution of stock). The plate was then incubated for 15 min at 37 °C in an incubator with atmospheric $CO_2$ and then transferred to a plate reader (BioTek; Synergy H1) with the lid removed. Total bioluminescence was measured in each well every 1–2.5 min (37 °C, 1 s integration time; 200 gain) for 30 min to establish a baseline. After the last baseline measurement, the plate was ejected and a multichannel pipette was used to quickly apply 10 μl levalbuterol or CNO (prepared in $CO_2$-independent medium; 10 times more concentrated than the desired concentration of either 0.01 μM, 0.1 μM, 1 μM, or 10 μM). The measurement was immediately continued for 1 h. Individual experiments were always carried out in triplicates for each condition. For each well, a fold change in bioluminescence was calculated by normalizing luminescence to the mean of the baseline measurement. For the final analysis, baseline-normalized values were pooled from individual experimental repetitions.

To evaluate constitutive GPCR activity and its effect on baseline cAMP levels, we used a different cAMP-dependent firefly luciferase suitable baseline comparisons (GloSensor-22F; Promega; E2301)[62]. As luciferase substrate, a 100X stock solution of cAMP reagent (Promega; E1290) was prepared in 10 mM HEPES and stored at −80 °C. HEK cells were transfected with GPCR or empty vector (100 ng), and GloSensor-22F (100 ng). After 24 h, medium was changed to 100 μl $CO_2$-independent medium (37 °C) containing 2% (v/v) cAMP reagent (1:50 dilution of stock). The plate was incubated for 2 h at room temperature

and atmospheric $CO_2$ before starting a 30 min measurement in the plate reader (25 °C, 1 s integration time; 200 gain). For each experiment, fold changes were calculated by normalizing GPCR-transfected conditions to the empty vector control. To do this, luminescence values at each time point were divided by the mean of the respective empty vector triplicate. For the final analysis, empty vector-normalized values were pooled from different experimental repetitions. To measure $G\alpha_i$-induced decreases in cAMP, we utilized a cAMP-dependent firefly luciferase suitable for $G\alpha_i$ signaling (GloSensor-22F; Promega; E2301)[62]. As luciferase substrate, a 100X stock solution of cAMP reagent (Promega; E1290) was prepared in 10 mM HEPES and stored at −80 °C. HEK cells were transfected with GPCR or empty vector (100 ng), and GloSensor-22F (100 ng). After 24 h, medium was changed to 80 µl $CO_2$-independent medium (37 °C) containing 2% (v/v) cAMP reagent (1:50 dilution of stock). The plate was incubated for 2 h at room temperature and atmospheric $CO_2$ before starting the measurement in the plate reader (25 °C, 1 s integration time; 200 gain). After a 15 min baseline, each well received 10 µl CNO (prepared in $CO_2$-independent medium; 10 times more concentrated than the desired concentration of 10 µM), or an equal amount of vehicle (medium only). The measurement was immediately continued for 30 min. Then, 10 µl forskolin (prepared in $CO_2$-independent medium; 100 µM) was added to each well for a final concentration of 10 µM. The measurement immediately continued for another 30 min. Vehicle controls always received the same transfection mix as their corresponding treated condition. Individual experiment were always carried out in triplicates for each condition. For each experiment, fold changes were calculated by normalizing ligand-treated conditions to the respective vehicle controls. To do this, luminescence values at each time point were divided by the mean of the respective vehicle control triplicate. For final analysis, vehicle control-normalized values were pooled from different experimental repetitions.

### Real-time measurement of β-arrestin 2 recruitment

As luciferase substrate, the Nano-Glo Live Cell Substrate (Promega; N2011) was used. HEK cells were transfected in a 96-well plate with GPCR-SmBiT (100 ng) and β-arrestin 2-LgBiT (100 ng). After 24 h, medium was replaced with 90 µl $CO_2$-independent medium (37 °C) containing 1% (v/v) Nano-Glo Live Cell Substrate (1:100 total dilution; added from a freshly prepared 1:20 pre-dilution in LCS Dilution Buffer supplied with the reagent). The plate was then incubated for 10 min at room temperature and subsequently transferred to a plate reader (BioTek; Synergy H1) with the lid removed. Total bioluminescence was measured in each well every 40 s (room temperature, 1 s integration time; 200 gain) for 15 min to establish a baseline. After the last baseline measurement, the plate was ejected and a multichannel pipette was used to quickly apply 10 µl levalbuterol or CNO (prepared in $CO_2$-independent medium; 10 times more concentrated than the desired concentration of 10 µM), or an equal amount of vehicle (medium only). The measurement immediately continued for 45 min. Vehicle controls always received the same transfection mix as their corresponding treated condition. Individual experiments were always carried out in triplicate for each condition. For each well, a fold change in bioluminescence was calculated by normalizing luminescence to the mean of the baseline measurement. These values were further used to obtain fold changes by normalizing ligand-treated conditions to the respective vehicle controls. To do this, baseline-normalized values at each time point were divided by the mean of the respective vehicle control triplicate. For final analysis, these values were pooled from different experimental repetitions.

### SRE reporter assay

Transcription-based luciferase reporter assays were performed with the Dual-Glo kit (Promega; E2920). HEK cells were transfected in a 96-well plate with GPCR or empty vector (95 ng), SRE-dependent firefly luciferase (95 ng; Promega; E1340)[60], and ubiquitously expressed renilla luciferase (9.5 ng) to normalize for inter-assay variability (renilla luciferase was inserted into pcDNA3.1(−) and provided as a gift from the Janovjak group formerly at ISTA). After 24 h, HEK-complete medium was replaced with 90 µl fresh HEK-complete medium. Each well was then treated with 10 µl levalbuterol or CNO (prepared in HEK-complete medium; 10 times more concentrated than the desired concentration of 10 µM), or an equal amount of vehicle (medium only). After 6 h incubation at 37 °C and 5% $CO_2$, 50 µl of medium were removed from each well and replaced by 50 µl Dual-Glo luciferase reagent. Following a 10 min incubation at room temperature, the plate was transferred to a plate reader to measure firefly luminescence (BioTek; Synergy H1; room temperature, 1 s integration time; 200 gain). Then, 50 µl Stop&Glo reagent was added to each well, followed by another 10 min incubation at room temperature, renilla luminescence was measured with the same parameters. For each well, a ratio was obtained by normalizing firefly luminescence to renilla luminescence as a control for transfection efficiency, cell number and enzyme activity. Vehicle controls always received the same transfection mix as their corresponding treated condition. Individual experiments were always carried out in triplicates for each condition. For each experiment, fold changes between ligand-treated conditions and the respective vehicle controls were obtained by dividing firefly-renilla ratios by the mean of the respective vehicle control triplicate. These values were pooled from different experimental repetitions for final analysis.

To evaluate constitutive GPCR activity and its effect on the SRE reporter, we compared GPCR-transfected conditions with empty vector controls. HEK cells were transfected as described under *HEK cell transfection* and after 24 h HEK-complete medium was replaced with 100 µl fresh HEK-complete medium. After 6 h incubation at 37 °C and 5% $CO_2$, luminescence was measured with the Dual-Glo kit (Promega; E2920) as stated above. For each well, a ratio was obtained by normalizing firefly luminescence to renilla luminescence as a control for transfection efficiency, cell number and enzyme activity. Individual experiments were always carried out in triplicates for each condition. For each experiment, fold changes between GPCR-transfected conditions and empty vector control were obtained by dividing firefly-renilla ratios by the mean of the empty vector triplicate. These values were pooled from different experimental repetitions for final analysis.

### CRE reporter assay

HEK cells were transfected as described for the *SRE reporter assay* with the exception of substituting the SRE reporter with CRE-dependent firefly luciferase[60] (Promega; E8471).

To quantify inhibition of $G\alpha_s$ activity in the competition assay, HEK-complete medium was replaced after 24 h with 80 µl fresh HEK-complete medium. Each well was then treated with 10 µl CNO (prepared in HEK-complete medium; 10 times more concentrated than the desired concentration of 10 µM), or an equal amount of vehicle (medium only). Immediately afterwards, 10 µl 5′-N-ethylcarboxamidoadenosine (NECA; prepared in HEK-complete medium; 50 µM) was added to all wells for a final concentration of 5 µM. After 6 h incubation at 37 °C and 5% $CO_2$, luminescence was detected and data were analyzed by normalizing to the respective vehicle controls as described for the *SRE reporter assay*.

To evaluate the effect of ligand-stimulated GPCR in the absence of NECA, medium was replaced 24 h after transfection with 90 µl fresh HEK-complete medium and each well was treated with 10 µl CNO or an equal amount of vehicle (medium only).

### Western blot to quantify ERK1/2 phosphorylation

HEK cells were transfected in 6-well plates with GPCR or empty vector (6 µg). After 24 h, cells were serum starved for 4 h by replacing medium with 1.9 ml HEK-complete medium without FBS. Then, cells were

treated with 100 µl levalbuterol or CNO (prepared in HEK-complete medium without FBS; 20 times more concentrated than the desired concentration of 10 µM). Control conditions were left untreated. Treated cells were harvested for protein isolation 2, 5 or 15 min after addition of ligand (cells were kept at 37 °C and 5% CO₂ during ligand exposure). Untreated cells were harvested immediately. Cells were harvested by placing the plate on ice, aspirating medium and adding 200 µl ice cold and freshly prepared lysis buffer (50 mM Tris pH 7.4, 300 mM NaCl, 1 mM EDTA, 1 mM Na3VO4, 1 mM NaF, 10% (v/v) Glycerol, 1% (v/v) IGEPAL CA-630, 1% (v/v) Protease inhibitor mix set 1 (Calbiochem; 539131); 1 Phosstop tablet (Sigma; 4906845001) per 10 ml) to each well. Cells were detached with a cell scraper, transferred to 1.5 ml microcentrifuge tubes and sonicated (15 s; room temperature; inside a water bath). Samples were then centrifuged (14,000 × $g$, 20 min, 4 °C) and supernatants were transferred to fresh tubes. A small volume of each sample was used to immediately measure protein concentration with the Pierce BCA Protein Assay kit (ThermoFisher; 23227). The rest was combined with 6X loading dye (375 mM Tris pH 6.8, 9% (w/v) SDS, 30% (w/v) glycerol, 0.06% (w/v) Bromophenol blue, 600 mM DTT), cooked for 5 min at 95 °C and stored at −20 °C for subsequent Western blot analysis. Three individual experiments were performed with one well per condition in each. SDS-PAGE was performed by loading 10 µg protein (approximately 10 µl) on 8% acrylamide gels with running buffer containing 25 mM Tris, 192 mM glycine, and 0.1% (w/v) SDS. Electrophoresis was started at 90 V constant (two gels per chamber) until samples transitioned from stacking to running gel. Electrophoresis continued at 110 V constant until the 25 kDa band of the marker left the gel (approximately 2 h). Proteins were transferred to PVDF membranes (Sigma; IPFL00005) via tank blotting (300 mA constant; 2 h; 4 °C; additional cooling insert) with transfer buffer containing 25 mM Tris, 192 mM glycine, and 20% (v/v) methanol. Successful transfer was briefly checked with Ponceau staining (0.1% (w/v) Ponceau S, 5% (v/v) acetic acid). Membranes were cut to include proteins ranging from 32–80 kDa and blocked with 5% (w/v) BSA in TBST (20 mM Tris, 150 mM NaCl, 0.1% (v/v) Tween 20) for 1 h at room temperature. Membranes were then incubated overnight at 4 °C with rabbit anti-phosphorylated ERK1/2 antibody (Cell Signaling Technology; 9101S; LOT: 30; RRID:AB_331646; 1:1,000) in TBST containing 5% (w/v) BSA. Next day, membranes were washed three times with TBST for 10 min each and exposed to donkey anti-rabbit secondary antibody conjugated to horse radish peroxidase (GE Healthcare; NA934V; LOT: 16976257; 1:10,000) in TBST containing 5% (w/v) BSA for 2 h at room temperature. Membranes were again washed three times with TBST followed by signal detection with either SuperSignal West Pico PLUS (Thermo Fisher; 34579) or SuperSignal West Femto (Thermo Fisher; 34094) and imaging (Amersham 600; GE Healthcare). Membranes were then stripped (pH 2.2, 0.2 M glycine) for 30 min at room temperature, washed three times and blocked again followed by incubation with rabbit anti-GAPDH antibody (Sigma; ABS16; LOT: 3275069; 1:1,000; overnight; 4 °C). Membranes were washed three times and subjected to secondary antibody using our standard procedure. The membranes were again washed and GAPDH signal was detected and imaged. Densitometry of bands (pERK1, pERK2, GADPH) was performed with Bio-Rad Image Lab 6.0.1. Densities of pERK1 and pERK2 were then summed to generate a single value (pERK1/2). To normalize for protein loading variability, each pERK1/2 value was divided by the respective GAPDH band on the same membrane after striping. Full scan blots of all experimental repetitions are provided in Supplementary Fig. S4.

## GPCR internalization assay

HEK cells were transfected with DREADD-β2AR-EGFP (600 ng) in coated 8-well chamber slides (ibidi; 80826) as described under *HEK cell transfection*. After 24 h, live cells were subjected to anti-VSV-G antibody conjugated to Cy3 (diluted 1:250) to label VSV-G-tagged GPCRs expressed on the cell surface. Antibody labeling took place in HEK-complete medium at 37 °C and 5% CO₂ for 30 min. Wells were then briefly washed with HEK-complete medium (37 °C) and 180 µl fresh HEK-complete medium was added. Each well received 20 µl of CNO (prepared in HEK-complete medium; 10 times more concentrated than the desired concentration of 10 µM), or an equal amount of vehicle (medium only). Cells were then incubated for different time periods (0, 15, 30, or 60 min) before fixation. The 0 min time point was only treated with vehicle and immediately fixed and serves as a baseline. Vehicle controls were included for each time point to control for the potential contribution of antibody labeling to GPCR internalization.

Fixation was carried out with cold 4% (w/v) PFA in PBS for 15 min at room temperature. After fixing, cells were washed once with PBS for 3 min and subjected to nuclear staining with Hoechst (New England BioLabs; 4082 S; diluted 1:5000 in PBS) for 5 min. Wells were briefly washed with PBS before adding Antifade mounting medium. Chamber slides were stored at 4 °C until imaging. All incubation steps were carried out with 200 µl of the respective solution under protection from light to avoid bleaching of Cy3-conjugated antibody and EGFP. Confocal imaging was performed with a 63x oil immersion objective. For each condition, several images each containing one to five cells were taken at random positions within the wells. Approximately 15 cells were acquired per condition with optimal resolution in x, y, and z (71 × 71 × 230 nm). Entire images were processed in Fiji 1.51.f by rolling ball background subtraction followed by a 2 × 2 × 2 median filter. Regions of interest (ROI) were then generated by cropping individual cells and saving them as new images. For further analysis of individual cells, maximum intensity projections of six consecutive z-slices around the center of each cell were obtained. VSV-G and EGFP signals were used to trace the perimeter along the cell surface which separates intra- and extracellular space. A threshold was applied on the VSV-G channel to separate signal from background. The threshold VSV-G area within the cell surface perimeter was then measured (µm²) to quantify internalized GPCRs. To check if this signal is derived from internalized receptors, we confirmed colocalization of VSV-G and EGFP in both channels.

## Lentiviral vectors

VSV-G enveloped lentiviruses were generated by the Molecular Biology Facility at ISTA. Briefly, HEK293T cells (5 × 10⁶) were seeded in 10 cm tissue culture dishes and transfected after 24 h with 6 µg packaging plasmid (psPAX2), 2.5 µg envelop plasmid (pMD2.G) and 10 µg transfer plasmid (PL-SIN-PGK-GPCR-P2A-EGFP-WPRE-miR9T). Culture supernatant containing lentivirus was harvested 24 and 48 h following transfection. Supernatants from both harvests were pooled, passed through a 0.45 µm filter, and stored at −80 °C for transduction of HMC3 cells. For primary microglia transduction, supernatants were concentrated through ultracentrifugation (112,000 × $g$; 1.5 h; 4 °C) using a 20% sucrose cushion. Pelleted virus was resuspended in PBS and stored at −80 °C.

For titration of lentivirus preparations, HEK293T cells were seeded into 6-well plates (10⁵ per well) together with a defined volume of virus in various dilutions. After 72 h, the percentage of EGFP-positive cells was quantified through FACS. Non-transduced cells were used to set the threshold for the EGFP signal. The titer was calculated as transforming units per milliliter (TU/ml) according to the following formula:

$$\frac{TU}{ml} = \frac{\#Cells\,transduced\left(10^5\right) * \%EGFP\text{-positive} * Virus\,dilution\,factor}{Virus\,volume\,in\,ml * 100}$$

(1)

## Generation of HMC3 cell lines stably expressing DREADD-based chimeras

HMC3 cells were seeded into 6-well plates (32,000 per well) together with lentiviral vectors encoding GPCR-P2A-EGFP at a multiplicity of

infection (MOI) of 5. Cultures were then expanded for subsequent cell sorting to obtain a pure transduced cell population. For this, cells were trypsinized, pelleted (200 × $g$; 5 min; room temperature) and resuspended in 0.22 μm sterile filtered FACS buffer containing 2% (w/v) FBS (Sigma; 12103 C; heat-inactivated for 30 min at 56 °C) and 1 mM EDTA in HBSS without $Ca^{2+}/Mg^{2+}$. EGFP-positive singlets were sorted into EMEM-complete medium using a Sony SH800SFP cell sorter with a 100 μm nozzle chip. Non-transduced cells were used as a negative control to set the threshold for the EGFP signal. The sorting mode was set to "purity" to ensure that only EGFP-expressing cells were included. Culturing of these cells was continued under the above-described maintenance conditions.

### Gene expression profiling in HMC3 cells with RT-qPCR

Non-transduced HMC3 cells or HMC3 cells stably expressing DREADD-based GPCR chimeras were seeded in 6-well plates at a density of 32,000 cells per well in a total volume of 2 ml. Assays were performed three days after seeding when cells were approximately 80% confluent. Cells were then treated by applying fresh EMEM-complete medium containing the respective compounds. Concentrations of levalbuterol, CNO or forskolin were always 10 μM. IFNγ/IL1β was added at 10 ng/ml each. Untreated control conditions only received fresh EMEM-complete medium. Every experimental repetition included one well per condition. After 6 h incubation (37 °C and 5% $CO_2$), wells were briefly washed with DPBS before proceeding with RNA isolation (innuPREP RNA Mini Kit 2.0; Analytik Jena; 845-KS-2040050) according to the manufacturer's instructions. cDNA was synthesized immediately afterwards (Lunascript RT Super Mix; New England BioLabs; E3010L) with 800–1000 ng total RNA as input (same amount for each condition within experimental repetitions) and stored at −20 °C.

For gene expression analysis, RT-qPCR (Luna Universal qPCR Master Mix; New England BioLabs; M3003L) was performed in 384 well plates (Bio-Rad; HSR4805) on a Roche Lightcycler 480 using the device's "Second Derivative Maximum Method". The total reaction volume was either 5 or 10 μl containing 1 μl of 1:40 or 1:10 diluted cDNA, respectively. The final concentration for each primer was 0.25 μM (Supplementary Table S2). Cycle conditions were 60 s at 95 °C for initial denaturation, followed by 40 cycles of denaturation (15 s; 95 °C) and annealing/extension (30 s; 60 °C). Each run was completed with a melting curve analysis to confirm amplification of only one amplicon. Each PCR reaction was run in triplicates from which a mean Cq value was calculated and used for further analysis. dCq values were obtained by normalizing mean Cq values to the geometric mean of four reference genes (*GAPDH*, *ACTB*, *OAZ1*, *RPL27*) measured within the same sample. ddCq values were then calculated by normalizing dCq values to the respective control condition (untreated cells or cells stimulated with IFNγ/IL1β alone) within each experimental repetition. Fold changes were obtained by transforming ddCq values from log2 to linear scale.

Equations for consecutive RT-qPCR normalization:

$$dCq = \text{geometric mean}_{\text{reference genes}} - Cq \qquad (2)$$

$$ddCq = dCq - dCq_{\text{control condition}} \qquad (3)$$

$$\text{Fold change} = 2^{ddCq} \qquad (4)$$

For Fig. 8, fold changes were obtained through normalizing to untreated cells. For final data visualization, the stimulation with IFNγ/IL1β alone was then set to 100% within each experimental repetition. In Supplementary Fig. S16d, e, fold changes were calculated by directly normalizing to IFNγ/IL1β alone. Log2-transformed fold changes were then used for principal component analysis in *R* using the "prcomp" function with the "center" and "scale" argument set to "TRUE".

Hierarchical clustering of log2 fold changes was carried out with the *pheatmap* package (RRID:SCR_016418).

### Next generation mRNA sequencing of HMC3 cells

HMC3 cell lines were treated and RNA was isolated as described under *Gene expression profiling in HMC3 cells with RT-qPCR*. RNA samples were immediately snap frozen on dry ice and then stored at −80 °C. Samples were collected in batches (experimental repetitions) to obtain a total of three replicates per experimental condition. Library preparation and sequencing were carried out by the Vienna BioCenter Core Facility. In brief, libraries were generated with the QuantSeq 3' mRNA kit (Lexogen) and sequenced on an Illumina NextSeq550 SR75 High platform. Transcript abundance was quantified with *Salmon*[96] and we used the resulting "quant.sf" files as input for our downstream analysis with the *DESeq2*[97] package in *R*.

The "quant.sf" files were imported with the *tximport* package with the "countsFromAbundance" argument set to "no". This generates count data and omits correction for transcript length which is not necessary for 3'-mRNA sequencing data. Count data were then imported into *DESeq2* with the "DESeqDataSetFromMatrix" command, using "Experimental repetition" and "Experimental condition" as predictor variables for the design formula. For principal component analysis and sample-to-sample distance (Euclidean) calculation, counts were transformed with the "rlog" command and the "blind" argument set to "TRUE", which avoids bias by disregarding experimental group dependencies. Principal component analysis was performed in *DESeq2* with the "plotPCA" command using all genes. Hierarchical clustering of sample-to-sample distances was carried out with the *pheatmap* package. To identify differentially expressed genes, all experimental groups were included in one model. Subsequently, desired comparisons between experimental groups were extracted by specifying contrasts and conducting the Wald test (*DESeq2* default). P-values were adjusted with the "Benjamini-Hochberg" procedure with an alpha threshold of 0.1. Finally, we filtered the output of these comparisons and included only genes with an absolute linear fold change greater than 2. Supplementary Data 2 provides a list of all genes that passed these criteria for differential expression. For visualization of differentially expressed genes via heatmaps (*pheatmap* package), correlation plots, or bar graphs (Fig. 9f, g, Supplementary Fig. S13b, Supplementary Fig. S14, Supplementary Fig. S15, Supplementary Fig. S16), we operated on normalized counts extracted directly from the *DESeq2* model with the "counts" function. Pearson correlation coefficients were calculated with the "cor.test" function of *R* after log2-transforming the mean of respective fold changes. Gene ontology enrichment of biological processes was performed with the *topGO* package[98,99]. GO terms were mapped to the "org.Hs.eg.db" annotation database including only nodes with a minimum of 10 associated genes. Differentially expressed genes were analyzed for enrichment against a background including all genes where *DESeq2* was able to calculate an adjusted p-value, which excludes non-detected and unreliable low-abundance genes with mostly 0 counts. Enrichment was identified through the Fisher test and using the "elim" algorithm, which aims to eliminate broad and unspecific terms of parent nodes in case a more informative child node can be allocated.

### Ca²⁺ imaging of HMC3 cells

HMC3 cells lines were seeded in uncoated 8-well chamber slides (ibidi; 80826; growth area: 1cm²) at 3,500 cells per well and within a total volume of 200 μl. After three days, cells reached approximately 80% confluency and were labeled with Fluo-4 (Invitrogen; F10471; reconstituted at 1X in supplied buffer; 37 °C) for 30 min at 37 °C and 5% $CO_2$. Afterwards, cells were further incubated at room temperature (protected from light) and atmospheric $CO_2$ for another 30 min. Labeling solution was then replaced with 270 μl $CO_2$-independent medium (Leibovitz's L15, 10% (v/v) FBS and 1% (v/v) penicillin-streptomycin).

Samples were transferred to a confocal microscope and $Ca^{2+}$ imaging was performed at room temperature using a 20x air objective with the pinhole fully opened. Single-plane 16 bit images were acquired with a frame rate of 500 ms for a total duration of 6 min. After a 3 min baseline recording, a pipette was used to carefully apply 30 µl of ATP, levalbuterol or CNO (prepared in $CO_2$-independent medium; 10 times more concentrated than the desired concentration), or an equal amount of vehicle (medium only). Final concentrations were 1 mM for ATP and 10 µM for levalbuterol and CNO.

Images were processed in Fiji 1.51 by applying a Gaussian filter with a sigma of 1.5. ROIs were drawn on the center of individual cells and intensity was measured for each frame. For the generation of graphs, the intensity of each cell was normalized to its average intensity throughout the entire 6 min recording. For Supplementary Fig. S11b, normalized intensities were further re-scaled between 0 and 1 within each panel. $Ca^{2+}$ events were automatically detected in Matlab 2017a with the software PeakCaller (https://hussmanautism.org/resources/software)[84] using the following parameters: required rise = 20% absolute; max. lookback = 700 pts; required fall = 30% absolute; max. lookahead = 700 pts; trend control = exponential moving average (2-sided); trend smoothness = 100; interpolate across closed shutters = true. To remove erroneously detected $Ca^{2+}$ events, the output was additionally filtered in R by including only peaks with a height greater than 0.2 and a FWHM greater than 5.

## Primary microglia cultures

Primary microglia were obtained with adaptations from Bronstein et al.[100]. For one preparation, three to four C57BL6/J mouse pups aged P0-P3 were used. Animals were sprayed with 70% (v/v) ethanol for disinfection before decapitation. Heads were placed in a 10 cm on ice containing cold HBSS without $Ca^{2+}/Mg^{2+}$. Brains were removed and placed into a fresh 10 cm dish with cold HBSS. Under a dissection microscope, meninges were removed before dissecting the cortices, which were subsequently collected in a tube containing 15 ml cold HBSS on ice. HBSS was aspirated and 4 ml of Trypsin-EDTA (ThermoFisher; 25300-054; 37 °C) was added. The tissue was then triturated with a 1000 µl pipette tip and incubated at 37 °C for 15 min in a water bath. Digestion was stopped by adding 4 ml of HEK-complete medium (37 °C). Samples were pelleted (500 × g for 5 min at room temperature) and resuspended in 4 ml of HEK-complete medium. The previous centrifugation step was repeated and pellets were resuspended in 10–15 ml HEK-complete medium. The cell suspension was passed through a 40 µm cell strainer (Szabo Scandic; 352340) and then transferred to a T75 flask to establish a mixed glia culture at 37 °C and 5% $CO_2$. After 3 days, medium was replaced with 10 ml of fresh HEK-complete medium. Following a total period of 10–14 days after dissection, microglia were harvested from mixed glia cultures through a combination of lidocaine treatment and shaking. A 150 mM lidocaine solution (Sigma; L5647) was prepared in HBSS containing $Ca^{2+}/Mg^{2+}$ and sterile filtered (0.22 µm). Lidocaine was added the T75 flask to a final concentration of 15 mM before placing them on a shaker inside a cell culture incubator (37 °C and 5% $CO_2$) at 70 rpm for 25–30 min. After this incubation, the supernatant containing detached microglia was collected in a 50 ml tube. The flask was briefly washed with 5 ml HBSS containing $Ca^{2+}/Mg^{2+}$ to gather any remaining microglia and the content was pooled with the previously collected supernatant. EDTA was added to a final concentration of 0.05 mM before pelleting microglia (1000 × g; 5 min; room temperature). Cells were resuspended in 500 µl of HEK-complete medium with a wide 1000 µl pipette tip to avoid shear stress. Live cells were counted from a dilution in trypan blue (Sigma; T8154). The concentration was adjusted with HEK-complete medium to 0.2–0.25 million cells/ml to seed approximately 40,000–50,000 cells per well in uncoated 8-well chamber slides (ibidi; 80826; growth area: 1cm²) within a total volume of 200 µl.

## Live imaging of primary microglia

Primary microglia were transduced with lentiviral vectors encoding for DREADD-β2AR-P2A-EGFP approximately 4–24 h after seeding. Virus was applied at an MOI of 0.5–3 which resulted in sparsely transduced cells and live imaging was carried out five to seven days after transduction. Three to four days before live imaging, HEK-complete medium was exchanged with freshly prepared TIC medium optimized for primary microglia culture[101]. TIC medium consisted of DMEM/F12 (ThermoFisher; 31331093; with GlutaMAX) containing 5 µg/ml N-acetyl-L-cysteine (Sigma; A9165), 5 µg/ml bovine insulin (Sigma; I6634), 100 µg/ml human apo-transferrin (Sigma; T1147), 100 ng/ml sodium selenite (Sigma; S5261), 2 ng/ml human TGF-β2 (PepoTech; 100-35B), 100 ng/ml murine IL34 (R&D Systems; 5195-ML-010/CF), and 1.5 µg/ml ovine wool cholesterol (Sigma; 700000P). For live imaging, primary microglia were labeled with tomato lectin conjugated to DyLight 649 (ThermoFisher; L32472), which was first subjected to ultrafiltration to reduce the concentration of cytotoxic $NaN_3$. The required amount of tomato lectin was diluted in PBS (5 ml) and applied to a Vivaspin 6 concentrator (Sartorius; VS0601; MWCO: 10 kDa; 5 ml volume). After centrifugation (4000 × g; 10–15 min; 4 °C), the concentrate was diluted in DMEM/F12 (37 °C) to obtain a final tomato lectin concentration of 5 µg/ml (1:200 dilution of stock). Labeling took place for 20 min (37 °C and 5% $CO_2$), after which medium was replaced with 270 µl $CO_2$-independent Leibovitz's L15 (ThermoFisher; 21083027; no phenol red; room temperature). Samples were then transferred to a confocal microscope and z-stacks were acquired with a 20x air objective every minute for a total period of 55 min at room temperature. In all samples, a tomato lectin and EGFP channel was obtained through simultaneous scanning. The tomato lectin channel was used as autofocus reference which was applied before each z-stack to compensate for vertical drifting. After a 10 min baseline recording, a pipette was used to carefully apply 30 µl of levalbuterol or CNO (prepared in Leibovitz's L15; 10 times more concentrated than the desired concentration of 10 µM), or an equal amount of vehicle (Leibovitz's L15 only).

Images were processed in Fiji 1.51 by converting z-stacks to maximum intensity projections, applying a gamma correction of 0.75 for better visualization of faint signals, followed by rolling ball background subtraction and a 1×1 median filter. Regions of interest were generated by cropping individual cells. In cases where lateral drifting occurred, image registration was performed with Fiji's *StackReg* plugin using the *Rigid Body* transformation. The tomato lectin signal was used to quantify changes in cell area for non-transduced primary microglia. Microglia transduced with DREADD-β2AR-P2A-EGFP were analyzed through the tomato lectin or EGFP channel, depending on which one provided the best signal. A threshold was applied to separate signal from background. The thresholded area was converted to a binary image and subjected again to a 1 × 1 median filter to remove unspecific signals. Any remaining signals that did not belong to the respective cell were either removed manually or with Fiji's *Analyze Particle* function. Subsequently, this binarized area was measured in µm² at all time points during the 55 min recording. For the purpose of data visualization, a fold change was calculated for each cell by normalizing area to the mean of the baseline measurement. For the final analysis, values from all cells were pooled at representative time points of 1, 5, 10, 25, 40, and 55 min.

## Animals

C57BL/6 J (#000664) mice were purchased from The Jackson Laboratories. All animals were housed within the Preclinical Facility at ISTA. The facility has been approved by the authorities (Austrian Federal Ministry of Science, Research and Economy) under the following license numbers: BMWFW-66.018/003-II/3b/2014 and BMWFW-66.018/2-WT/U/3b/2015. All animals are housed in commercially available individually ventilated cages (IVCs) made of

Polysulfon under precisely defined standard laboratory conditions (room temperature $22 \pm 1\,^\circ C$; relative humidity $55 \pm 10$ %; photoperiod 12 L:12D), supplied with a standard diet (rat/mouse maintenance diet (V1534-300) or mouse breeding diet (V1124-300), ssniff Spezialitäten GmbH) and autoclaved water ad libitum. In addition, all animals are housed according to the maximum numbers per cage according to the Österreichische Tierversuchsverordnung (522. Verordnung) with a solid cage floor, dust-free bedding (LTE E-001 woodchips, Tapvai Estonia OÜ) and nesting material (HS Zellstofftupfer, Henry Schein; Sizzle nest, Plexx B.V.). Single housing will be used in exceptional cases only and temporary (e.g. in case of aggressive behavior or during recovery from surgery). Supplementary enriched environment is provided in each cage (play tunnels, gnawing sticks, house, and running wheel). Only skilled staff is allowed to handle the animals. All animals are inspected once per day, and a veterinarian is available on campus any time. All animal procedures are approved by the Bundesministerium für Wissenschaft, Forschung und Wirtschaft (bmwfw) Tierversuchsgesetz 2012 (TVG 2012), BGBl. I No. 114/2012, idF BGBl. I No. 31/2018 under the numbers 66.018/0005-WF/V/3b/2016, 66.018/0010-WF/V/3b/2017, 66.018/0025-WF/V/3b/2017, 66.018/0001_V/3b/2019, 2020-0.272.234.

## Statistical analysis

All analyses were performed with *R*. Data were collected in excel files and imported into R via the *xlsx* or *readxl* package. Linear regression models were generated with the *lme4* package[102] and after changing the default contrast for unordered variables (*e.g.* experimental condition) to "contr.sum". This allows to run type III Anova on the model to evaluate the overall contribution of unordered effects on the response variable. Post-hoc tests were performed via the *multcomp* package[103] with default parameters. If not otherwise indicated, all possible pairwise comparisons were performed. Significance levels are indicated by asterisks ($^{n.s.}p > 0.05$; $^*p \le 0.05$; $^{**}p \le 0.01$; $^{***}p \le 0.001$). Details about *R* environment, attached packages, statistical models, as well as results of the statistical analysis for each figure are found in Supplementary Data 3.

All graphs for data visualization were generated with the *ggplot2* package. Error bars or ribbons represent either standard error of the mean (SEM) calculated by the "mean_se" function (part of *hmisc* package; called through *ggplot2*) or show 95% confidence intervals around a smoothed line generated by the "geom_smooth" function (called through *ggplot2*; using the "loess" method for fitting).

## Real-time measurement of increases in cAMP levels upon ligand treatment

We used linear regression to predict the log-transformed luminescence values (baseline-normalized) by an interaction of Time (repeated measurements at regular intervals) and Experimental condition, which is an interaction of Treatment period (Baseline or Ligand), Receptor (GPCR or Empty vector), Ligand (CNO or Levalbuterol) and Concentration ($0.01\,\mu M$, $0.1\,\mu M$, $1\,\mu M$ or $10\,\mu M$). A random effect (Experimental repetition) was included to account for the dependency of data, which results from repeated measurements within each individual experiment. This model was used to test whether ligand treatment of individual Receptor-Ligand-Concentration interactions results in significant differences from the baseline measurement.

## Baseline cAMP levels in the absence of ligand (constitutive activity)

We used a two-sided one-sample T-test to investigate whether ligand-treated conditions are significantly different from a value of 1, which represents the empty vector control of the respective experimental repetition.

## SRE reporter assays

We used a two-sided one-sample T-test to investigate whether ligand-treated conditions are significantly different from a value of 1, which represents either the vehicle control or the empty vector control of the respective experimental repetition.

## Real-time assay measurement of β-arrestin 2 recruitment

We used linear regression to predict luminescence values (normalized to baseline and further to the respective vehicle control) by and interaction of Time (repeated measurements at regular intervals) and Experimental condition, which is an interaction of Treatment period (Baseline or Ligand) and Receptor (β2AR-SmBiT, DREADD-β2AR-SmBiT, DREADD-GPR65-SmBiT, or DREADD-GPR109A-SmBiT). A random effect (Experimental repetition) was included to account for the dependency of data, which results from repeated measurements within each individual experiment. This model was used to test whether ligand treatment of each Receptor results in significant differences from the baseline measurement.

## Western blot to quantify ERK1/2 phosphorylation

We used linear regression to predict ratios between pERK1/2 and GAPDH by Experimental condition, which represents different treatment durations (untreated, 2 min, 5 min, or 15 min). A random effect (Experimental repetition) was included to account for the dependency of data that are derived from the same experimental repetition.

To analyze basal ERK1/2 phosphorylation in untreated samples, we used a two-sided two-sample *T*-test and compared GPCR- with empty vector-transfected conditions.

## GPCR internalization assay

We used linear regression to predict internalized area in $\mu m^2$ (measured individually per cell based on thresholded VSV-G signal) by Experimental condition, which is an interaction of Ligand (CNO or Vehicle) and Treatment period (0 min, 15 min, 30 min or 60 min). This model was used to test whether CNO treatment shows significant differences from vehicle controls at corresponding Treatment periods.

## Real-time measurement of decreases in cAMP levels upon ligand treatment

We used linear regression to predict the log-transformed luminescence values (normalized to baseline and further to the respective vehicle control) by Experimental condition, which is an interaction of Treatment period (Baseline, Ligand or Forskolin) and Receptor (GPCR or Empty vector). A random effect (Experimental repetition) was included to account for the dependency of data, which results from repeated measurements within each individual experiment. This model was used to test Receptor and Empty vector for significant differences between their three Treatment periods. The time intervals of repeated measurements were not included as a predictor in the model as it was not necessary to improve the fit. This is because measured values are rather uniformly distributed within the three different Treatment periods, meaning that this variably can already explain most of the variability in the data.

## CRE reporter assay

We used a two-sided one-sample T-test to investigate whether ligand-treated conditions are significantly different from a value of 1, which represents the vehicle control of the respective experimental repetition.

## Gene expression profiling in HMC3 cells using RT-qPCR

We used a two-sided one-sample T-test to confirm that recombinant cytokine stimulation induces inflammatory gene expression. We compared the linear fold change of stimulated conditions to a value of

1, which represents the untreated control of the respective experimental repetition.

For further comparison of different treatment conditions, we used linear regression. We predicted ddCq values for individual transcripts by Experimental condition, which represents the treatment with different compounds alone or in combination. A random effect (Experimental repetition) was included to account for the dependency of data derived from the same experimental repetition. Separate models were generated for each investigated transcript (IL6, TNF, and IL1β) and cell line (non-transduced, DREADD-β2AR, DREADD-GPR65, and DREADD-GPR109A). These models were used to test for significant differences between different treatments.

### Next generation mRNA sequencing of HMC3 cells
We used *DESeq2*[97] to model gene expression and included Experimental repetition and Experimental condition as predictor variables. Experimental repetition accounts for sample dependencies and batch effects. Experimental condition is an interaction of cell line and treatment. One model was generated for the entire data set and desired comparisons were subsequently extracted by setting contrasts for the Experimental conditions to be tested against each other. The results are provided in Supplementary Data 2.

### Live imaging of primary microglia
We used linear regression to model the change of total cell area in μm² by using an interaction of the two predictors Time and Experimental condition. The experimental condition itself is an interaction of Treatment period (Baseline or Ligand) and Experimental group (Non-transduced_Levalbuterol, DREADD-β2AR_CNO, Non-transduced_Vehicle, or Non-transduced_CNO). A random effect (Cell ID) was included to account for the dependency of data, which results from repeated measurements on individual cells. This random effect also accounts for size differences between cells. This model was used to test for significant differences between the two Treatment periods within each Experimental group.

Alternatively, we also compared Experimental groups with each other within designated Time points (1, 5, 10, 25, 40, 55 min). For this, we used baseline-normalized values as response variable and an interaction of Time point and Experimental group as predictor. No random effect was included. This linear model was used to test for significant differences between the four Experimental groups within each Time point.

### Statistics and reproducibility
All VSV-G immunostainings to confirm GPCR surface expression (Figs. 2b, 7a, b, Supplementary Figs. 2c–g, S6b, 9a–c, 16c) have been successfully repeated in at least two experimental repetitions.

### Reporting summary
Further information on research design is available in the Nature Research Reporting Summary linked to this article.

## Data availability
The mRNA sequencing data of the retina transcriptome from Siegert et al.[48] has been accessible through GEO Series accession number GSE33089. Information on GPCR sequences was collected from IUPHAR/BPS (www.guidetopharmacology.org). Protein crystal structures were collected from the PDB database (www.rcsb.org) for visualization of bovine RHO (www.rcsb.org/structure/1u19), rat CHRM3 (www.rcsb.org/structure/4U15) as surrogate for hM3Dq, and human β2 AR (www.rcsb.org/structure/2RH1). The mRNA sequencing data have been deposited in NCBI's Gene Expression Omnibus and are accessible through GEO Series accession number GSE194125. The data for each graph generated in this study are provided in the Source Data file and under doi.org/10.15479/AT:ISTA:11542. Source data are provided with this paper.

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

## Acknowledgements

The authors thank the Scientific Service Units at ISTA, in particular the Molecular Biology Service of the Lab Support Facility, Imaging & Optics Facility, and the Preclinical Facility, and the Novarino group, Harald Janoviak, and Marco Benevento for sharing reagents and expertise. This research was supported by a DOC Fellowship (24979) awarded to R.S. by the Austrian Academy of Sciences.

## Author contributions

Conceptualization and Methodology: R.S. and S.S.; Formal Analysis: R.S.; Validation and Investigation: R.S., M.K-D, A.V., G.C.; Writing of Original Draft and Visualization: R.S. and S.S.; Supervision: S.S.

## Competing interests

The authors declare no competing interests.
