## [Peer Review File · Nature Communications]

Chimeric GPCRs mimic distinct signaling pathways and modulate microglia responsesEditorial Note: Parts of this Peer Review File have been redacted as indicated to maintain the confidentiality of unpublished data.

REVIEWER COMMENTS

Reviewer #1 (Remarks to the Author):

The goal of this paper is to generate new DREADD versions of GPCRs that have important functions in microglia. The authors make 3 of these by swapping the internal loops and C-terminal of the receptor of interest onto the CNO-activated DREADD. The DREADD chimeras recapitulate the signaling of the expected receptor: instead of Gq for the original muscarinic receptor based DREADD, Gs for B2R and GPR65 and Gi for GPR109A. The functionality carries on to several downstream signaling systems for the DREADD-B2R, including cAMP production, MAP kinase activity, phosphorylation of ERK, induction of CRE-dependent reporter gene expression and expression in microglial cells of inflammatory genes as well as induction of filopodia, and for the DREADD-GPR65 cAMP production and MAP kinase activity and expression in microglial cells of inflammatory genes. The DREADD-GPR109A story is not clear in these analyses. The authors conclude that they have created CNO-activated versions of the 3 receptors that function as do the normal receptors.

The analysis shows that DREADD-B2R and DREADD-GPR65 do the same thing. This is consistent with their Gs coupling, but it is not clear if they function differently from the previously described Gs-DREADD. The great hope is that each chimera will have its own unique features that acts completely like its parent receptor that donated the internal portions. To determine this one would need broader assays that reveal the differences between receptors even when they couple to the same G protein. The most promising version of this that the authors approach is in transcriptomics analysis, which if done broadly enough (by looking at hundreds or thousands of genes) may well identify unique features. However, the authors look only at a small trio of important proteins for inflammation in microglia where the Gs receptors all do the same thing, so this question is not answered. It would help if the authors would compare Gs-DREADD to their chimeras.

The paper is a challenge to read. To gather the information one needs to understand the experiments and the figures one has to jump between Results, Figures, Legends, Methods and Supplement, and even then the information is not always apparent.

Specific Points

1) In Fig 2g activation of DREADD-B2R with CNO or of B2R with Levalbuterol reduced luciferase activity, consistent with expectation for Gs activation. The same was seen with empty vector. The authors attribute this to endogenous B2R in HEK cells. However, the reduction in luciferase is greater with empty vector (endogenous B2R alone) than with B2R (transfected B2R + endogenous). This is opposite of expectation and should be resolved, perhaps in another cell line that does not express native B2R.

2) Fig 4 staining is a little hard to understand. Why is there no VSV-G or EGP signal at 0 min in CNO condition? The expectation is that the vast majority would be at the plasma membrane.

Results and Methods do not describe how internalization was quantified. Fig 4d indicates that internalized area was measured. Why is area the right measure and not fluorescence? What area? GFP or VSV-G? In some images these coincide, as expected, but in others very large GFP areas have very little VSV-G labeling in very small spots. This makes the meaning of the staining pattern unclear.

The number of consecutive optical slices is mentioned but not their thickness or interval. A threshold was applied and background subtraction. Were these done on individual cells or uniformly for all cells in the well?

Methods describe that a handful of cells were imaged for each condition. How were these selected? Is each dot in Fig 4d a cell or an average of cells from a well? How much of the large scatter is due to differences in internalization and how much to differences in expression level and detection limit?

3) Fig 6e shows very small DREADD-GPR109A cAMP reduction, but also shows no effect of forskolin in either DREADD-GPR109A or untransduced cells. This raises concern about the assay. The results would be easier to interpret if compared to GPR109A with its native ligand, as done with B2R.

4) Fig. 6f,g and Suppl Fig. S5a, b are difficult to understand. This is made harder by the fact that one needs to jump between Results, Figure Legend, Methods and main + suppl fig to try to reconstruct what was done and by the difference in they-axis between 6g (fold change in ligand vs vehicle [where the vehicle is not mentioned anywhere else]) and S5b which shows the effect of NECA as a luminescence ratio. This would be a minor point to fix, except for the apparent inconsistency that raises alarms. In S5b NECA robustly stimulates luciferase in untransduced HEK cells but in 6g it has little effect on "Empty Vector".

The authors state in Results "Subsequently, we transfected HEK cells with DREADD-GPR109A and applied CNO together with 5 μ M NECA. As anticipated, CNO significantly inhibited Gas-mediated transcription from the CRE. Why was NECA needed for this? Would not CNO activation of DREADD-GPR109A done the same as was the case with DREADD-B2R?"

It seems that the function of DREADD-GPR109A is much weaker than that of DREADD-B2R and this leads to these convoluted extra measures. On the other hand, beta arrestin recruitment is similar between DREADD-GPR109A and DREADD-B2R, so the mismatch in cAMP and CRE activation is hard to understand.

5) The transcriptomics analysis to see how well the DREADD chimeras recapitulate effects on gene expression of classical stimuli in microglia is the right test and could be very telling. The authors focused on 3 important genes and found congruence, which is encouraging. But this seems underpowered to really make a claim for full recapitulation.

Clustering analysis as in Fig. 7 would be more appropriate if the authors were examining a large number of genes. The conclusion that the DREADD chimeras are better at recapitulating the effects of IFN γ +IL1b than forskolin is easily seen by looking at the effects on IL1b expression.

6) In Fig 5e authors should describe more clearly what they are analyzing statistically? Is this the slope during the baseline period versus slope post-addition of drug?

Is there a statistical difference between CNO effect on cells expressing DREADD-B2R versus untransduced cells?

Reviewer #2 (Remarks to the Author):

In this study, Schulz et al. designed and engineered the DREADD-based GPCR chimeras to mimic the activation of microglial enriched GPCRs, including β 2AR, GPR65, and GPR109A, in vitro. Further, they verified the functional properties of these chimeric GPCRs from several aspects, including signaling transduction, receptor internalization, microglial morphological alteration, and modulation of inflammatory gene expression. The results suggest chimeric DREADD-based GPCRs could recapitulate some functions of GPCR-of-interest. This is a great attempt to manipulate the desired GPCR instead of the indiscriminate activation of the DREADD-based GPCR. Overall, the experiments were well designed and nicely performed. This study may extend DREADD approaches through a specific GPCR for studying their signaling pathways and function not only in microglia but also in other cells. Here are some concerns that need to be addressed.

Major concerns:

1. Authors suggest chimeric GPCRs can offer a fine-tuned dissection of specific GPCRs. This is an interesting idea but needs more experiments to support this suggestion. Particularly, how DREADD- β 2AR is different from traditional Gs-DREADD? Would chimeric GPCRs recruit different GRK? In Figure 2, the authors compared the function of Gq-DREADD (CHRM3) with DREADD- β 2AR (Gs) and

non-chimeric β 2AR (Gs). I highly recommend that authors include Gs-DREADD and Gi-DREADD as the control in some of their experiments.

2. The authors have performed nice validation of DREADD- β 2AR function, including kinase activity, cAMP induction, internalization, and phosphorylation of Erk. There are various signal transduction mechanisms based on the type of G-proteins that modulate intercellular calcium dynamics.

Considering the microglial calcium is highly tuned to neuronal activity (PMID: 32716294; PMID: 33369790), it might be interesting to add calcium signaling as additional readout after activation of chimeric DREADD- β 2AR, DREADD-GPR65, and GPR109A.

3. The authors transfected HEK cells with DREADD- β 2AR-P2A-EGFP vector and confirmed cytoplasmic EGFP localization on the cell membrane. Also, internalization of EGFP signal after CNO administration (Fig. 4). These results are convincing, but I would suggest the authors to examine the internalization of EGFP signal also in culture microglia in addition to HEK cells. Also, wondering whether EGFP may interfere with the protein trafficking?

4. It has been reported that increasing cAMP by activating β 2AR promotes filopodia but collapses large processes (PMID: 31167136). Similarly, inhibition of β 2AR increase microglial process surveillance in vivo (PMID: 31636449). I understand the technical challenge for the authors to identify filopodia and large processes in culture microglia. However, they might be able to use images in Figure 5 with higher magnification to analyze filopodia dynamics. At least they should discuss the caveat of the functional readout of filopodia in culture microglia (instead of microglial processes in vivo).

5. Figure 7 used HMC3 cells for the DREADD-based chimeras modulation of microglia under inflammatory conditions. The reason to choose this microglia-like cell line, is because of suboptimal transduction efficacies with available vectors the authors mentioned. However, the authors did successfully package DREADD- β 2AR-P2A-EGFP construct into lentiviral vectors and transduced primary microglia to observe filopodia formation. The authors should use primary microglia to manipulate chimeric GRCP in gene expression under inflammatory conditions.

6. In the neuropathic pain model, activating microglial Gi-DREADD could decrease IL-1 β expression (PMID: 33221486), while in this study, activating DREADD-GPR109 did not. In addition, activation of DREADD- β 2AR or DREADD-GPR65 did not change IL1 β expression. Please discuss the discrepancy and how G protein signaling regulates the expression of inflammatory genes.

Reviewer #3 (Remarks to the Author):

This is an interesting and potentially important paper which expands the chemogenetic toolkit to include potential orphan GPCR constructs which appear useful for interrogating their biology.

Specific comments:

1. A Gs-DREADD has previously been described based on the b1-AR and I'm surprised it wasn't mentioned (ref: <https://www.pnas.org/content/106/45/19197.long>). Thus the b2-AR GS DREADD is not unique but might be useful and this should be mentioned.

2. The previously described GS-DREADD (from the Wess group at NIH) had constitutive activity; it is not at all clear that constitutive activity was measured at any of these constructs and I'd be surprised if they do not display at least some basal activity. The standard way to measure this is to vary the expression of the construct and to measure basal activity (this would require another type of cAMP assay) and compare with a receptor relatively devoid of constitutive activity (see *Journal of Biological Chemistry* 268 (7), 4625-4636).

3. In Fig 3 it appears there is some basal ERK data for the DREADD. This should be tested statistically (the figure looks to be N=1 which is not suitable)

Point-by-point response to REVIEWER COMMENTS

To all Reviewers:

We thank the reviewers for their valuable input and have addressed each comment using the following color-code:

Reviewers' comments, in **black**

Reviewer #1, in **green**

Reviewer #2, in **blue**

Reviewer #3, in **brown**

In addition, we performed the following new experiments in the revised manuscript:

1. We included two additional controls for DREADD function, rM3Ds and hM4Di, and validated their effect on the level of cAMP and MAPK activity. The results are shown in **Figure 2**. As a note, we replaced the previously used term "DREADD alone" with the actual DREADD name hM3Dq.
2. We evaluated constitutive activity in cAMP and MAPK signaling in the absence of ligand stimulation. The results are shown in **Figure 3** and **Supplementary Figure S8**.
3. We investigated whether our DREADD-GPCRs mimic receptors that can trigger Ca²⁺ signaling in microglia. To address this, we performed Ca²⁺ imaging in HMC3 cells. The results are shown in **Supplementary Figure S11**.
4. We performed RNA sequencing of the HMC3 cell lines to compare how endogenous β 2AR and chimeric DREADD- β 2AR, DREADD-GPR65, or DREADD-GPR109A modulate the IFN γ /IL1 β -induced inflammatory gene expression signature. The results are shown in **Figure 9** and **Supplementary Figures S12-15**.

Reviewer #1 (Remarks to the Author):

The goal of this paper is to generate new DREADD versions of GPCRs that have important functions in microglia. The authors make 3 of these by swapping the internal loops and C-terminal of the receptor of interest onto the CNO-activated DREADD. The DREADD chimeras recapitulate the signaling of the expected receptor: instead of Gq for the original muscarinic receptor based DREADD, Gs for B2R and GPR65 and Gi for GPR109A. The functionality carries on to several downstream signaling systems for the DREADD-B2R, including cAMP production, MAP kinase activity, phosphorylation of ERK, induction of CRE-dependent reporter gene expression and expression in microglial cells of inflammatory genes as well as induction of filopodia, and for the DREADD-GPR65 cAMP production and MAP kinase activity and expression in microglial cells of inflammatory genes. The DREADD-GPR109A story is not clear in these analyses. The authors conclude that they have created CNO-activated versions of the 3 receptors that function as do the normal receptors.

The analysis shows that DREADD-B2R and DREADD-GPR65 do the same thing. This is consistent with their Gs coupling, but it is not clear if they function differently from the previously described Gs-DREADD.

We included $G\alpha_s$ -coupled DREADD (rM3Ds) in our cAMP and MAPK (SRE reporter) assay (Fig. 2e-g, i) to clarify this point. As expected from the $G\alpha_s$ -coupling, we found that rM3Ds behaves similarly to our DREADD- β 2AR. Our assays point out that DREADD- β 2AR mimics β 2AR more closely regarding their fold changes.

The great hope is that each chimera will have its own unique features that acts completely like its parent receptor that donated the internal portions. To determine this one would need broader assays that reveal the differences between receptors even when they couple to the same G protein. The most promising version of this that the authors approach is in transcriptomics analysis, which if done broadly enough (by looking at hundreds or thousands of genes) may well identify unique features. However, the authors look only at a small trio of important proteins for inflammation in microglia where the Gs receptors all do the same thing, so this question is not answered. It would help if the authors would compare Gs-DREADD to their chimeras.

We performed mRNA sequencing of our DREADD-GPCR expressing HMC3 cell lines. For more information, see specific point #5.

The paper is a challenge to read. To gather the information one needs to understand the experiments and the figures one has to jump between Results, Figures, Legends, Methods and Supplement, and even then the information is not always apparent.

For clarity, we decided to include small headings over each subpanel, which should make it more straightforward to follow the data.

Specific Points

1. In Fig 2g (now Fig. 2i) activation of DREADD-B2R with CNO or of B2R with Levalbuterol reduced luciferase activity, consistent with expectation for Gs activation. The same was seen with empty vector. The authors attribute this to endogenous B2R in HEK cells. However, the reduction in luciferase is greater with empty vector (endogenous B2R alone) than with B2R (transfected B2R

+ endogenous). This is opposite of expectation and should be resolved, perhaps in another cell line that does not express native B2R.

To assess the contribution of endogenous β 2AR, we performed our cAMP and MAPK (SRE reporter) assay with different concentrations of levalbuterol and also included norepinephrine, another β 2AR agonist with comparably lower potency (**Supplementary Fig. S3**). These new data showed that in the SRE assay, empty vector responses never exceed β 2AR overexpression and the expected difference between the two conditions becomes apparent at lower ligand concentrations (**Supplementary Fig. S3b**). Notably, β 2AR overexpression and empty vector are at the same level when using higher concentrations and a possible reason for this is the sensitive nature of this assay, which relies on an endpoint measurement after 6 hours of ligand exposure. During this time period, even weaker signals might accumulate and converge to the maximum response. On the other hand, when measuring cAMP levels in real time (**Supplementary Fig. S3a**), β 2AR overexpression always led to the stronger response compared to empty vector, regardless of ligand concentration. Besides including these results as new **Supplementary Figure S3**, we also adapted the main figure (**Fig. 2i**) and annotated the concentration of levalbuterol, which shows the expected difference more clearly.

2. Fig 4 (now **Fig. 5**) staining is a little hard to understand. Why is there no VSV-G or EGP signal at 0 min in CNO condition? The expectation is that the vast majority would be at the plasma membrane.

In the original manuscript, our intention with the “0min” time point was to show the VSV-G staining immediately after VSV-G antibody labeling and to represent the baseline without stimulation. We realized that our figure design was misleading and that a black space holder can be easily misinterpreted as a negative image. Therefore, we relabeled the treatment condition and show the example image as baseline, before treatment.

Results and Methods do not describe how internalization was quantified. Fig 4d (now **Fig. 5e**) indicates that internalized area was measured. Why is area the right measure and not fluorescence? What area? GFP or VSV-G? In some images these coincide, as expected, but in others very large GFP areas have very little VSV-G labeling in very small spots. This makes the meaning of the staining pattern unclear.

We revisited our initial method section (see italic text below) and ensured that we had described the measurement of thresholded VSV-G-positive area within individual cells.

*“Regions-of-interest were generated by cropping individual cells. For analysis, maximum intensity projection of six consecutive z-slices around the center of each cell were obtained. VSV-G and EGFP signals were used to **trace the perimeter along the cell surface which separates intra- and extracellular space**. A threshold was applied to separate signal from background. The **threshold VSV-G area within the perimeter was then measured** to quantify internalized GPCRs. To check if this signal is derived from internalized receptors, we confirmed colocalization of VSV-G and EGFP in both channels”.*

We decided to quantify area instead of fluorescence intensity because the latter depends on various imaging and image processing parameters. In contrast, the area of VSV-G-positive accumulation will reliably increase upon time with receptor internalization.

We also included a new schematic (**Fig. 5b**) to highlight the interplay between VSV-G and EGFP signals. The advantage of our strategy is that we are not only relying on the fused EGFP tag,

which shows receptors on the cell membrane but also labels receptors retained within the cell (potential sites are ER, Golgi, lysosomes), as is shown in **Figure 5c**. By using the VSV-G labelling to quantify internalization, we circumvent false positive results. At the same time, we can use co-localization with EGFP to confirm that VSV-G signals are derived from labelled receptors.

The number of consecutive optical slices is mentioned but not their thickness or interval. A threshold was applied and background subtraction. Were these done on individual cells or uniformly for all cells in the well?

We refined our methods section and included the resolution in x, y, and z (71 x 71 x 230nm) to provide the missing information about optical slice thickness. We also refined our description of the image analysis process to clarify the order in which the different steps have been conducted:

“Confocal imaging was performed with a 63x oil immersion objective. For each condition, several images each containing one to five cells were taken at random positions within the same well. Approximately 15 cells were acquired per condition with optimal resolution in x, y, and z (71 x 71 x 230nm). Entire images were processed in Fiji 1.51.f by rolling ball background subtraction followed by a 2x2x2 median filter. Regions-of-interest (ROI) were then generated by cropping individual cells and saving them as new images. For further analysis of individual cells, maximum intensity projections of six consecutive z-slices around the center of each cell were obtained. VSV-G and EGFP signals were used to trace the perimeter along the cell surface which separates intra- and extracellular space. A threshold was applied on the VSV-G channel to separate signal from background. The threshold VSV-G area within the cell surface perimeter was then measured (μm^2) to quantify internalized GPCRs. To check if this signal is derived from internalized receptors, we confirmed colocalization of VSV-G and EGFP in both channels.”

Methods describe that a handful of cells were imaged for each condition. How were these selected? Is each dot in Fig 4d (now **Fig. 5e**) a cell or an average of cells from a well? How much of the large scatter is due to differences in internalization and how much to differences in expression level and detection limit?

Our revised method section includes this information:

“For each condition, several images each containing one to five cells were taken at random positions within the same well. Approximately 15 cells were acquired per condition with optimal resolution in x, y, and z (71 x 71 x 230nm).”

We also checked our previous figure legend (see italic text below) and found that it describes each dot as an individual cell.

*Stimulation of DREADD- β 2AR-EGFP-transfected HEK cells with CNO (magenta) or vehicle (grey). **Each dot** shows internalized **VSV-G-positive area of a single cell**. Error bars: standard error of the mean within each condition. Magenta and grey lines connect the mean values of CNO or vehicle exposure times, respectively. Linear regression analysis: $p^* < 0.05$; $p^{n.s.} > 0.05$.*

As each data point represents an individual cell, the variability within the conditions could be a reflection of expression level differences but we cannot rule out that a limitation to detect very small amounts of internalized receptor might play role as well. To limit variability, we ensured that all cells in this experiment stemmed from the same passage, were seeded and transfected at the same time and with the same transfection mix.

3. Fig 6e (now **Fig. 7e**) shows very small DREAD-GPR109A cAMP reduction, but also shows no effect of forskolin in either DREAD-GPR109A or untransduced cells. This raises concern about

the assay. The results would be easier to interpret if compared to GPR109A with its native ligand, as done with B2R.

Based on the reviewer's comment, we adapted **Supplementary Figure S7a**, which shows raw luminescence values. These plots demonstrate that there is indeed strong cAMP induction upon forskolin treatment. We adapted the figure legend and included the reason for displaying raw values.

The revised **Figure 7e (previously Fig. 6e)** shows fold changes in CNO-treated conditions normalized to the respective vehicle control (which is represented by the dashed line). This normalization strategy does not demonstrate the cAMP increase upon forskolin application, as normalization to vehicle will always centre the data around a value of "1". Normalization is necessary to visualize the relative decrease of 15% induced by CNO stimulation, which is maintained in the presence of competing forskolin. At the same time, normalization also helps to account for inter-assay variabilities that influence the fold change upon forskolin stimulation in repetitions performed on different days. Therefore, we always normalized to the respective vehicle controls within each repetition. The image below exemplifies the difficulty of interpreting raw values due to different scales of baseline and forskolin induction, and due to high inter-assay variability (ribbons indicate 95% confidence intervals).

4. Fig. 6f,g (now **Fig. 7f, g**) and Suppl Fig. S5a, b (now **Supplementary Fig. S7a-b**) are difficult to understand. This is made harder by the fact that one needs to jump between Results, Figure Legend, Methods and main + suppl fig to try to reconstruct what was done and by the difference in they-axis between 6g (fold change in ligand vs vehicle [where the vehicle is not mentioned anywhere else]) and S5b which shows the effect of NECA as a luminescence ratio. This would be a minor point to fix, except for the apparent inconsistency that raises alarms. In S5b NECA robustly stimulates luciferase in untransduced HEK cells but in 6g it has little effect on "Empty Vector".

Figure 7f demonstrate the strategy of a competition assay, which is a common strategy for detecting $G\alpha_i$ responses (doi: 10.1016/j.bcp.2015.09.010). To interfere with $G\alpha_i$ signaling, we used NECA as competing agent to stimulate $G\alpha_s$.

Supplementary Figure S7b confirms $G\alpha_s$ induction through NECA and justifies the use of a 5 μ M concentration for the actual competition assay. **Supplementary Figure S7b** deliberately shows raw values (for this type of assay the firefly/renilla ratio is the raw value) to demonstrate the dose-dependency of this induction.

The actual result of the competition assay (**Fig. 7g**) uses a normalization strategy to evaluate the extent of competition. CNO treatments were normalized to the respective vehicle controls, which are represented by a value of “1” (dashed line). CNO treatment itself in the absence of DREADD-GPR109A does not compete with $G\alpha_s$, thus the empty vector condition does not significantly deviate from the vehicle control. This is also reflected in **Supplementary Figure S7b**, where CNO does not interfere with $G\alpha_s$ induction. Our normalization strategy allows quantification of relatively small differences and accounts for inter-assay variabilities, which are a common factor with luciferase assays.

We adapted the figure legend of **Supplementary Figure S7b** and described why raw values were chosen for this visualization. We also ensured that our method section describes how normalized values were calculated from raw data.

The authors state in Results “Subsequently, we transfected HEK cells with DREADD-GPR109A and applied CNO together with 5 μ M NECA. As anticipated, CNO significantly inhibited $G\alpha_s$ -mediated transcription from the CRE. Why was NECA needed for this? Would not CNO activation of DREADD-GPR109A do the same as was the case with DREADD-B2R?”

The detection of cAMP-mediated effects is rather straightforward for signal increases. However, with DREADD-GPR109A, we aimed to quantify a cAMP decrease, which is technically more challenging. Due to the natural low levels of baseline cAMP, they are often obscured by the detection limit of an assay. This is a common struggle in the field and is also addressed in publications (doi: 10.1016/j.bcp.2015.09.010). As a workaround, $G\alpha_i$ effects are often evaluated through competition with a compound that elevates cAMP levels above their baseline.

Nevertheless, we conducted the suggested experiment in the absence of NECA. As expected, we did not see a significant reduction with DREADD-GPR109A compared to the respective vehicle control. These data are now included in **Supplementary Figure S7c**. In the result section, we included the following statement:

*“As a note, CNO stimulation of DREADD-GPR109A did not dampen CRE reporter activity without simultaneous induction through NECA (**Supplementary Fig. S7c**), possibly because the assay is not suitable for detecting minor reductions from baseline levels⁶².”*

It seems that the function of DREADD-GPR109A is much weaker than that of DREADD-B2R and this leads to these convoluted extra measures. On the other hand, beta arrestin recruitment is similar between DREADD-GPR109A and DREADD-B2R, so the mismatch in cAMP and CRE activation is hard to understand.

Up- and downregulation of cAMP levels and β -arrestin 2 recruitment are detected through assays operating on different pathways and detection limits. Therefore, it is not possible to directly compare the strength of $G\alpha_s$ - and $G\alpha_i$ -coupled receptors based their behavior in different assay systems.

5. The transcriptomics analysis to see how well the DREADD chimeras recapitulate effects on gene expression of classical stimuli in microglia is the right test and could be very telling. The authors focused on 3 important genes and found congruence, which is encouraging. But this seems underpowered to really make a claim for full recapitulation.

We thank the reviewer for the suggestion and performed next generation mRNA sequencing of our HMC3 cell lines under inflammatory conditions (**Fig. 9**). Our results revealed a strong correlation between the signatures of β 2AR, DREADD- β 2AR and DREADD-GPR65 across more than 200 genes and allowed us to assess the capability of recapitulation in more depth.

Clustering analysis as in Fig. 7 (now **Fig. 8**) would be more appropriate if the authors were examining a large number of genes. The conclusion that the DREADD chimeras are better at recapitulating the effects of IFN γ +IL1 β than forskolin is easily seen by looking at the effects on IL1 β expression.

We decided to remove this clustering analysis based on our RT-qPCR results of three genes. Instead, we performed hierarchical clustering on our mRNA sequencing data to assess overall similarities (**Supplementary Fig. S12c**) and recapitulation of signature transcriptome profiles (**Fig. 9f**, **Supplementary Fig. S14**). We describe the outcome of this analysis in the results section:

*“To substantiate that DREADD- β 2AR recapitulates the endogenous β 2AR response, we performed next generation mRNA sequencing of our HMC3 cell lines with GPCR signaling under inflammatory conditions (**Supplementary Fig. S12a**). Principal component analysis and hierarchical clustering of sample-to-sample distances resulted in three clusters (**Supplementary Fig. S12b-c**, respectively): Cluster 1 summarized the biological triplicate of non-transduced HMC3 cells without exposure to inflammatory cytokines. Cluster 2 contained all cell lines that have been treated with IFN γ /IL1 β but without simultaneous GPCR stimulation. The only exception is DREADD-GPR109A. Ligand stimulation of DREADD- β 2AR, DREADD-GPR65, and endogenous β 2AR in non-transduced cells altered the inflammatory signature and formed Cluster 3. CNO-treated DREADD-GPCR109A stayed in Cluster 2, supporting the previous RT-qPCR data (**Fig. 8e**) and indicating that this receptor is not involved in modulating inflammation in HMC3 cells.”*

We found that DREADD- β 2AR recapitulates endogenous β 2AR with high similarity across more than 200 genes (**Fig. 9**). At the same time, we found that DREADD-GPR65, which is also coupled to G α_s , has a similar signature. We address this in the discussion section and state the following:

“We did not identify gene clusters unique to β 2AR/DREADD- β 2AR and DREADD-GPR65, suggesting that overall, these GPCRs modulate IFN γ /IL1 β -mediated inflammation in HMC3 cells interchangeably.”

6. In Fig 5e (now **Fig. 6e**) authors should describe more clearly what they are analyzing statistically? Is this the slope during the baseline period versus slope post-addition of drug?

We adapted the figure legend of **Figure 6e** and stated the following:

*“Linear regression analysis modeling cell area (μm^2) of individual cells across time to compare baseline with treatment period: $p^{***} < 0.001$; $p^{**} < 0.01$; $p^{n.s.} > 0.05$.”*

A more thorough description is provided in the method section on statistical analysis:

“We used linear regression to model the change of total cell area in μm^2 by using an interaction of the two predictors Time and Experimental condition. Experimental condition itself is an interaction of Treatment period (Baseline or Ligand) and Experimental group (Untransduced_Levalbuterol, DREADD- β 2AR_CNO, Untransduced_Vehicle, or Untransduced_CNO). A random effect (Cell ID) was included to account for the dependency of data, which results from repeated measurements on individual cells. This random effect also

accounts for size differences between cells. This model was used to test for significant differences between the two Treatment periods within each Experimental group.”

Supplementary Table S5 provides further details about the model, its predictors and the post-hoc comparisons.

Model	model <- lmer(Value ~ Time * Experimental condition + (1 Cell ID), data = Data, REML = TRUE)
Value	Total cell area in μm^2 .
Time	Numerical variable representing time points of measurements included in the analysis: 1, 5, 10, 25, 40, and 55min.
Experimental condition	Interaction term of Treatment period (Baseline or Ligand) and Experimental group (Untransduced_Levalbuterol, DREADD- β 2AR_CNO, Untransduced_Vehicle, or Untransduced_CNO). This generates the following conditions:  • Baseline:Untransduced_Levalbuterol • Ligand: Untransduced_Levalbuterol • Baseline:DREADD-β2AR_CNO • Ligand:DREADD-β2AR_CNO • Baseline:Untransduced_Vehicle • Ligand: Untransduced_Vehicle • Baseline:Untransduced_CNO • Ligand: Untransduced_CNO
Cell ID	Random effect accounting for the dependency of data due to repeated measurements on the same cell. Also accounts for size differences between cells.
Data	A single data frame containing all data at the 1, 5, 10, 25, 40, and 55min time points.
Posthoc contrasts	Within each Experimental group we compared the two Treatment periods with each other:  • Baseline vs. Ligand

Is there a statistical difference between CNO effect on cells expressing DREADD-B2R versus untransduced cells?

We thank the reviewer for suggesting this statistical comparison. Our initial analysis uses a random effect and takes individual cells and the resulting data dependency into account. We performed the suggested alternative test and conducted pairwise comparisons of the four experimental groups within each time point. In the treatment period, vehicle and CNO controls were significantly different from endogenous β 2AR stimulation and chimeric DREADD- β 2AR stimulation. We included a new graph in **Supplementary Figure S6d** to accompany this analysis. More details can be found in the respective methods section on statistics and in **Supplementary Table S5**, which explains the model, its predictors and the post-hoc comparisons.

Reviewer #2 (Remarks to the Author):

In this study, Schulz et al. designed and engineered the DREADD-based GPCR chimeras to mimic the activation of microglial enriched GPCRs, including β 2AR, GPR65, and GPR109A, in vitro. Further, they verified the functional properties of these chimeric GPCRs from several aspects, including signaling transduction, receptor internalization, microglial morphological alteration, and modulation of inflammatory gene expression. The results suggest chimeric DREADD-based GPCRs could recapitulate some functions of GPCR-of-interest. This is a great attempt to manipulate the desired GPCR instead of the indiscriminate activation of the DREADD-based GPCR. Overall, the experiments were well designed and nicely performed. This study may extend DREADD approaches through a specific GPCR for studying their signaling pathways and function not only in microglia but also in other cells. Here are some concerns that need to be addressed.

Major concerns:

1. Authors suggest chimeric GPCRs can offer a fine-tuned dissection of specific GPCRs. This is an interesting idea but needs more experiments to support this suggestion. Particularly, how DREADD- β 2AR is different from traditional Gs-DREADD? Would chimeric GPCRs recruit different GRK? In Figure 2, the authors compared the function of Gq-DREADD (CHRM3) with DREADD- β 2AR (Gs) and non-chimeric β 2AR (Gs). I highly recommend that authors include Gs-DREADD and Gi-DREADD as the control in some of their experiments.

We thank the reviewer for the excellent suggestion. We included rM3Ds and hM4Di as additional DREADDs in our validation on the level of cAMP and MAPK activity (Fig. 2e-f, i). As expected, we found that the $G\alpha_s$ -coupled rM3Ds performed similarly to our DREADD- β 2AR. Our data indicate that DREADD- β 2AR mimicked the non-chimeric β 2AR response with higher fidelity regarding fold change of cAMP induction and SRE reporter inhibition.

2. The authors have performed nice validation of DREADD- β 2AR function, including kinase activity, cAMP induction, internalization, and phosphorylation of Erk. There are various signal transduction mechanisms based on the type of G-proteins that modulate intercellular calcium dynamics. Considering the microglial calcium is highly tuned to neuronal activity (PMID: 32716294; PMID: 33369790), it might be interesting to add calcium signaling as additional readout after activation of chimeric DREADD- β 2AR, DREADD-GPR65, and GPR109A.

To address this, we utilized our stable HMC3 cell lines and performed Ca^{2+} imaging using the Ca^{2+} -sensitive dye Fluo-4 (Supplementary Fig. S11). As expected, HMC3 cells responded rapidly to ATP with increased fluorescence intensity (Supplementary Fig. S11b) and Ca^{2+} peaks identified through the PeakCaller software (Supplementary Fig. S11c). No Ca^{2+} transients were induced upon stimulation of endogenous β 2AR with levalbuterol (LB) or our three chimeric DREADD-GPCRs with CNO (Supplementary Fig. S11b-c).

3. The authors transfected HEK cells with DREADD- β 2AR-P2A-EGFP vector and confirmed cytoplasmic EGFP localization on the cell membrane. Also, internalization of EGFP signal after CNO administration (Fig. 4, now Fig. 5). These results are convincing, but I would suggest the

authors to examine the internalization of EGFP signal also in culture microglia in addition to HEK cells. Also, wondering whether EGFP may interfere with the protein trafficking?

Unfortunately, we were not able to address this point due to the natural biology of primary microglia and the challenge posed by their expression of Fc-receptors.

In our internalization assay, we used VSV-G antibody labelling and C-terminally tagged EGFP. The advantage of this strategy is that we are not only relying on the fused EGFP tag, which shows receptors on the cell membrane but also labels receptors retained within the cell (potential sites are ER, Golgi, lysosomes), as is shown in **Figure 5b**. By using the VSV-G labelling to quantify internalization, we circumvent false positive results. At the same time, we can use co-localization with EGFP to confirm that VSV-G signals are derived from labelled receptors.

Conducting this experiment in primary microglia is challenged by the fact that these cells are nonspecifically labeled with VSV-G antibodies, probably due to the expression of Fc-receptors. The image below exemplifies false positive VSV-G antibody labeling in untreated primary microglia. Even with the use of Fc-receptor blockers, we were unable to mitigate this unspecific signal enough to obtain a clean staining required for reliable quantification.

Primary microglia without any DREADD-GPCR expression stained for VSV-G under non-permeabilizing conditions

As an alternative strategy, we were considering to perform the internalization assay in our HMC3 cell lines with stable DREADD- β 2AR expression, but found this model system unsuitable compared to HEK cells. The advantage of HEK cells is their spherical shape, which allows a straightforward separation of signals from surface-expressed receptor or receptors that have been internalized (see **Fig. 5c**). However, HMC3 cells are mostly flat and do not allow optical slicing along the z-axis without including the surface. We include an orthogonal view below as example, showing the flat shape of HMC3 cells in YZ and XZ, which obscures the extra- or intracellular origin of the signal.

4. It has been reported that increasing cAMP by activating β 2AR promotes filopodia but collapses large processes (PMID: 31167136). Similarly, inhibition of β 2AR increase microglial process surveillance in vivo (PMID: 31636449). I understand the technical challenge for the authors to identify filopodia and large processes in culture microglia. However, they might be able to use images in Figure 5 (now Fig. 6) with higher magnification to analyze filopodia dynamics. At least they should discuss the caveat of the functional readout of filopodia in culture microglia (instead of microglial processes in vivo).

Unfortunately, the majority of primary microglia did not display defined processes that would allow a more sophisticated analysis. We found that filopodia in our microglia cultures were mostly present as a rim surrounding the cell body, which extended/enlarged upon β 2AR stimulation and lead to an increase in total cell area.

However, we investigated the dynamics of microglia motility in our data from Figure 6 and calculated a surveillance index (doi.org/10.1002/glia.23719). This index measures the number of image pixels surveyed per unit time. The image below shows the results of this analysis and did not point to altered motility profiles:

In the revised manuscript, we included the following statement in the results section to point out the discrepancy of filopodia extension in cultured cells compared to the more complex dynamics *in-vivo*:

“It is worth mentioning that filopodia extension in cultured microglia does not present the complexity of microglial process dynamics observed in-vivo^{45,49}. Yet, given the difficulty of microglial transduction in-vivo⁶⁹, our simpler but more accessible in-vitro system suggests that DREADD-β2AR successfully mimics β2AR signaling in microglia and modulates their function.”

- Figure 7 (now Fig. 8) used HMC3 cells for the DREADD-based chimeras modulation of microglia under inflammatory conditions. The reason to choose this microglia-like cell line, is because of suboptimal transduction efficacies with available vectors the authors mentioned. However, the authors did successfully package DREADD-β2AR-P2A-EGFP construct into lentiviral vectors and transduced primary microglia to observe filopodia formation. The authors should use primary microglia to manipulate chimeric GRCP in gene expression under inflammatory conditions.

We agree with the reviewer’s statement that it would be exciting to perform this experiment in primary microglia. However, we had to decide against primary microglia as *in-vitro* model due to the sparse transduction efficiency. We included an image in **Supplementary Figure S6c** to exemplify suboptimal transduction efficiency. Our RT-qPCR results show modulation of IL6 and TNF, but the magnitude of the effect only operates in the range of a 2- to 3-fold change. Reliable quantification of such responses is challenging if less than 50% of the cell population expresses the receptor. On the other hand, our stable HMC3 cell lines reliably expressed DREADD-GPCRs allowing us to robustly detect hundreds of genes after inflammatory stimulation with mRNA sequencing.

6. In the neuropathic pain model, activating microglial Gi-DREADD could decrease IL-1 β expression (PMID: 33221486), while in this study, activating DREADD-GPR109 did not. In addition, activation of DREADD- β 2AR or DREADD-GPR65 did not change IL1 β expression. Please discuss the discrepancy and how G protein signaling regulates the expression of inflammatory genes.

The reviewer points out an interesting observation. We feel that it is beyond the scope of our data or this study to pin down the precise mechanisms underlying such discrepancies. On the other hand, the response diversity might be partly the results of different model systems (cell lines, mouse models), which might have a unique response to certain signaling pathways. At the same time, we speculate that the mode by which inflammation/tissue damage is induced plays an important role.

[REDACTED]

We included the following statement in the discussion, to highlight cell line-specific factors and that certain inflammatory genes might be differently effected in the various *in-vitro* systems:

*“Following CNO stimulation, DREADD-GPR65 modified inflammatory gene expression similar to β 2AR (Fig. 8d), whereas DREADD-GPR109A did not alter IL6, TNF, or IL1 β mRNA levels during IFN γ /IL1 β -induced inflammation (Fig. 8e). A previous study⁷² observed an anti-inflammatory effect of GPR65 in primary mouse microglia by inhibiting LPS-induced IL1 β expression after acidification of the culture medium. Our results suggest that this effect is not present in HMC3 cells when using the cytokines IFN γ /IL1 β to trigger inflammation. Another study⁷³ reported an anti-inflammatory role of GPR109A in the murine N9 microglia-like cell line by downregulating LPS-induced TNF and IL1 β after dimethyl fumarate treatment, an immunomodulatory drug and GPR109A agonist. Such discrepancies highlight the response diversity with different *in vitro* systems and inflammatory mediators^{72,73}.”*

A: Schematic showing how optic nerve crush serves as a model for microglia activation. **B:** Experimental timeline and injection paradigm. hM4Di signaling was stimulated through daily CNO injection for five days following optic nerve crush (ONC). **C:** Example images showing the quantification of microglial CD68 expression through surface renderings. Microglia are immunostained with the macrophage marker Iba1. **D:** Representative maximum intensity projections of retinal microglia in the inner plexiform layer (IPL). Bottom panels provide a zoomed-in view of the area indicated by the respective white squares. The CD68 channel has been masked through surface rendering to only show signal within microglia. **E:** Quantification of microglial CD68. Graph shows percentage of total microglia surface volume occupied by CD68. Circles and triangles indicate male and female animals, respectively. Error bars represent standard error of the mean of four animals. Linear regression analysis: $p^* < 0.05$; $p^{n.s.} > 0.05$.

Reviewer #3 (Remarks to the Author):

This is an interesting and potentially important paper which expands the chemogenetic toolkit to include potential orphan GPCR constructs which appear useful for interrogating their biology.

Specific comments:

1. A Gs-DREADD has previously been described based on the b1-AR and I'm surprised it wasn't mentioned (ref: <https://www.pnas.org/content/106/45/19197.long>). Thus the b2-AR GS DREADD is not unique but might be useful and this should be mentioned.

We thank the reviewer for pointing out the importance to mention the rM3Ds DREADD. This $G\alpha_s$ -coupled DREADD has been created by replacing ICL2 and ICL3 of the rat equivalent of hM3Dq with the corresponding domains of turkey β 1AR. As such, it represents a DREADD-GPCR and bears similarities to our approach. Our strategy differs by including all three ICL domains (ICL1-3) and the C-terminus for which we generated a library of 292 potential GPCRs of interest. To highlight the difference between the rM3Ds and our approach, we included the following statement in the discussion:

“We generated a table with the protein sequences of putative signaling domains for all 292 GPCRs-of-interest included in our alignment, separated into ICL1-3 and C-terminus (Supplementary Table S1). These sequences can be inserted in-between the hM3Dq ligand binding domains as outlined in Supplementary Figure S16 to generate a CNO-responsive chimera mimicking a GPCR-of-interest, which provides a framework for straightforward in-silico design. Thereby, our approach also complements the previously published DREADD chimera rM3Ds⁴³, which has been generated by combining the rat equivalent of hM3Dq with ICL2-3 from turkey β 1AR to achieve canonical $G\alpha_s$ -coupling. Our design strategy utilizes all signaling domains including ICL1-3 and C-terminus for high fidelity recapitulation of a GPCR-of-interest. Our library of signaling domains also offers the means to create a large variety of possible chimeras and to evaluate the contribution of different ICLs and C-termini to certain pathways.”

It is likely that rM3Ds performs similarly to our DREADD- β 2AR as far as canonical $G\alpha_s$ signalling is concerned. To address this, we included rM3Ds in our validation on the level of cAMP and MAPK (SRE reporter) activity in our revised manuscript (Fig .2e-f, i, respectively). As expected, rM3Ds increased cAMP and decreased SRE reporter activity like DREADD- β 2AR. We found that DREADD- β 2AR mimicked non-chimeric β 2AR more closely and induced similar fold changes.

2. The previously described GS-DREADD (from the Wess group at NIH) had constitutive activity; it is not at all clear that constitutive activity was measured at any of these constructs and I'd be surprised if they do not display at least some basal activity. The standard way to measure this is to vary the expression of the construct and to measure basal activity (this would require another type of cAMP assay) and compare with a receptor relatively devoid of constitutive activity (see Journal of Biological Chemistry 268 (7), 4625-4636).

The paper that introduced rM3Ds (<https://www.pnas.org/content/106/45/19197.long>) used a cAMP-dependent ELISA assay to measure baseline cAMP levels. We have previously used such assays but due to pandemic-related delivery issues we are unable to obtain them at the moment. Thus, we decided for the cAMP-dependent 22F luciferase reporter (doi: 10.1016/j.bcp.2015.09.010), which is suitable for baseline comparisons. We evaluated

constitutive activity against empty vector-transfected HEK cells, our DREADD-based chimeras, and non-chimeric β 2AR (**Fig. 3b-c, Supplementary Fig. S8b-c**). We were able to reproduce the elevated cAMP baseline levels of rM3Ds. In comparison, our DREADD- β 2AR showed lower constitutive cAMP activity.

Additionally, we also investigated constitutive kinase activity on the MAPK pathway through evaluating SRE reporter signals against empty vector controls lacking DREADD-GPCR expression (**Fig. 3e, Supplementary Fig. S8e**). Again, rM3Ds showed high constitutive activity and inhibited SRE reporter activity stronger than our DREADD- β 2AR.

3. In Fig 3 (now **Fig. 4**) it appears there is some basal ERK data for the DREADD. This should be tested statistically (the figure looks to be N=1 which is not suitable)

We are thankful for this comment, which made us aware that the original graph was not reflecting the number of repeated experiments that we show in **Supplementary Figure S4**. We have revised the graph which is now found in **Figure 4c** to show the number of replicates appropriately by showing individual data points. We also included a link in the figure legend to **Supplementary Figure S4**.

We now also include **Supplementary Figure S5**, which focuses on untreated samples (reflecting basal conditions) and confirmed that basal ERK1/2 phosphorylation is present.

REVIEWER COMMENTS

Reviewer #1 (Remarks to the Author):

The authors have done a good job of addressing the questions raised in the review with clarifications, added information, expanded analysis and added experiments.

A main remaining issue is about the advantage of the new chimeric constructs between specific GPCRs and a DREADD: do the chimeras have unique properties that make a step-forward for chemogenetic control of GPCR signaling?

The goal of this paper is to generate new DREADD versions of GPCRs that have important functions in microglia. The authors swap the internal loops and C-terminal of receptors of interest onto the CNO-activated DREADD. The DREADD chimeras recapitulate the signaling of the expected receptor: Gs for B2R and GPR65 and Gi for GPR109A. The activation of DREADD-B2R and DREADD-GPR65 leads to robust effects, but DREADD-GPR109A is weak (adjustment of the presentation in Fig. 7e is helpful, but still leaves clear the relative weakness compared to the two Gs chimeras).

A key question (asked by this reviewer and reviewer 2) is whether the chimeras provide new tools that did not exist before.

-DREADD-GPR109A is weak, will it be useful?

-DREADD-B2R and DREADD-GPR65 give strong signaling, but are they better mimics of signaling by the native receptor than the previously described Gs-DREADD (rM3Ds)?

The place to look would be in target cells of interest (e.g. microglia). The authors provide new data, but from HEK cells. The data indicate that the chimeras behave similarly to the established Gs-DREADD, but that DREADD- β 2AR and β 2AR stimulate with similar ~30-fold change compared to half that amount with rM3Ds. This difference could be accounted for by superior efficacy, and could be meaningful if true also in the target cell. But other factors could explain the result, such as a difference in expression level. The new data do not resolve the question.

Another way of addressing the question about special properties of the new chimeras is with transcriptomics. Do the chimeras act like their parent receptors and differ from rM3Ds? The authors addressed this with next generation mRNA sequencing of HMC3 cell lines under inflammatory conditions. They report a strong correlation between the signatures of β 2AR, DREADD- β 2AR and DREADD-GPR65 across more than 200 genes. This analysis is to be commended for extending well beyond the initial analysis of 3 genes. However, the findings suggest that it does not matter which Gs-coupled DREADD one uses. So the question remains unanswered.

Reviewer #2 (Remarks to the Author):

Very much appreciate the authors efforts in revising the manuscript. They have addressed almost all my questions and the study is significantly improved with the new results. The study will provide additional chemogenetic tools in the field. I have a few remaining minor concerns:

1. Even though the targeted GPCRs are highly expressed in microglia, currently these new chemogenetic tools are only applied in microglia in vitro. This limited the overall significance of the study. The generation of transgenic mice with microglia specific expression will be needed to validate these tools in vivo.

2. I am not convinced that DREADD-beta2AR mimics beta2AR more closely. The extracellular domain and transmembrane domain of DREADD-beta2AR are based on the hM3Dq. Please check whether hM3Dq and rM3Ds have different CNO binding affinities. If they are different, it may not be appropriate to use 10um CNO and directly compare the results from the cAMP assay. The authors should tone down the claim.

3. I appreciate that the authors included the analysis of microglia motility in their revised manuscript. However, isn't it surprising that activating beta2AR increases the total area of microglia but not motility?

Reviewer #3 (Remarks to the Author):

The authors have addressed my relatively minor comments fully.

Point-by-point response to the reviewers' comments

Reviewer's comments:	black
Our reply:	green
References to text adaptations:	orange

Reviewer #1 (Remarks to the Author):

The authors have done a good job of addressing the questions raised in the review with clarifications, added information, expanded analysis and added experiments.

A main remaining issue is about the advantage of the new chimeric constructs between specific GPCRs and a DREADD: do the chimeras have unique properties that make a step-forward for chemogenetic control of GPCR signaling?

The goal of this paper is to generate new DREADD versions of GPCRs that have important functions in microglia. The authors swap the internal loops and C-terminal of receptors of interest onto the CNO-activated DREADD. The DREADD chimeras recapitulate the signaling of the expected receptor: Gs for B2R and GPR65 and Gi for GPR109A. The activation of DREADD-B2R and DREADD-GPR65 leads to robust effects, but DREADD-GPR109A is weak (adjustment of the presentation in Fig. 7e is helpful, but still leaves clear the relative weakness compared to the two Gs chimeras).

1. A key question (asked by this reviewer and reviewer 2) is whether the chimeras provide new tools that did not exist before.

We would like to clarify that our intention has not been to replace the existing DREADD system by engineering a new generation of receptors. Instead, we provide with the CNO-responsive GPCR chimeras possibilities to specifically addresses orphan receptors with unidentified ligands. This group includes more than 100 potential drug targets as well as the majority of olfactory receptors¹⁻³. At the same time, our DREADD-GPCRs offers a strategy to explore GPCRs-of-interest for their potential non-canonical signaling cascades that are likely not captured with the currently available DREADDs.

We validated the principles with three GPCRs-of-interest that were selective for microglia, however, we also included a straightforward design for the other 292 orphan receptors (**Supplementary Table S1, Supplementary Fig. S17**).

In the revised manuscript, we included the following statement:

“In our study, we offer a straightforward design for CNO-responsive chimeras to mimic a GPCR-of-interest. This will be especially useful to study GPCRs with yet unidentified pathways, orphan receptors with unknown ligands¹⁻³, or GPCRs with non-canonical signaling properties that might not be captured by available DREADDs.”

- DREADD-GPR109A is weak, will it be useful?

We engineered DREADD-GPR109A as a microglial GPCR-of-interest and to provide proof-of-concept that our strategy can be extended to GPCRs coupled to other signaling pathways. Our data with DREADD-GPR109A suggest that we were indeed able to trigger the expected $G\alpha_i$ response⁴ by suppressing cAMP synthesis and competition with $G\alpha_s$ (Fig. 7b, e-g). As the reviewer pointed out, the magnitude of this response was weak compared to DREADD- β 2AR/DREADD-GPR65 (Fig. 7c versus Fig. 7e). A possible explanation might be the technical aspects underlying the quantification of $G\alpha_i$ signaling⁵.

“DREADD-GPR65 and DREADD-GPR109A triggered their expected cascades by either inducing (Fig. 7c-d) or inhibiting (Fig. 7e-g) $G\alpha_s$ signaling, respectively. As a note, the comparatively weaker DREADD-GPR109A response is in line with the technical challenges of quantifying inhibitory effects on cAMP⁵.”

- DREADD-B2R and DREADD-GPR65 give strong signaling, but are they better mimics of signaling by the native receptor than the previously described Gs-DREADD (rM3Ds)? The place to look would be in target cells of interest (e.g. microglia). The authors provide new data, but from HEK cells. The data indicate that the chimeras behave similarly to the established Gs-DREADD, but that DREADD- β 2AR and β 2AR stimulate with similar ~30-fold change compared to half that amount with rM3Ds. This difference could be accounted for by superior efficacy, and could be meaningful if true also in the target cell. But other factors could explain the result, such as a difference in expression level. The new data do not resolve the question. Another way of addressing the question about special properties of the new chimeras is with transcriptomics. Do the chimeras act like their parent receptors and differ from rM3Ds? The authors addressed this with next generation mRNA sequencing of HMC3 cell lines under inflammatory conditions. They report a strong correlation between the signatures of β 2AR, DREADD- β 2AR and DREADD-GPR65 across more than 200 genes. This analysis is to be commended for extending well beyond the initial analysis of 3 genes. However, the findings suggest that it does not matter which Gs-coupled DREADD one uses. So the question remains unanswered.

We thank the reviewer for this valuable point. Indeed, our initial RNA sequencing data in microglia-like cells shows that both, DREADD- β 2AR and DREADD-GPR65, similarly shift the overall inflammatory response and highly correlated with the endogenous β 2AR signature. We speculate that this high degree of similarity is initiated through the

canonical $G\alpha_s$ signaling in modulating inflammatory gene expression. We included the following sentence in the discussion:

“The highly correlated signatures of $\beta 2AR$, DREADD- $\beta 2AR$ and DREADD-GPR65 suggest that canonical $G\alpha_s$ -coupling modulates $IFN\gamma/IL1\beta$ -mediated inflammation in a similar manner.”

To identify unique features of the two $G\alpha_s$ -coupled chimeras, we revisited our RNA sequencing data set and directly compared the DREADD- $\beta 2AR$ and DREADD-GPR65 signatures. While the majority of genes were affected in the same manner, hierarchical clustering revealed genes that were different between DREADD- $\beta 2AR$ and DREADD-GPR65 (**Supplementary Fig. S16a**).

To compare this expression pattern with another $G\alpha_s$ -coupled receptor, we established for this revision a microglia-like cell line with stable rM3Ds expression (**Supplementary Fig. 16c**) and performed RT-qPCR to validate the signature genes from this hierarchical clustering. We found that DREADD- $\beta 2AR$ closely recapitulated the endogenous $\beta 2AR$ response while rM3Ds induced a different transcriptional profile (**Supplementary Fig. S16d-e**). Our new findings exemplify that DREADD-based chimeras coupled to the same canonical pathway can induce unique transcription profiles despite their ability to modulate the overall response in the same direction.

We included this new dataset into the **Supplementary Fig. S16** of the revised manuscript and included the following text in the result and discussion section:

“Even though DREADD- $\beta 2AR$ and DREADD-GPR65 had highly correlated signatures due to canonical $G\alpha_s$ -coupling, we found unique features upon hierarchical clustering of their respective response signatures (**Supplementary Fig. S16a, Supplementary Table S2**). A small set of genes located in cluster A and C exhibited a distinct expression pattern that not particularly enriched for a distinct biological pathway (**Supplementary Fig. S16b**). To compare this unique response with another $G\alpha_s$ -coupled GPCR, we generated stable rM3Ds-expressing HMC3 cells (**Supplementary Fig. S16c**) and performed RT-qPCR on cluster A and C genes. In addition, we included the three inflammatory genes IL6, TNF and IL1 β , which we have quantified in our previously established HMC3 cell lines (**Fig. 8**). We found that the rM3Ds response was distinguished from endogenous $\beta 2AR$, DREADD- $\beta 2AR$ and DREADD-GPR65 based on principal component analysis (**Supplementary Fig. S16d**) and hierarchical clustering (**Supplementary Fig. S16e**). At the same time, DREADD- $\beta 2AR$ intermingled with endogenous $\beta 2AR$ and was separated from DREADD-GPR65 (**Supplementary Fig. S16d-e**). Notably, rM3Ds also differed in the expression pattern of TNF and IL1 β . rM3Ds did not induce the robust downregulation of TNF but instead increased IL1 β , which was not affected by endogenous $\beta 2AR$ and our DREADD-GPCRs (**Supplementary Fig. S16e**). Upregulation of IL6 was similar between rM3Ds and the other receptors, indicating that this might be a more conserved feature of $G\alpha_s$ signaling. Our data suggest that $G\alpha_s$ -driven modulation of gene expression can display subtle differences depending on individual GPCRs and show that DREADD- $\beta 2AR$ successfully mimics endogenous $\beta 2AR$ more closely compared to rM3Ds.”

“Interestingly, we identified a unique transcriptional signature for a small set of genes for each DREADD-chimera. We confirmed these genes with RT-qPCR and confirmed that this expression pattern is distinct from $G\alpha_s$ -coupled rM3Ds (Supplementary Fig. S16). Our data shows that DREADD-based chimeras coupled to the same canonical pathway are capable of recapitulating unique transcriptional profiles.”

Reviewer #2 (Remarks to the Author):

Very much appreciate the authors efforts in revising the manuscript. They have addressed almost all my questions and the study is significantly improved with the new results. The study will provide additional chemogenetic tools in the field. I have a few remaining minor concerns:

1. Even though the targeted GPCRs are highly expressed in microglia, currently these new chemogenetic tools are only applied in microglia *in vitro*. This limited the overall significance of the study. The generation of transgenic mice with microglia specific expression will be needed to validate these tools *in vivo*.

We agree with the reviewer that the next step would be to generate microglia *in-vivo* models to identify the individual role of the GPCR-of-interest. On the other hand, our strategy is not limited to microglia and can be applied to any other cell type that expresses the GPCR-of-interest. Therefore, we included a statement at the end of the discussion:

“While microglia *in-vitro* models are critical for neuroimmunological research, it is important to note that different culture systems display distinct genetic signatures and only partially reflect the phenotype of microglia *in-vivo* ⁶. Therefore, it would be ultimately desirable to apply DREADD-chimeras in an *in-vivo* context. However, *in-vivo* targeting of microglia is a major challenge within the field due to a current lack of efficient and specific vectors ⁷. Yet, GPCR signaling is critical for many other cell types that might be more accessible for chimeric GPCR expression. Our strategy complements existing methods for GPCR investigation and offers an alternative approach to dissect GPCR signaling in various contexts and model systems.”

2. I am not convinced that DREADD-beta2AR mimics beta2AR more closely. The extracellular domain and transmembrane domain of DREADD-beta2AR are based on the hM3Dq. Please check whether hM3Dq and rM3Ds have different CNO binding affinities. If they are different, it may not be appropriate to use 10 μ M CNO and directly compare the results from the cAMP assay. The authors should tone down the claim.

We thank the reviewer for this comment and we performed the suggested experiment (Supplementary Fig. S3b). We stimulated DREADD- β 2AR and rM3Ds with different CNO concentrations and found that both responses are at their maximum in the presence of 10 μ M (determined through induction of cAMP synthesis upon CNO application). This suggests that the results in Figure 2 can be compared with each other.

We included the following text in the result section:

“The DREADD-β2AR and rM3Ds responses were both saturated at 10μM CNO (**Supplementary Fig. S3b**), suggesting that the data in **Figure 2e-g** reflect their maximal cAMP induction capabilities.”

As suggested, we also toned down the claim that DREADD-β2AR mimics non-chimeric β2AR more closely based on our validation in HEK cells. In the discussion section, we now state the following:

“Additionally, we compared DREADD-β2AR with rM3Ds and found that both GPCRs imitated the cAMP and MAPK activity of β2AR with high fidelity (**Fig. 2e-g, i**) while our construct displayed lower constitutive activity (**Fig. 3**).”

3. I appreciate that the authors included the analysis of microglia motility in their revised manuscript. However, isn't it surprising that activating beta2AR increases the total area of microglia but not motility?

Unfortunately, we cannot offer a definitive explanation for this observation but we might speculate that while filopodia formation increases cell area it does not alter the overall ability of primary microglia to remain their initial mobility.

Reviewer #3 (Remarks to the Author):

The authors have addressed my relatively minor comments fully.

References

1. Tang, X., Wang, Y., Li, D., Luo, J. & Liu, M. Orphan G protein-coupled receptors (GPCRs): biological functions and potential drug targets. *Acta Pharmacologica Sinica* **33**, 363–371 (2012).
2. Alexander, S. P. H. *et al.* Class A Orphans (version 2020.5) in the IUPHAR/BPS Guide to Pharmacology Database. *IUPHAR/BPS Guide to Pharmacology CITE* **2020**, (2020).
3. Bikle, D., Bräuner-Osborne, H., Brown, E. M., Conigrave, A. & Shoback, D. Class C Orphans (version 2019.4) in the IUPHAR/BPS Guide to Pharmacology Database. *IUPHAR/BPS Guide to Pharmacology CITE* **2019**, (2019).
4. Stefan Offermanns, Steven L. Colletti, Adriaan P. IJzerman, Timothy W. Lovenberg, Graeme Semple, A. W. IUPHAR/BPS Guide to PHARMACOLOGY: GPR109A. Available at: <http://www.guidetopharmacology.org/GRAC/ObjectDisplayForward?objectId=312>. (Accessed: 30th August 2017)
5. Gilissen, J. *et al.* Forskolin-free cAMP assay for Gi-coupled receptors. *Biochemical Pharmacology* (2015). doi:10.1016/j.bcp.2015.09.010
6. Butovsky, O. *et al.* Identification of a unique TGF-β-dependent molecular and functional signature in microglia. *Nature Neuroscience* (2014). doi:10.1038/nn.3599
7. Maes, M. E., Colombo, G., Schulz, R. & Siegert, S. Targeting microglia with lentivirus and AAV: Recent advances and remaining challenges. *Neuroscience Letters* (2019). doi:10.1016/j.neulet.2019.134310

REVIEWER COMMENTS

Reviewer #2 (Remarks to the Author):

The authors have fully addressed my concerns.

Black: Reviewer's comment

Green: Our reply

REVIEWERS' COMMENTS

Reviewer #2 (Remarks to the Author):

The authors have fully addressed my concerns.

We thank the Reviewer for the overall comments along the revision.